# Pretrained Transformer Efficiently Learns Low-Dimensional Target Functions In-Context

**Kazusato Oko**[1,3], **Yujin Song**[2,3], **Taiji Suzuki**[2,3], **Denny Wu**[4,5]

[1]University of California, Berkeley,  [2]University of Tokyo,  [3]RIKEN AIP
[4]New York University,  [5]Flatiron Institute

oko@berkeley.edu, song-yujin139@g.ecc.u-tokyo.ac.jp,
taiji@mist.i.u-tokyo.ac.jp, dennywu@nyu.edu

## Abstract

Transformers can efficiently learn in-context from example demonstrations. Most existing theoretical analyses studied the in-context learning (ICL) ability of transformers for linear function classes, where it is typically shown that the minimizer of the pretraining loss implements one gradient descent step on the least squares objective. However, this simplified linear setting arguably does not demonstrate the statistical efficiency of ICL, since the pretrained transformer does not outperform directly solving linear regression on the test prompt. In this paper, we study ICL of a nonlinear function class via transformer with nonlinear MLP layer: given a class of *single-index* target functions $f_*(\boldsymbol{x}) = \sigma_*(\langle \boldsymbol{x}, \boldsymbol{\beta} \rangle)$, where the index features $\boldsymbol{\beta} \in \mathbb{R}^d$ are drawn from a $r$-dimensional subspace, we show that a nonlinear transformer optimized by gradient descent (with a pretraining sample complexity that depends on the *information exponent* of the link functions $\sigma_*$) learns $f_*$ in-context with a prompt length that only depends on the dimension of the distribution of target functions $r$; in contrast, any algorithm that directly learns $f_*$ on test prompt yields a statistical complexity that scales with the ambient dimension $d$. Our result highlights the adaptivity of the pretrained transformer to low-dimensional structures of the function class, which enables sample-efficient ICL that outperforms estimators that only have access to the in-context data.

## 1 Introduction

Pretrained transformers [VSP+17] possess the remarkable ability of *in-context learning (ICL)* [BMR+20], whereby the model constructs a predictor from a prompt sequence consisting of pairs of labeled examples without updating any parameters. A common explanation is that the trained transformer can implement a learning algorithm, such as gradient descent on the in-context examples, in its forward pass [HvMM+19, DSD+22, VONR+23]. Such explanation has been empirically studied in synthetic tasks such as linear regression [GTLV22, ASA+22], which has motivated theoretical analyses of the statistical and computational complexity of ICL in learning simple function classes.

Many recent theoretical works on ICL focus on learning the function class of *linear models* using linear transformers trained via empirical risk minimization. In this setting, it can be shown that minima of the pretraining loss implements one (preconditioned) gradient descent step on the least squares objective computed on the test prompt [ZFB23, ACDS23, MHM23]. This implies that the forward pass of the pretrained transformer can learn linear targets with $n \gtrsim d$ in-context examples, hence matching the sample complexity of linear regression on the test prompt. Subsequent works also studied how the distribution and "diversity" of pretrained tasks affect the ICL solution in similar problem settings [WZC+23, RPCG23, ZWB24, LLZV+24].

38th Conference on Neural Information Processing Systems (NeurIPS 2024).

The motivation of our work is the observation that the simple setting of learning linear models with linear transformers does not fully capture the statistical efficiency and adaptivity of ICL. Specifically,

- A linear transformer has limited expressivity and hence only implements simple operations in the forward pass, such as one gradient step for least squares regression. This limits the class of algorithms that can be executed in-context, and consequently, the pretrained transformer cannot outperform directly solving linear regression on the test prompt. We therefore ask the question:

  *With the aid of MLP layer, can a pretrained transformer learn a nonlinear function class in-context, and outperform baseline algorithms that only have access to the test prompt?*

- A key feature of ICL is the *adaptivity* to structure of the learning problem. For example, [GTLV22] empirically showed that transformers can match the performance of either ridge regression or LASSO, depending on parameter sparsity of the target class; [RPCG23] observed that transformers transitions from a weighted-sum estimator to ridge regression as the number of pretraining tasks increases. Such adaptivity enables the pretrained model to outperform algorithms that only have access to the test prompt, which cannot take into account the "prior" distribution of target functions. Hence a natural question to ask is

  *Can a pretrained transformer adapt to certain structures of the target function class, and how does such adaptivity contribute to the statistical efficiency of ICL?*

## 1.1 Our Contributions

### 1.1.1 Function Class: Gaussian Single-index Models

To address the above questions, we study the in-context learning of arguably the simplest *nonlinear* extension of linear regression where a nonlinearity is applied to the response. Specifically, we consider the following class of *single-index* target functions, where the $t$-th pretraining task is constructed as

$$\boldsymbol{x}_1^t, \boldsymbol{x}_2^t, \ldots, \boldsymbol{x}_N^t, \boldsymbol{x}^t \overset{\text{i.i.d.}}{\sim} \mathcal{N}(0, \boldsymbol{I}_d), \quad y_i^t = \sigma_*^t(\langle \boldsymbol{x}_i^t, \boldsymbol{\beta}^t \rangle) + \varsigma_i^t, \tag{1.1}$$

where $\sigma_*^t : \mathbb{R} \to \mathbb{R}$ is the unknown link function, and $\boldsymbol{\beta}^t \in \mathbb{R}^d$ is the index feature vector which is drawn from some fixed $r$-*dimensional subspace* for some $r \ll d$. The task is to learn this input-output relation by reading the context $(\boldsymbol{x}_1, y_1, \ldots, \boldsymbol{x}_N, y_N)$ and predict the output corresponding to the query $\boldsymbol{x}$ (See Section 2 for details). This problem setting is based on the following considerations.

- **Nonlinearity of function class.** Due to the nonlinear link function, single-index targets cannot be learned by linear transformers. For this class of functions, the statistical efficiency of simple algorithms has been extensively studied: given a link function with degree $P$ and information exponent $Q$ (defined as the index of the smallest non-zero coefficient in the Hermite expansion), we know that kernel methods require at least $n \gtrsim d^P$ samples [GMMM21, DWY21], whereas two-layer neural network trained by gradient descent can achieve better sample complexity $n \gtrsim d^{\Theta(Q)}$ [BAGJ21, BBSS22]. Hence these algorithms, if directly applied to the test prompt, require a context length *polynomial* in the ambient dimensionality $d$, which arguably deviates from practical settings of ICL where the pretrained transformer learns from a few in-context examples which may come from high dimensional data space. These algorithms serve as a baseline for comparing the statistical efficiency of ICL.

- **Two types of low-dimensional structures.** Prior works on gradient-based feature learning highlights the adaptivity of neural network to *low-dimensional functions*, i.e., single-index model $\sigma_*(\langle \boldsymbol{x}_i, \boldsymbol{\beta} \rangle)$ that depends on one direction (index features) $\boldsymbol{\beta}$ in $\mathbb{R}^d$ [AAM22, BES+22, BMZ23]. In such setting, the complexity of gradient descent is dominated by the search for direction $\boldsymbol{\beta} \in \mathbb{R}^d$, the difficulty of which can be characterized by various computational lower bounds [DLS22, AAM23, DPVLB24].
  Importantly, our pretraining problem setup introduces *another notion of low dimensionality*: the "distribution" of target functions is low-dimensional, since the index features for each task $\boldsymbol{\beta}_t$ are drawn from a rank-$r$ subspace. This low-dimensionality of *function class* cannot be exploited by any algorithms that directly estimate the target function from test prompt, but as we will see, transformers can adapt to this additional structure via gradient-based pretraining, which reduces the search problem (for the index features $\boldsymbol{\beta}$) to $r$-dimensional in the in-context phase. Therefore, when $r \ll d$, we expect pretrained transformers to outperform baseline algorithms on the in-context examples (kernel methods, neural network, etc.).

**Empirical observations.** We pretrain a GPT-2 model [RWC+19] (with the same configurations as the in-context linear regression setting in [GTLV22]) to learn the Gaussian single-index task (1.1) with degree-3 link function, and compare its in-context sample complexity against baseline algorithms (see Section 4 for details). In Figure 1 we observe that the pretrained transformer achieves low prediction risk using fewer in-context examples than two baseline algorithms: kernel ridge regression, and neural network trained by gradient descent. Moreover, we observe that unlike the baseline methods,

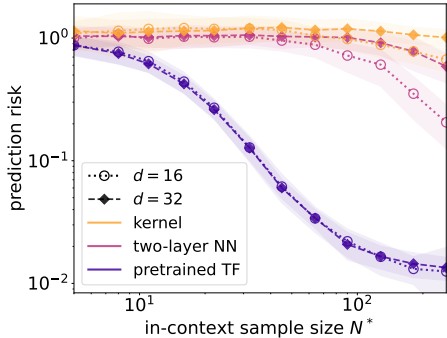

Figure 1: In-context generalization error (with standard deviation) of kernel ridge regression, neural network + gradient descent, and pretrained transformer. The target function is a polynomial single-index model. We fix $r = 8$ and vary $d = 16, 32$.

*Transformer achieves an in-context sample complexity (almost) independent of the ambient dimensionality $d$.*

The goal of this work is to rigorously establish a $d$-independent in-context sample complexity in an idealized theoretical setting for a shallow transformer optimized via gradient-based pretraining.

### 1.1.2 Main Result: Learning Single-index Models In-Context

We characterize the sample complexity of learning (1.1) in-context, using a nonlinear transformer optimized by gradient descent. Each single-index task is specified by an unknown index feature vector $\boldsymbol{\beta} \in \mathbb{R}^d$ drawn from some $r$-dimensional subspace, and a link function $\sigma_*$ with degree $P$ and information exponent $Q \leq P$; we allow the degree and information exponent to vary across tasks, to model the scenario where the difficult of pretraining tasks may differ. We show that pretraining of the nonlinear MLP layer can extract the low-dimensional structure of the function class, and the attention layer efficiently approximates the nonlinear link function. Our main theorem upper bounds the in-context generalization error of the pretrained transformer.

**Theorem** (Informal). *Let $f : (\boldsymbol{x}_1, y_1, \ldots, \boldsymbol{x}_N, y_N, \boldsymbol{x}) \mapsto y$ be a transformer with nonlinear MLP layer pretrained with gradient descent (Algorithm 1) on the single-index regression task (1.1). With probability at least 0.99, the model $f$ achieves in-context prediction risk $\mathbb{E}|f(\boldsymbol{x}) - f_*(\boldsymbol{x})| - \tau = o_d(1)$, where $\tau$ is the noise level, if the number of pretraining tasks $T$, the number of training examples $N$, the test prompt length $N^*$, and the network width $m$ satisfy (we omit polylogarithmic factors)*

$$\underbrace{T \gtrsim d^{\Theta(Q)}, \quad NT \gg T \gtrsim d^{\Theta(Q)}}_{pretraining}, \quad \underbrace{N^* \gtrsim r^{\Theta(P)}}_{inference}, \quad \underbrace{m \gtrsim r^{\Theta(P)}}_{approximation},$$

*where $Q, P$ are the information exponent and the highest degree of link functions, respectively.*

To the best of our knowledge, this is the first end-to-end optimization and statistical guarantee for in-context regression of this nonlinear function class. We make the following remarks.

- The required sample size for *pretraining* $T, N$ scale with the ambient dimensionality $d$. In particular, the number of pretraining tasks $T$ is parallel to the complexity of learning a single-index model with information exponent $Q$ using a two-layer neural network. On the other hand, the sample complexity for the *in-context* phase only depends on the dimensionality of the function class $r \ll d$.

- Note that any estimator that only has access to the in-context examples requires $n \gtrsim d$ samples to learn the single-index model, as suggested by the information theoretic lower bound (e.g., see [MM18, BKM+19]). Therefore, when $r \ll d$, we see a separation between pretrained transformer and algorithms that directly learn from the test prompt, such as linear/kernel regression and neural network + gradient descent. This highlights the adaptivity (via pretraining) of transformers to low-dimensional structures of the target function class.

- Our analysis of pretraining reveals the following mechanism analogous to [GHM+23]: the nonlinear MLP layer extract useful features and adapt to the low-dimensionality of the function class, whereas the attention layer performs in-context function approximation on top of the learned features.

### 1.2 Related Works

**Theory of in-context learning.** Many recent works studied the in-context learning ability of transformers trained by gradient descent. [ZFB23, ACDS23, MHM23, WZC+23, ZWB24] studied

the training of linear transformer models to learn linear target functions in-context by implementing one gradient descent step in the forward pass. Similar optimization results have been established for looped linear transformers [GSR+24], transformers with SoftMax attention [HCL23, NDL24, CSWY24a, CSWY24b] or nonlinear MLP layer [KS24, LWL+24]. Our problem setting resembles [KS24], where a nonlinear MLP block is used to extract features, followed by a linear attention layer; the main difference is that we establish end-to-end guarantees on the optimization and sample complexity for learning a concrete nonlinear function class, whereas [KS24] focused on convergence of optimization. [CCS23] showed that transformers can learn nonlinear functions in-context via a functional gradient mechanism, but no statistical and optimization guarantees were given. If we do not take gradient-based optimization into account, the function class that can be implemented in-context by transformers has been characterized in many prior works [BCW+23, GHM+23, ZZYW23, JLLVR24, SGS+24, KNS24]. These results typically aim to encode specific algorithms (LASSO, gradient descent, etc.) in the forward pass or directly analyze the Bayes-optimal estimator.

**Gradient-based learning of low-dimensional functions.** The complexity of learning low-dimensional functions with neural network has been extensively studied in the feature learning theory literature. Typical target functions include single-index models [BAGJ21, BES+22, BBSS22, MHPG+23, DNGL23, BES+23] and multi-index models [DLS22, AAM22, AAM23, BBPV23]. While a shallow neural network can efficiently approximate such low-dimensional functions, the efficiency of gradient-based training is governed by properties of the nonlinearity $\sigma^*$. In the single-index setting, prior works established a sufficient sample size $n \gtrsim d^{\Theta(K)}$, where $K \in \mathbb{N}$ is the *information exponent* for algorithms utilizing correlational information [BAGJ21, BBSS22, DNGL23, MHWSE23], or the *generative exponent* for algorithms that employ suitable label transformations [DPVLB24, LOSW24, ADK+24, JMS24]. Moreover, $n \asymp d$ samples are information theoretically necessary without additional structural assumptions; this entails that estimators that only access the test prompt inevitably pay a sample size that scales with the ambient dimensionality. As we will see, pretrained transformers can avoid this "curse of dimensionality" by exploiting the low-dimensionality of the task distribution. Low-dimensional structure of the *function class* similar to our setting has been assumed to study the efficiency of transfer learning [DLS22] and multi-task learning [CHS+23].

## 2 Problem Setting

**Notations.** $\| \cdot \|$ denotes the $\ell_2$ norm for vectors and the $\ell_2 \to \ell_2$ operator norm for matrices. For a vector $\boldsymbol{w}$, we use $\boldsymbol{w}_{a:b}$ for $a \le b$ to denote the vector $[w_a, w_{a+1}, \ldots, w_b]^\top$. $\mathbf{1}_N$ denotes the all-one vector of size $N$. The indicator function of $A$ is denoted by $\mathbb{I}_A$. Let $N$ be a nonnegative integer; then $[N]$ denotes the set $\{n \in \mathbb{Z} \mid 1 \le n \le N\}$. For a nonnegative integer $i$, the $i$-th Hermite polynomial is defined as $\mathrm{He}_i(z) = (-1)^i e^{\frac{z^2}{2}} \frac{\mathrm{d}^i}{\mathrm{d}z^i} e^{\frac{-z^2}{2}}$. For a set $S$, $\mathrm{Unif}(S)$ denotes the uniform distribution over $S$. We denote the unit sphere $\{\boldsymbol{x} \in \mathbb{R}^d \mid \|\boldsymbol{x}\| = 1\}$ by $\mathbb{S}^{d-1}$. $\tilde{O}(\cdot), \tilde{\Omega}(\cdot)$ represent $O(\cdot)$ and $\Omega(\cdot)$ notations where polylogarithmic terms are hidden. We write $a \lesssim b$ when there exists a constant $c$ such that $a \le cb$. If both $a \lesssim b$ and $b \lesssim a$ holds, we write $a \asymp b$.

### 2.1 Data Generating Process

#### 2.1.1 In-context learning

We first introduce the basic setting of ICL [BMR+20] of simple function classes as investigated in [GTLV22, ASA+22]. In each *task*, the learner is given a sequence of inputs and outputs $(\boldsymbol{x}_1, y_1, \ldots, \boldsymbol{x}_N, y_N, \boldsymbol{x})$ referred to as *prompt*, where $\boldsymbol{x}_i, \boldsymbol{x} \in \mathbb{R}^d$ and $y_i \in \mathbb{R}$. The labeled examples $\boldsymbol{X} = (\boldsymbol{x}_1 \quad \cdots \quad \boldsymbol{x}_N) \in \mathbb{R}^{d \times N}$, $\boldsymbol{y} = (y_1 \quad \cdots \quad y_N)^\top \in \mathbb{R}^N$ are called *context*, and $\boldsymbol{x}$ is the *query*. Given input distribution $\boldsymbol{x}_1, \ldots, \boldsymbol{x}_N, \boldsymbol{x} \stackrel{\text{i.i.d.}}{\sim} \mathcal{D}_{\boldsymbol{x}}$, the output $y_i$ is expressed as

$$y_i = f_*(\boldsymbol{x}_i) + \varsigma_i, \quad i \in [N],$$

where $f_*$ is the true function describing the input-output relation and $\varsigma_i \stackrel{\text{i.i.d.}}{\sim} \mathcal{D}_\varsigma$ is i.i.d. label noise. Note that $f_*$ also varies across tasks — we assume $f_*$ is drawn i.i.d. from some true distribution $\mathcal{D}_{f_*}$. In the pretraining phase, we optimize the model parameters given training data from $T$ distinct tasks $\{(\boldsymbol{x}_1^t, y_1^t, \ldots, \boldsymbol{x}_M^t, y_M^t, \boldsymbol{x}^t, y^t)\}_{t=1}^T$, which is composed of prompts $\{(\boldsymbol{x}_1^t, y_1^t, \ldots, \boldsymbol{x}_M^t, y_M^t, \boldsymbol{x}^t)\}_{t=1}^T$ and responses $\{y^t\}_{t=1}^T$ for queries $\{\boldsymbol{x}^t\}_{t=1}^T$, where $y^t = f_*^t(\boldsymbol{x}^t) + \varsigma^t$ and $\varsigma^t \stackrel{\text{i.i.d.}}{\sim} \mathcal{D}_\varsigma$.

We say a model learns these functional relations *in-context*, when the model can predict the output $f_*(\boldsymbol{x})$ corresponding to query $\boldsymbol{x}$ by solely examining the context $(\boldsymbol{X}, \boldsymbol{y})$, without updating model parameters for each task. Given the pretrained model $f(\boldsymbol{X}, \boldsymbol{y}, \boldsymbol{x}; \boldsymbol{\theta})$ with parameter $\boldsymbol{\theta}$ which predicts the label of query $\boldsymbol{x}$ from context $(\boldsymbol{X}, \boldsymbol{y})$, we define the *expected ICL risk* as

$$\mathcal{R}_{N^*}(f) := \mathbb{E}[|f(\boldsymbol{X}_{1:N^*}, \boldsymbol{y}_{1:N^*}, \boldsymbol{x}; \boldsymbol{\theta}) - y|], \tag{2.1}$$

where $y = f_*(\boldsymbol{x}) + \varsigma$ and the expectation is taken over the in-context data: $\boldsymbol{x}_1, \ldots, \boldsymbol{x}_{N^*}, \boldsymbol{x} \sim \mathcal{D}_{\boldsymbol{x}}, f_* \sim \mathcal{D}_{f_*}, \varsigma_1, \ldots, \varsigma_{N^*}, \varsigma \sim \mathcal{D}_\varsigma$. Note that we take the expectation with respect to contexts $(\boldsymbol{X}_{1:N^*}, \boldsymbol{y}_{1:N^*}) \in \mathbb{R}^{d \times N^*} \times \mathbb{R}^{N^*}$ of length $N^*$, in order to examine the behavior of ICL at a specific context length.

### 2.1.2 Gaussian single-index models

We consider the situation where the true input-output relation is expressed by *single-index models*, i.e., functions that only depend on the direction of the index vector $\boldsymbol{\beta}$ in the input space. An efficient learning algorithm should adapt to this feature and identify the relevant subspace from high-dimensional observations; hence this problem setting has been extensively studied in the deep learning theory literature [BL19, BES+22, BBSS22, MHPG+23, MZD+23] to demonstrate the adaptivity of gradient-based feature learning.

**Assumption 1.** *Let $\tau \geq 0$ be the noise level. The prompt $(\boldsymbol{x}_1, y_1, \ldots, \boldsymbol{x}_N, y_N, \boldsymbol{x})$ is generated as*

$$\boldsymbol{x}_1, \boldsymbol{x}_2, \ldots, \boldsymbol{x}_N, \boldsymbol{x} \overset{\text{i.i.d.}}{\sim} \mathcal{D}_{\boldsymbol{x}} = \mathcal{N}(0, \boldsymbol{I}_d), \quad y_i = f_*(\boldsymbol{x}_i) + \varsigma_i, \quad f_*(\boldsymbol{x}_i) = \sigma_*(\langle \boldsymbol{x}_i, \boldsymbol{\beta} \rangle),$$

*where $\varsigma_1, \ldots, \varsigma_N \overset{\text{i.i.d.}}{\sim} \text{Unif}(\{-\tau, \tau\})$, and the distribution $\mathcal{D}_{f_*}$ of the true function $f_*$ is specified as:*

1. *__Index Features.__ Let $\mathcal{S}$ be an $r \leq d$-dimensional linear subspace of $\mathbb{R}^d$. We draw $\boldsymbol{\beta}$ uniformly from the unit sphere $\mathbb{S}(\mathcal{S})$ in $\mathcal{S}$, i.e., from $\mathbb{S}(\mathcal{S}) := \{\boldsymbol{\beta} \mid \boldsymbol{\beta} \in \mathcal{S}, \|\boldsymbol{\beta}\| = 1\}$.*

2. *__Link Function.__ $\sigma_*(z) = \sum_{i=Q}^P \frac{c_i}{i!} \text{He}_i(z)$, where $2 \leq Q \leq P$. We draw the Hermite coefficients $\{c_i\}_{i=Q}^P$ from any distribution satisfying*

$$\mathbb{E}[c_Q^2] = \Theta_{d,r}(1) \neq 0, \sum_{i=Q}^P c_i^2 \leq R_c^2 \ (a.s.) \ \text{ and } (c_Q, \ldots, c_P) \neq (0, \ldots, 0) \ (a.s.). \tag{2.2}$$

Throughout the paper, we assume that $P \ll d, r$ and $r \ll d$; specifically, we take $P = \Theta_{d,r}(1)$ and $r \lesssim d^{1/2}$. Note that the condition $r \ll d$ entails that the class of target functions is *low-dimensional*, and as we will see, such structure can be adapted by the transformer via pretraining.

**Remark 1.** *We make the following remarks on the assumption of single-index function class.*

- *For each task, the target is a single-index model with different index features drawn from some rank-$r$ subspace, and different link function with degree at most $P$ and information exponent (defined as the index of the lowest degree non-zero coefficient in the Hermite expansion of the link function, i.e, $\min\{i \mid c_i \neq 0\}$ in this case; see [BAGJ21, DH18]) at least $Q$. This heterogeneity reflects the situation where the difficulty of learning the input-output relation varies across tasks. Note that we allow for different distributions of the Hermite coefficients $\{c_i\}$: for example, we may set $(c_Q, \ldots, c_P) \sim \text{Unif}\left\{(c_Q, \ldots c_P) | \sum_{i=Q}^P \frac{c_i^2}{i!} = 1\right\}$ (manifold of coefficients satisfying $\mathbb{E}_{\boldsymbol{x}}[f_*(\boldsymbol{x})] = 1$), $\text{Unif}(\{0, 1\}^{P-Q+1} \setminus (0, \ldots, 0))$, or $\text{Unif}(\{(1, \ldots, 0), \ldots, (0, \ldots, 1)\})$.*

- *The condition $Q \geq 2$ (i.e., the Hermite expansion of $\sigma_*$ does not contain constant and linear terms) ensures that the gradient update detects the entire $r$-dimensional subspace instead of the trivial rank-1 component. For generic polynomial $\sigma_*$, this assumption can be satisfied by a simple preprocessing step that subtracts the low-degree components, as done in [DLS22].*

## 2.2 Student Model: Transformer with Nonlinear MLP Layer

We consider a transformer composed of a single-layer self-attention module preceded by an embedding module using a nonlinear multi-layer perceptron (MLP). Let $\boldsymbol{E} \in \mathbb{R}^{d_e \times d_N}$ be an embedding

matrix constructed from prompt $(\boldsymbol{x}_1, y_1, \ldots, \boldsymbol{x}_N, y_N, \boldsymbol{x})$. A single-layer SoftMax self-attention module [VSP$^+$17] is given as

$$f_{\text{Attn}}(\boldsymbol{E}; \boldsymbol{W}^P, \boldsymbol{W}^V, \boldsymbol{W}^K, \boldsymbol{W}^Q) = \boldsymbol{E} + \boldsymbol{W}^P \boldsymbol{W}^V \boldsymbol{E} \cdot \text{softmax}\left(\frac{(\boldsymbol{W}^K \boldsymbol{E})^\top \boldsymbol{W}^Q \boldsymbol{E}}{\rho}\right), \quad (2.3)$$

where $\rho$ is the temperature, and $\boldsymbol{W}^K, \boldsymbol{W}^Q \in \mathrm{R}^{d_k \times d_e}$, $\boldsymbol{W}^V \in \mathrm{R}^{d_v \times d_e}$ and $\boldsymbol{W}^P \in \mathrm{R}^{d_e \times d_v}$ are the key, query, value, and projection matrix, respectively. Following prior theoretical works [ZFB23, ACDS23, MHM23, KNS24], we remove the SoftMax and instead analyze the *linear* attention with $\rho = N$; it has been argued that such simplification can reproduce phenomena in practical transformer training [ACS$^+$23]. We further simplify the original self-attention module (2.3) by merging $\boldsymbol{W}^P \boldsymbol{W}^V$ as $\boldsymbol{W}^{PV} \in \mathbb{R}^{d_e \times d_e}$ and $(\boldsymbol{W}^K)^\top \boldsymbol{W}^Q$ as $\boldsymbol{W}^{KQ} \in \mathbb{R}^{d_e \times d_e}$, and consider the following parameterization also introduced in [ZFB23, ACDS23, WZC$^+$23],

$$\boldsymbol{W}^{PV} = \begin{bmatrix} * & * \\ \boldsymbol{0}_{1 \times (d_e - 1)} & v \end{bmatrix}, \boldsymbol{W}^{KQ} = \begin{bmatrix} \boldsymbol{K} & * \\ \boldsymbol{0}_{1 \times (d_e - 1)} & * \end{bmatrix}, \quad (2.4)$$

where $v \in \mathbb{R}$ and $\boldsymbol{K} \in \mathbb{R}^{(d_e - 1) \times (d_e - 1)}$. Then, the simplified attention module is written as $\tilde{f}_{\text{Attn}}(\boldsymbol{E}; \boldsymbol{W}^{PV}, \boldsymbol{W}^{KQ}) = \boldsymbol{E} + \boldsymbol{W}^{PV} \boldsymbol{E}\left(\frac{\boldsymbol{E}^\top \boldsymbol{W}^{KQ} \boldsymbol{E}}{N}\right)$, and we take the right-bottom entry as the prediction of $y$ corresponding to query $\boldsymbol{x}$.

Prior analyses of linear transformers [ZFB23, ACDS23, MHM23] defined the embedding matrix $\boldsymbol{E}$ simply as the input-output pairs; however, the combination of linear embedding and liner attention is not sufficient to learn nonlinear single-index models. Instead, we set $d_e = m + 1, d_N = N + 1$ for $m \in \mathbb{N}$ and construct $\boldsymbol{E}$ using an MLP layer:

$$\boldsymbol{E} = \begin{bmatrix} \sigma(\boldsymbol{w}_1^\top \boldsymbol{x}_1 + b_1) & \cdots & \sigma(\boldsymbol{w}_1^\top \boldsymbol{x}_N + b_1) & \sigma(\boldsymbol{w}_1^\top \boldsymbol{x} + b_1) \\ \vdots & \ddots & \vdots & \vdots \\ \sigma(\boldsymbol{w}_m^\top \boldsymbol{x}_1 + b_m) & \cdots & \sigma(\boldsymbol{w}_m^\top \boldsymbol{x}_N + b_m) & \sigma(\boldsymbol{w}_m^\top \boldsymbol{x} + b_m) \\ y_1 & \cdots & y_N & 0 \end{bmatrix} \in \mathbb{R}^{(m+1) \times (N+1)}, \quad (2.5)$$

where $\boldsymbol{w}_1, \ldots, \boldsymbol{w}_m \in \mathbb{R}^d$ and $b_1, \ldots, b_m \in \mathbb{R}$ are trainable parameters and $\sigma : \mathbb{R} \to \mathbb{R}$ is a nonlinear activation function; we use $\sigma(z) = \text{ReLU}(z) = \max\{z, 0\}$. This is to say, we define the embedding as the "hidden representation" $\sigma(\boldsymbol{w}^\top \boldsymbol{x} + b)$ of a width-$m$ two-layer neural network. For concise notation we write $\boldsymbol{W} = (\boldsymbol{w}_1, \ldots, \boldsymbol{w}_m)^\top, \boldsymbol{b} = (b_1, \ldots, b_m)^\top$.

**Remark 2.** *We make the following remarks on the considered architecture.*

- *The MLP embedding layer before attention has been adopted in recent works [GHM$^+$23, KS24, KNS24]. This setting can be interpreted as an idealized version of the mechanism that lower layers of the transformer construct useful representation, on top of which upper attention layers implement the in-context learning algorithm.*

- *Our architecture is also inspired by recent theoretical analyses of gradient-based feature learning, where it is shown that gradient descent on the MLP layer yields adaptivity to features of the target function and hence improved statistical efficiency [AAM22, DLS22, BES$^+$22].*

Combining the attention $\tilde{f}_{\text{Attn}}$ and MLP embedding (2.5), we can express the model prediction $y$ as

$$f(\boldsymbol{X}, \boldsymbol{y}, \boldsymbol{x}; \boldsymbol{W}, \boldsymbol{\Gamma}, \boldsymbol{b}) = \left\langle \frac{\boldsymbol{\Gamma} \sigma(\boldsymbol{W}\boldsymbol{X} + \boldsymbol{b}\boldsymbol{1}_N^\top)\boldsymbol{y}}{N}, \begin{bmatrix} \sigma(\boldsymbol{w}_1^\top \boldsymbol{x} + b_1) \\ \vdots \\ \sigma(\boldsymbol{w}_m^\top \boldsymbol{x} + b_m) \end{bmatrix} \right\rangle, \quad (2.6)$$

where $\boldsymbol{\Gamma} = v\boldsymbol{K}^\top$ and $\sigma(\boldsymbol{W}\boldsymbol{X} + \boldsymbol{b}\boldsymbol{1}_N^\top)$ denotes the $m \times N$ matrix whose $(i, j)$-th entry is $\sigma(\boldsymbol{w}_i^\top \boldsymbol{x}_j + b_i)$; see Appendix E for full derivation. We refer to $\boldsymbol{\Gamma}$ as the *attention matrix*.

## 2.3 Pretraining: Empirical Risk Minimization via Gradient Descent

We pretrain the transformer (2.6) by the gradient-based learning algorithm specified in Algorithm 1, which is inspired by the layer-wise training procedure studied in the neural network theory literature [AAM22, DLS22, BES$^+$22]. In particular, we update the trainable parameters in a sequential manner.

---

**Algorithm 1:** Gradient-based training of transformer with MLP layer

---

**Input** : Learning rate $\eta_1$, weight decay $\lambda_1, \lambda_2$, prompt length $N_1, N_2$, number of tasks $T_1, T_2$, attention matrix initialization scale $\gamma$.

**1 Initialize** $\boldsymbol{w}_j^{(0)} \sim \text{Unif}(\mathbb{S}^{d-1})$ $(j \in [m])$; $b_j^{(0)} \sim \text{Unif}([-1,1])$ $(j \in [m])$;
$\Gamma_{j,j}^{(0)} \sim \text{Unif}(\{\pm\gamma\})$ $(j \in [m])$ *and* $\Gamma_{i,j}^{(0)} = 0$ $(i \neq j \in [m])$.

**2 Stage I: Gradient descent for MLP layer**

**3**    Draw data $\{(\boldsymbol{x}_1^t, y_1^t, \ldots, \boldsymbol{x}_{N_1}^t, y_{N_1}^t, \boldsymbol{x}^t, y^t)\}_{t=1}^{T_1}$ with prompt length $N_1$.

**4**    $\boldsymbol{w}_j^{(1)} \leftarrow \boldsymbol{w}_j^{(0)} - \eta_1 \Big[ \nabla_{\boldsymbol{w}_j^{(0)}} \frac{1}{T_1} \sum_{t=1}^{T_1} (y^t - f(\boldsymbol{X}^t, \boldsymbol{y}^t, \boldsymbol{x}^t; \boldsymbol{W}^{(0)}, \boldsymbol{\Gamma}^{(0)}, \boldsymbol{b}^{(0)}))^2 + \lambda_1 \boldsymbol{w}_j^{(0)} \Big]$.

**5 Initialize** $b_j \sim \text{Unif}([-\log d, \log d])$.

**6 Stage II: Empirical risk minimization for attention layer**

**7**    Draw data $\{(\boldsymbol{x}_1^t, y_1^t, \ldots, \boldsymbol{x}_{N_2}^t, y_{N_2}^t, \boldsymbol{x}^t, y^t)\}_{t=T_1+1}^{T_1+T_2}$ with prompt length $N_2$.

**8**    $\boldsymbol{\Gamma}^* \leftarrow \arg\min_{\boldsymbol{\Gamma}} \frac{1}{T_2} \sum_{t=T_1+1}^{T_1+T_2} (y^t - f(\boldsymbol{X}^t, \boldsymbol{y}^t, \boldsymbol{x}^t; \boldsymbol{W}^{(1)}, \boldsymbol{\Gamma}, \boldsymbol{b}))^2 + \frac{\lambda_2}{2} \|\boldsymbol{\Gamma}\|_F^2$.

**Output** : Prediction function $\boldsymbol{x} \to f(\boldsymbol{X}, \boldsymbol{y}, \boldsymbol{x}; \boldsymbol{W}^{(1)}, \boldsymbol{\Gamma}^*, \boldsymbol{b})$.

---

- In Stage I we optimize the parameters of the MLP (embedding) layer, which is a non-convex problem due to the nonlinear activation function. To circumvent the nonlinear training dynamics, we follow the recipe in recent theoretical analyses of gradient-based feature learning [DLS22, BES⁺22, BEG⁺22]; specifically, we zoom into the *early phase* of optimization by taking one gradient step on on the regularized empirical risk. As we will see, the first gradient step already provides a reliable estimate of the target subspace $\mathcal{S}$, which enables the subsequent attention layer to implement a sample-efficient in-context learning algorithm.

- In Stage II we train the attention layer, which is a convex problem and the global minimizer can be efficiently found. We show that the optimized attention matrix $\boldsymbol{\Gamma}$ performs regression on the polynomial basis (defined by the MLP embedding), which can be seen as an in-context counterpart to the second-layer training (to learn the polynomial link) in [DLS22, AAM23, OSSW24].

## 3 Transformer Learns Single-index Models In-Context

### 3.1 Main Theorem

Our main theorem characterizes the pretraining and in-context sample complexity of the transformer with MLP layer (2.6) optimized by layer-wise gradient-based pretraining outlined in Algorithm 1.

**Theorem 1.** *Given Assumption 1 and $r \lesssim d^{\frac{1}{2}}$. We pretrain the transformer (2.6) using Algorithm 1 with $m = \tilde{\Omega}(r^P)$, $T_1 = \tilde{\Omega}(d^{Q+1}r^Q)$, $N_1 T_1 = \tilde{\Omega}(d^{2Q+1}r)$, $\gamma \asymp \frac{1}{m^{\frac{3}{2}} r^{\frac{1}{2}} d^Q}, \eta_1 \asymp m^{\frac{3}{2}} r d^{2Q-\frac{1}{2}} \cdot (\log d)^{-C_\eta}$ for constant $C_\eta$. Then, for appropriately chosen regularization parameters $\lambda_1, \lambda_2 > 0$, with probability at least 0.99 over the data distribution and random initialization, the ICL prediction risk (2.1) with test prompt length $N^*$ for the output $f$ of Algorithm 1 can be upper bounded as*

$$\mathcal{R}_{N^*}(f) - \tau = \tilde{O}\left( \sqrt{\frac{r^{3P}}{m} + r^{4P}\left( \frac{1}{T_2} + \frac{1}{N_2} + \frac{1}{N^*} \right)} \right).$$

Theorem 1 suggests that to achieve low in-context prediction risk, it is sufficient to set $T_1, N_1 = \tilde{\Omega}(d^{\Theta(Q)})$, and $m, T_2, N_2, N^* = \tilde{\Omega}(r^{\Theta(P)})$. Observe that the *pretraining* complexity $T_1, N_1$ scales with the ambient dimensionality $d$, but the *in-context* sample complexity $N^*$ only scales with the dimensionality of the function class $r \ll d$. This illustrates the in-context efficiency of pretrained transformers and aligns with our observations in the GPT-2 experiment reported in Figure 1.

**Remark 3.** *We make the following remarks.*

- *The sample complexity highlights different roles of the two sources of low dimensionality in our setting. The low dimensionality of single-index $f_*$ entails that the pretraining cost scales as $N \gtrsim d^{\Theta(Q)}$, which is consistent with prior analyses on gradient-based feature learning [BAGJ21, DNGL23]. On the other hand, the low dimensionality of function class (i.e., $\boldsymbol{\beta}$ lies in $r$-dimensional*

*subspace) leads to an in-context sample complexity that scales as $N^* \gtrsim r^{\Theta(P)}$, which, roughly speaking, is the rate achieved by polynomial regression or kernel models on $r$-dimensional data.*

- *The multiplicative scaling between $N_1$ and $T_1$ in the sample complexity suggests that one can tradeoff between the two quantities, that is, pretraining on more diverse tasks (larger $T_1$) can reduce the required pretraining context length $N_1$.*

- *Similar low-dimensional function class has been considered in the setting of transfer or multi-task learning with two-layer neural networks [DLS22, CHS$^+$23], where the first-layer weights identify the span of all target functions, and the second-layer approximates the nonlinearity. However, the crucial difference is that we do not update the network parameters based on the in-context examples; instead, the single-index learning is implemented by the forward pass of the transformer.*

**Comparison against baseline methods.** Below we summarize the statistical complexity of commonly-used estimators that only have access to $N^*$ in-context examples, but not the pretraining data. Note that the in-context sample complexity all depends on the ambient dimensionality $d \gg r$.

- **Kernel models.** Recall that our target function is a degree-$P$ polynomial, and hence kernel ridge regression requires $N^* \gtrsim d^P$ in-context examples [GMMM21, DWY21].
- **CSQ learners.** The correlational statistical query (CSQ) lower bound suggests that an algorithm making use of correlational information requires $N^* \gtrsim d^{\Theta(Q)}$ samples to learn a single-index model with information exponent $Q$ [DLS22, AAM22]. This sample complexity can be achieved by online SGD training of shallow neural network [BAGJ21, DNGL23].
- **Information theoretic limit.** Since the single-index model (1.1) contains $d$ unknown parameters, we can infer an information theoretic lower bound of $N^* \gtrsim d$ samples to estimate this function. This complexity can be achieved (up to polylog factors) by tailored SQ algorithms [CM20, DPVLB24] or modified gradient-base training of neural network [LOSW24, ADK$^+$24, JMS24].

### 3.2 Proof Sketch of Main Theorem

We provide a sketch of derivation for Theorem 1. The essential mechanism is outlined as follows: after training the MLP embedding via one gradient descent step, the MLP parameters $\{\boldsymbol{w}_j^{(1)}\}$ align with the common subspace of the target functions $\mathcal{S}$. Subsequently, the (linear) attention module estimates the input-output relation $f_*$ (which varies across tasks) on this $r$-dimensional subspace. We explain these two ingredients in the ensuing sections.

#### 3.2.1 Training the MLP Layer

We first show that the first gradient descent step on $\boldsymbol{W}$ results in significant alignment with the target subspace $\mathcal{S}$, using a proof strategy pioneered in [DLS22]. Note that under sufficiently small initialization and appropriately chosen weight decay, we have

$$\boldsymbol{w}_j^{(1)} \simeq \eta_1 \cdot \tfrac{2}{T_1} \sum_{t=1}^{T_1} y^t \nabla_{\boldsymbol{w}_j^{(0)}} f(\boldsymbol{X}^t, \boldsymbol{y}^t, \boldsymbol{x}^t; \boldsymbol{W}^{(0)}, \boldsymbol{\Gamma}^{(0)}, \boldsymbol{b}^{(0)}).$$

The key observation is that this correlation between $y$ and the gradient of model output contains information of $\mathcal{S}$. We establish the concentration of this empirical gradient around the expected (population) one, which determines the required rates of $T_1$ and $N_1$. For the population gradient, we make use of the Hermite expansion of $\sigma_{b_j}(z) = \sigma(z + b_j)$ and show that its leading term is proportional to $\begin{bmatrix} \boldsymbol{w}_{1:r}^{(0)} \\ \boldsymbol{0}_{d-r} \end{bmatrix}$, which is contained in the target subspace $\mathcal{S}$; see Appendix B for details.

#### 3.2.2 Attention Matrix with Good Approximation Property

Next we construct the attention matrix $\bar{\boldsymbol{\Gamma}}$ which satisfies the following approximation property:

**Proposition 2** (Informal). *There exists $\bar{\boldsymbol{\Gamma}}$ such that with high probability,*

$$\left| f(\boldsymbol{X}^t, \boldsymbol{y}^t, \boldsymbol{x}^t; \boldsymbol{W}^{(1)}, \bar{\boldsymbol{\Gamma}}, \boldsymbol{b}) - y^t \right| - \tau = \tilde{O}\left( \sqrt{r^{3P}/m + r^{2P}/N_2} \right)$$

*for all $t \in \{T_1 + 1, \ldots, T_2\}$. Moreover, we have $\|\bar{\boldsymbol{\Gamma}}\|_F = \tilde{O}(r^{2P}/m)$.*

We provide an intuitive explanation of this construction. Recall that the target function can be written in its Hermite expansion $f_*(\boldsymbol{x}) = \sum_{i=Q}^{P} \frac{c_i}{i!} \mathrm{He}_i(\langle \boldsymbol{x}, \boldsymbol{\beta} \rangle)$. We build an orthonormal basis for $f_*$ as follows. Let $\{\boldsymbol{\beta}_1, \ldots, \boldsymbol{\beta}_r\}$ be an orthonormal basis of $\mathcal{S}$. Then, any $f_*$ can be expressed as a linear combination of functions in $\mathcal{H} = \{\prod_{j=1}^{r} \mathrm{He}_{p_j}(\langle \boldsymbol{\beta}_j, \cdot \rangle) \mid Q \leq p_1 + \cdots + p_r \leq P, p_1 \geq 0, \ldots, p_r \geq 0\}$, where $\mathbb{E}_{\boldsymbol{x} \sim \mathcal{N}(0, I_d)}[h(\boldsymbol{x})h'(\boldsymbol{x})] = \mathbb{I}_{h=h'}$ holds for $h, h' \in \mathcal{H}$. We write $\mathcal{H} = \{h_1, \ldots, h_{B_P}\}$ with $B_P = |\mathcal{H}| = \Theta(r^P)$, and observe that a two-layer neural network can be constructed to approximate each $h_n$. Specifically, there exist vectors $\boldsymbol{a}^1, \ldots, \boldsymbol{a}^{B_P} \in \mathbb{R}^m$ such that $\sum_{j=1}^{m} a_j^n \sigma(\langle \boldsymbol{w}_j^{(1)}, \boldsymbol{x} \rangle + b_j) \simeq h_n(\boldsymbol{x})$ for each $n \in [B_P]$.

Consequently, we can build the desired attention matrix using coefficients $\boldsymbol{a}^1, \ldots, \boldsymbol{a}^{B_P}$. Let $\boldsymbol{A} = \begin{pmatrix} \boldsymbol{a}^1 & \cdots & \boldsymbol{a}^{B_P} \end{pmatrix} \in \mathbb{R}^{m \times B_P}$ and $\bar{\boldsymbol{\Gamma}} = \boldsymbol{A}\boldsymbol{A}^\top$, the attention module can recover the true function as

$$\left\langle \tfrac{1}{N_2} \bar{\boldsymbol{\Gamma}} \sigma(\boldsymbol{W}^{(1)} \boldsymbol{X}^t + \boldsymbol{b}\mathbf{1}_{N_2}^\top) \boldsymbol{y}^t, \sigma(\boldsymbol{W}^{(1)}\boldsymbol{x} + \boldsymbol{b}) \right\rangle$$

$$= \sum_{n=1}^{B_P} \left( \tfrac{1}{N_2} \sum_{i=1}^{N_2} \left( \sum_{j=1}^{m} a_j^n \sigma(\langle \boldsymbol{w}_j^{(1)}, \boldsymbol{x}_i \rangle + b_j) \right) y_i \right) \left( \sum_{j=1}^{m} a_j^n \sigma(\langle \boldsymbol{w}_j^{(1)}, \boldsymbol{x} \rangle + b_j) \right)$$

$$\stackrel{(a)}{\simeq} \sum_{n=1}^{B_P} \left( \tfrac{1}{N_2} \sum_{i=1}^{N_2} h_n(\boldsymbol{x}_i) y_i \right) h_n(\boldsymbol{x}) \stackrel{(b)}{\simeq} \sum_{n=1}^{B_P} \mathbb{E}[h_n(\boldsymbol{x}) f_*(\boldsymbol{x})] h_n(\boldsymbol{x}) = f_*(\boldsymbol{x}). \tag{3.1}$$

Roughly speaking, the self-attention architecture computes the correlation between the target function and basis element $h_i$ to estimate the corresponding coefficient in this basis decomposition. We evaluate (a) the approximation error of two-layer neural network in Appendix C.1, and (b) the discrepancy between the empirical and true correlations in Appendix C.2.

Note that the approximation errors and the norm of $\bar{\boldsymbol{\Gamma}}$ at this stage scale only with $r$ up to poly-logarithmic terms; this is because $\boldsymbol{W}^{(1)}$ already identifies the low-dimensional target subspace $\mathcal{S}$. Consequently, the in-context sample size $N^*$ only scales with the target dimensionality $r \ll d$.

### 3.2.3   Generalization Error Analysis

Finally, we transfer the learning guarantee from the constructed $\bar{\boldsymbol{\Gamma}}$ to the (regularized) empirical risk minimization solution $\boldsymbol{\Gamma}^*$. By the equivalence between optimization with $L_2$ regularization and norm-constrained optimization, there exists $\lambda_2$ such that $\|\boldsymbol{\Gamma}^*\|_F \leq \|\bar{\boldsymbol{\Gamma}}\|_F$ and the empirical ICL loss by $\boldsymbol{\Gamma}^*$ is no larger than that of $\bar{\boldsymbol{\Gamma}}$. Hence, we can bound the generalization error by $\boldsymbol{\Gamma}^*$ using a standard Rademacher complexity bound for norm-constrained transformers provided in Appendix D.1. One caveat here is that the context length $N^*$ at test time may differ from training time; hence we establish a context length-free generalization bound, which is discussed in Appendix D.2.

## 4   Synthetic Experiments

### 4.1   Experimental Setting

We pretrain a GPT-2 model [RWC+19] to learn the Gaussian single-index function class (1.1). Specifically, we consider the 12-layer architecture (with 22.3M parameters) used in [GTLV22] for in-context linear regression. The pretraining data is generated from random single-index models: for each task $t$, the context $\{(\boldsymbol{x}_i^t, y_i^t)\}_{i=1}^{N}$ is generated as $\boldsymbol{x}_i^t \stackrel{\text{i.i.d.}}{\sim} \mathcal{N}(0, \boldsymbol{I}_d)$ and $y_i^t = \sum_{i=Q}^{P} \frac{c_i^t}{i!} \mathrm{He}_i(\langle \boldsymbol{x}_i^t, \boldsymbol{\beta}^t \rangle)$, where $(c_Q^t, \ldots c_P^t) \stackrel{\text{i.i.d.}}{\sim} \mathrm{Unif}\left\{(c_Q, \ldots c_P) \mid \sum_{i=Q}^{P} \frac{c_i^2}{i!} = 1\right\}$ and $\boldsymbol{\beta}^t \stackrel{\text{i.i.d.}}{\sim} \mathrm{Unif}\left\{\boldsymbol{\beta} \mid \boldsymbol{\beta} = [\beta_1, \ldots, \beta_r, 0, \ldots, 0]^\top, \|\boldsymbol{\beta}\| = 1\right\}$. See Appendix F for further details.

### 4.2   Empirical Findings

**Ambient dimension-free sample complexity.**   In Figure 2 we examine how the in-context sample complexity of the GPT-2 model depends on the ambient dimensionality $d$ and the function class dimensionality $r$. For each problem setting the model is pretrained for $100,000$ steps using the data of degree $P = 4$ and information exponent $Q = 2$ (see Appendix F for details). In Figure 2(a) we observe that for fixed $r = 8$, varying the ambient dimensionality $d = 16, 32, 64$ leads to negligible change in the model performance for the in-context phase. In contrast, Figure 2(b) illustrates that for fixed $d$, the required sample size $N^*$ scales with the dimensionality of the function class $r = 2, 4, 8$. This confirms our theoretical finding that transformers can adapt to low-dimensional structure of the distribution of target functions via gradient-based pretraining.

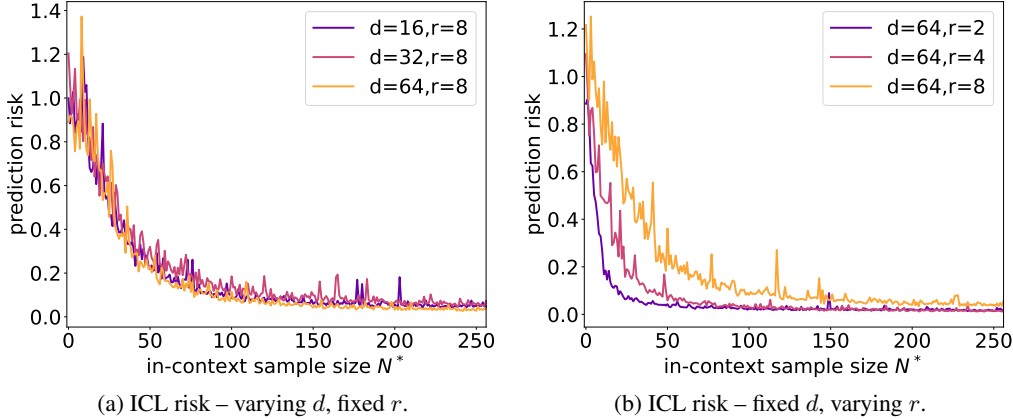

(a) ICL risk – varying $d$, fixed $r$.  (b) ICL risk – fixed $d$, varying $r$.

Figure 2: In-context sample complexity of GPT-2 model pretrained on Gaussian single-index function (see Section 4.1 for details) of degree-4 polynomial. Observe that $(a)$ the ICL risk curve overlaps for different ambient dimensions $d$ but the same target (subspace) dimensionality $r$, and $(b)$ the required sample size $N^*$ becomes larger as $r$ increases.

**Superiority over baseline algorithms.** In Figure 1 we compare the in-context sample complexity of the GPT-2 model pretrained by data of $Q = 3$ and $P = 2$ against two baseline algorithms that directly learn $f_*$ on the test prompt: $(i)$ kernel ridge regression with the Gaussian RBF kernel $k(\boldsymbol{x}, \boldsymbol{x}') = \exp\left(-\|\boldsymbol{x} - \boldsymbol{x}'\|^2/\sigma^2\right)$, and $(ii)$ two-layer neural network with ReLU activation $f_{\mathrm{NN}}(\boldsymbol{x}) = \frac{1}{m}\sum_{i=1}^{m} a_i \sigma(\langle \boldsymbol{x}, \boldsymbol{w}_i \rangle)$ trained by the Adam optimizer [KB15]. We observe that for $r = 8, d = 16, 32$, the pretrained transformer outperforms both KRR and two-layer NN; moreover, the performance of these two baseline algorithms deteriorates significantly as the ambient dimensionality $d$ becomes larger.

## 5 Conclusion and Future Direction

We study the complexity of in-context learning for the Gaussian single-index models using a pretrained transformer with nonlinear MLP layer. We provide an end-to-end analysis of gradient-based pretraining and establish a generalization error bound that takes into account the number of pretraining tasks, the number of pretraining and in-context examples, and the network width. Our analysis suggests that when the distribution of target functions exhibits low-dimensional structure, transformers can identify and adapt to such structure during pretraining, whereas any algorithm that only has access to the test prompt necessarily requires a larger sample complexity.

We outline a few limitations and possible future directions.

- The in-context sample complexity we derived $r^{\Theta(P)}$ corresponds to that of polynomial regression or kernel methods in $r$-dimensional space. This rate is natural as discussed in Section 3.2.2, where the linear self-attention module extracts the coefficients with respect to fixed basis functions (of size $r^{\Theta(P)}$). An interesting question is whether transformers can implement a more efficient in-context algorithm that matches the complexity of gradient-based feature learning in $r$ dimensions. This can be achieved if the pretrained model learns features in-context.

- Our pretraining complexity is based on one GD step analysis similar to [DLS22, BES+22] which makes use of the correlational information; hence the information exponent of the link functions plays an important role. We conjecture that the sample complexity can be improved if we modify the pretraining procedure or training objective, as done in [DTA+24, LOSW24, ADK+24, JMS24].

## Acknowledgement

KO was partially supported by JST, ACT-X Grant Number JPMJAX23C4. TS was partially supported by JSPS KAKENHI (24K02905) and JST CREST (JPMJCR2015). This research is unrelated to DW's work at xAI.

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

**Table of Contents**

# A Preliminaries

We consider the high-dimensional setting, i.e., our result holds for all $d \geq D$ where $D$ is a constant which does not depend on $d$ and $r$. Throughout the proofs, we take $\mathcal{S} = \{(x_1, \ldots, x_r, 0, \ldots, 0)^\top \mid x_1, \ldots, x_r \in \mathbb{R}\}$ and $\boldsymbol{\beta} \sim \mathrm{Unif}(\mathbb{S}(\mathcal{S})) = \mathrm{Unif}(\{(\beta_1, \ldots, \beta_r, 0, \ldots, 0)^\top \mid \beta_1^2 + \cdots + \beta_r^2 = 1\})$. Note that this assumption is without loss of generality by the rotational invariance of the Gaussian distribution.

## A.1 Definition of High Probability Event

**Definition 3.** *We say that an event $A$ occurs* with high probability *when there exists a sufficiently large constant $C^*$ which does not depend on $d, r$ and*

$$1 - Pr[A] \leq O(d^{-C^*})$$

*holds.*

We implicitly assume that we can redefine the constant $C^*$ to be sufficiently large as needed. The lemma below is a basic example and will be used in the proofs:

**Lemma 4.** *Let $x \sim \mathcal{N}(0, 1)$. Then, $|x| \lesssim \sqrt{\log d}$ holds with high probability.*

**Proof.** From the tail bound of Gaussian, $\mathbb{P}(|x| \geq t) \leq 2 \exp(-t^2/2)$ holds. Hence we get $\mathbb{P}(|x| \geq \sqrt{2C^* \log d}) \leq O(d^{-C^*})$. $\qquad\qquad\square$

Note that $C^*$ can be redefined by changing the hidden constant in $|x| \lesssim \sqrt{\log d}$.

If $A_1, \ldots, A_M$ occurs with high probability where $M = O(\mathrm{poly}(d))$, then $A_1 \cap \cdots \cap A_M$ also occurs with high probability (by redefining $C^*$). In particular, throughout this paper we assume that $m, N_1, N_2, T_1, T_2 = O(\mathrm{poly}(d))$, which allows us to take such unions.

## A.2 Tensor Notations

In this paper, a $k$-tensor is a multidimensional array which has $k$ indices: for example, matrices are 2-tensors. Let $\boldsymbol{A}$ be a $k$-tensor. $A_{i_1, \ldots, i_k}$ denotes $(i_1, \ldots, i_k)$-th entry of $\boldsymbol{A}$. Let $\boldsymbol{A}$ be a $k$-tensor and $\boldsymbol{B}$ be a $l$-tensor where $k \geq l$. $\boldsymbol{A}(\boldsymbol{B})$ denotes a $k - l$ tensor whose $(i_1, \ldots, i_{k-l})$-th entry is

$$\boldsymbol{A}(\boldsymbol{B})_{i_1, \ldots, i_{k-l}} = \sum_{j_1, \ldots, j_l} A_{i_1, \ldots, i_{k-l}, j_1, \ldots, j_l} B_{j_1, \ldots, j_l},$$

and is defined only when sizes are compatible. If $k = l$, we sometimes write $\boldsymbol{A}(\boldsymbol{B})$ as $\boldsymbol{A} \circ \boldsymbol{B}$ or $\langle \boldsymbol{A}, \boldsymbol{B} \rangle$. Let $\boldsymbol{v} \in \mathbb{R}^d$ be a vector and $k$ be a positive integer. Then, $\boldsymbol{v}^{\otimes k} \in \mathbb{R}^{d \times \cdots \times d}$ denotes a $k$-tensor whose $(i_1, \ldots, i_k)$-th entry is $v_{i_1} \cdots v_{i_k}$.

Let $f(\boldsymbol{x}) : \mathbb{R}^d \to \mathbb{R}$ be a $d$-variable differentiable function. A $k$-tensor $\nabla^k f(\boldsymbol{x})$ is defined as

$$\left( \nabla^k f(\boldsymbol{x}) \right)_{i_1, \ldots, i_k} = \frac{\partial}{\partial x_{i_1}} \cdots \frac{\partial}{\partial x_{i_k}} f(\boldsymbol{x}).$$

The following properties can be verified easily.

**Lemma 5.** *For tensors $\boldsymbol{A}$ and $\boldsymbol{B}$, $\|\boldsymbol{A}(\boldsymbol{B})\|_F^2 \leq \|\boldsymbol{A}\|_F^2 \|\boldsymbol{B}\|_F^2$ holds.*

**Lemma 6.** *For a vector $\boldsymbol{v}$, $\|\boldsymbol{v}^{\otimes k}\|_F^2 = (\|\boldsymbol{v}\|_2^2)^k$ holds.*

## A.3 Hermite Polynomials

We frequently use (probablists') Hermite polynomials, which is defined as

$$\mathrm{He}_i(z) = (-1)^i e^{\frac{z^2}{2}} \frac{\mathrm{d}^i}{\mathrm{d}z^i} e^{\frac{-z^2}{2}},$$

where $i$ is a nonnegative integer. We often make use of the orthogonality property: $\mathbb{E}_{z \sim \mathcal{N}(0,1)}[\mathrm{He}_i(z) \mathrm{He}_j(z)] = i! \delta_{i,j}$. The Hermite expansion for $\sigma : \mathbb{R} \to \mathbb{R}$ is defined as $\sigma(z) = \sum_{i \geq 0} \frac{a_i}{i!} \mathrm{He}_i(z)$ where $a_i = \mathbb{E}_{z \sim \mathcal{N}(0,1)}[\sigma(z) \mathrm{He}_i(z)]$. Similarly, the multivariate Hermite

expansion for $f : \mathbb{R}^d \to \mathbb{R}$ is defined as $f(\boldsymbol{z}) = \sum_{i_1 \geq 0, \ldots, i_d \geq 0} \frac{a_{i_1,\ldots,i_d}}{(i_1)! \cdots (i_d)!} \mathrm{He}_{i_1}(z_1) \cdots \mathrm{He}_{i_d}(z_d)$, where $a_{i_1,\ldots,i_d} = \mathbb{E}_{z_1,\ldots,z_d \sim \mathcal{N}(0,1)}[f(\boldsymbol{z})\mathrm{He}_{i_1}(z_1) \cdots \mathrm{He}_{i_d}(z_d)]$. The coefficient $a_{i_1,\ldots,i_d}$ can also be obtained by $a_{i_1,\ldots,i_d} = \mathbb{E}_{z_1,\ldots,z_d \sim \mathcal{N}(0,1)}\left[ \frac{\partial^{i_1}}{\partial z_1^{i_1}} \cdots \frac{\partial^{i_d}}{\partial z_d^{i_d}} f(\boldsymbol{z}) \right]$.

Consider $f_*$ in our problem setting (Assumption 1). We can bound $\mathbb{E}[f_*(\boldsymbol{x})^2]$ and $\mathbb{E}[f_*(\boldsymbol{x})^4]$ as follows.

**Lemma 7.** *Under Assumption 1, $\mathbb{E}_{\boldsymbol{x},f_*}[f_*(\boldsymbol{x})^2] = \Theta_{d,r}(1)$ and $\mathbb{E}_{\boldsymbol{x},f_*}[f_*(\boldsymbol{x})^4] = O_{d,r}(1)$ holds.*

**Proof.** From (2.2) and $P = \Theta_{d,r}(1)$,

$$
\mathbb{E}_{\boldsymbol{x}\sim\mathcal{N}(0,\boldsymbol{I}_d),f_*\sim\mathcal{D}_{f_*}}[f_*(\boldsymbol{x})^2] = \mathbb{E}_{\boldsymbol{z}\sim\mathcal{N}(0,1),\{c_i\}}\left[ \left( \sum_{i=Q}^{P} \frac{c_i}{i!}\mathrm{He}_i(z) \right)^2 \right]
$$

$$
= \mathbb{E}\left[ \sum_{i=Q}^{P} \frac{c_i^2}{i!} \right]
$$

$$
= \Theta_{d,r}(1).
$$

We can bound $\mathbb{E}_{\boldsymbol{x}}[f_*(\boldsymbol{x})^4] = \mathbb{E}_{z\sim\mathcal{N}(0,1),\{c_i\}}\left[ \left( \sum_{i=Q}^{P} \frac{c_i}{i!}\mathrm{He}_i(z) \right)^4 \right]$ naively as follows: let $\zeta_P^4 = \max_{Q \leq i \leq P} \mathbb{E}_{z\sim N(0,1)}[\mathrm{He}_i(z)^4]$. $\zeta_P$ is an $O(1)$ quantity depending only on $P$ and $Q$. Then,

$$
\mathbb{E}_{z\sim\mathcal{N}(0,1),\{c_i\}}\left[ \left( \sum_{i=Q}^{P} \frac{c_i}{i!}\mathrm{He}_i(z) \right)^4 \right]
$$

$$
\leq \sum_{Q \leq i,j,k,l \leq P} \frac{\mathbb{E}[|c_i c_j c_k c_l|]}{i!j!k!l!} \mathbb{E}[|\mathrm{He}_i(z)\mathrm{He}_j(z)\mathrm{He}_k(z)\mathrm{He}_l(z)|]
$$

$$
\leq \sum_{Q \leq i,j,k,l \leq P} R_c^4 \mathbb{E}_z[\mathrm{He}_i(z)^4]^{1/4} \mathbb{E}_z[\mathrm{He}_j(z)^4]^{1/4} \mathbb{E}_z[\mathrm{He}_k(z)^4]^{1/4} \mathbb{E}_z[\mathrm{He}_l(z)^4]^{1/4}
$$

$$
\leq P^4 R_c^4 \zeta_P^4.
$$

$\square$

The lemma below is useful to find a basis of the set of single-index functions.

**Lemma 8.** *Suppose $\boldsymbol{\beta} \in \mathbb{S}(\mathcal{S})$. Then,*

$$
\mathrm{He}_p(\langle \boldsymbol{x}, \boldsymbol{\beta} \rangle) = \sum_{\substack{p_1 \geq 0, \ldots, p_r \geq 0}}^{p_1 + \cdots + p_r = p} \frac{p!}{p_1! \cdots p_r!} \cdot \beta_1^{p_1} \cdots \beta_r^{p_r} \cdot \mathrm{He}_{p_1}(x_1) \cdots \mathrm{He}_{p_r}(x_r)
$$

*holds.*

**Proof.** Note that $\mathbb{E}_{x_1,\ldots,x_r \sim \mathcal{N}(0,1)}\left[ \frac{\partial^{i_1}}{\partial x_1^{i_1}} \cdots \frac{\partial^{i_r}}{\partial x_r^{i_r}} \mathrm{He}_p(\langle \boldsymbol{x}, \boldsymbol{\beta} \rangle) \right]$ is nonzero only when $i_1 + \cdots + i_r = p$, because $\frac{\partial^{i_1}}{\partial x_1^{i_1}} \cdots \frac{\partial^{i_r}}{\partial x_r^{i_r}} \mathrm{He}_p(\langle \boldsymbol{x}, \boldsymbol{\beta} \rangle) = p(p-1) \cdots (p - (i_1 + \cdots + i_r) + 1)\beta_1^{i_1} \cdots \beta_r^{i_r} \mathrm{He}_{p-(i_1+\cdots+i_r)}(\langle \boldsymbol{x}, \boldsymbol{\beta} \rangle)$ and $\mathbb{E}_{x_1,\ldots,x_r \sim \mathcal{N}(0,1)}[\mathrm{He}_q(\langle \boldsymbol{x}, \boldsymbol{\beta} \rangle)] = \mathbb{E}_{z\sim\mathcal{N}(0,1)}[\mathrm{He}_q(z)] = 0$ if $q > 0$. When $i_1 + \cdots + i_r = p$, then $\mathbb{E}_{x_1,\ldots,x_r \sim \mathcal{N}(0,1)}\left[ \frac{\partial^{i_1}}{\partial x_1^{i_1}} \cdots \frac{\partial^{i_r}}{\partial x_r^{i_r}} \mathrm{He}_p(\langle \boldsymbol{x}, \boldsymbol{\beta} \rangle) \right] = p!\beta_1^{i_1} \cdots \beta_r^{i_r}$ holds. Thus, from the multivariate Hermite expansion we obtain the claim. $\square$

**Corollary 9.** *Let $f_*(\boldsymbol{x}) = \sum_{i=Q}^{P} \frac{c_i}{i!}\mathrm{He}_i(\langle \boldsymbol{x}, \boldsymbol{\beta} \rangle)$. Then, $\mathbb{E}_{\boldsymbol{x}}[\nabla^k f_*(\boldsymbol{x})] = c_k \boldsymbol{\beta}^{\otimes k}$ if $Q \leq k \leq P$ and otherwise it is the zero tensor.*

## A.4 Other Auxiliary Lemmas

**Lemmas on random vectors.** Let $\chi(d)$ be the distribution of $\sqrt{z_1^2 + \cdots + z_d^2}$ where $z_i \overset{\text{i.i.d.}}{\sim} \mathcal{N}(0,1)$. Note that the Gaussian vector $\boldsymbol{v} \sim \mathcal{N}(0, I_d)$ can be decomposed as $\boldsymbol{v} = z\boldsymbol{w}$ where $z \sim \chi(d)$ and $\boldsymbol{w} \sim \text{Unif}(\mathbb{S}^{d-1})$, because the norm and the direction of the Gaussian vector are independent of each other.

**Lemma 10** (Example 2.11 in [Wai19])**.** *For $z \sim \chi(d)$ and $0 < t < 1$,*

$$\mathbb{P}[|z^2 - d| \geq dt] \leq 2\exp(-dt^2/8)$$

*holds.*

**Lemma 11.** *Let $C > 0$. For $\boldsymbol{w} \sim \text{Unif}(\mathbb{S}^{d-1})$, $\|\boldsymbol{w}_{1:r}\|^2 \leq \frac{4Cr}{d}\log d$ holds with probability at least $1 - 2\exp(-d/32) - 2rd^{-C}$ (then with high probability).*

**Proof.** Let $z^2 \sim \chi^2(d)$ be independent from $\boldsymbol{w}$. We observe that the distribution of $z^2\|\boldsymbol{w}_{1:r}\|^2$ is the $\chi^2(r)$-distribution. First, from Lemma 10, with probability at least $1 - 2\exp(-d/32)$, $z^2 \geq d/2$ holds. Moreover, we can say that $(z')^2 \sim \chi^2(r)$ satisfies $(z')^2 \leq 2Cr\log d$ with high probability for sufficiently large constant: let us decompose $(z')^2 = v_1^2 + \cdots + v_r^2$ where $v_i \sim \mathcal{N}(0,1)$; from Lemma 4, $v_i^2 \leq 2C\log d$ holds with probability at least $1 - 2d^{-C}$, and thus $(z')^2 \leq 2Cr\log d$ holds with probability at least $1 - 2rd^{-C}$. Taking the union bound yields the result. $\qquad\square$

**Lemma 12.** *For $k \geq 1$ and $r \leq d$,*

$$\mathbb{E}_{\boldsymbol{w} \sim \mathbb{S}^{d-1}}[\|\boldsymbol{w}_{1:r}\|^{2k}] = O_{d,r}\left(\left(\frac{r}{d}\right)^k\right)$$

*holds.*

**Proof.** Since $\|\boldsymbol{w}_{1:r}\|^2$ is a polynomial of $\boldsymbol{w}$, Lemma 24 in [DLS22] yields $\mathbb{E}_{\boldsymbol{w} \sim \mathbb{S}^{d-1}}[\|\boldsymbol{w}_{1:r}\|^{2k}] \lesssim \mathbb{E}_{\boldsymbol{w} \sim \mathbb{S}^{d-1}}[\|\boldsymbol{w}_{1:r}\|^4]^{k/2}$. Here, $\mathbb{E}_{\boldsymbol{w} \sim \text{Unif}(\mathbb{S}^{d-1})}[\|\boldsymbol{w}_{1:r}\|^4] = \frac{\mathbb{E}_{v_i \sim \mathcal{N}(0,1)}[(v_1^2 + \cdots + v_r^2)^2]}{\mathbb{E}_{z \sim \chi^4(d)}[z]} = \frac{3r + r(r-1)}{d(d+2)} = \Theta(r^2/d^2)$ yields the result. $\qquad\square$

**Lemma 13** (Sub Gaussian vector)**.** *Let $\boldsymbol{x}$ be a $d$-dimensional random vector and suppose that $\langle \boldsymbol{u}, \boldsymbol{x} \rangle$ is zero-mean $\sigma^2$-sub Gaussian for any $\boldsymbol{u} \in \mathbb{R}^d$ such that $\|\boldsymbol{u}\| = 1$. Then, $\|\boldsymbol{x}\| = \tilde{O}(\sigma\sqrt{d})$ holds with high probabillity.*

**Lemma 14.** *Let $X$ be a $\sigma^2$- sub Gaussian random variable and $Y$ be a bounded random variable satisfying $|Y| \leq R$ almost surely. Then, $XY - \mathbb{E}[XY]$ is $O(\sigma^2 R^2)$-sub Gaussian.*

**Proof.** $XY$ is $\sigma^2 R^2$-Gaussian by examining the tail probability. Hence, $XY - \mathbb{E}[XY]$ is $O(\sigma^2 R^2)$-sub Gaussian from Lemma 2.6.8 in [Ver18]. $\qquad\square$

**Polynomial concentration.** We frequently use the following bounds on an output data $y$ (which is a polynomial of $\boldsymbol{x}$).

**Lemma 15** (Theorem 1.5 in [GSS21])**.** *Let $\boldsymbol{x} \sim \mathcal{N}(0, \boldsymbol{I}_n)$ and $f$ be a degree-$D$ $n$-variable polynomial function. Then, for all $t \geq 0$ it holds that*

$$\mathbb{P}(|f(\boldsymbol{x}) - \mathbb{E}_{\boldsymbol{x}}[f(\boldsymbol{x})]| \geq t) \leq 2\exp\left(-\frac{1}{C_D M^2}\min_{1 \leq k \leq D}\left(\frac{t}{\|\mathbb{E}_{\boldsymbol{x}}[\nabla^k f(\boldsymbol{x})]\|_F}\right)^{2/k}\right),$$

*where $M$ is an absolute constant and $C_D$ is a constant depending only on $D$ (if $\|\mathbb{E}_{\boldsymbol{x}}[\nabla^k f(\boldsymbol{x})]\|_F = 0$ then we ignore the term with $k$).*

**Lemma 16.** *Let $\boldsymbol{x} \sim \mathcal{N}(0, I_d)$ and $y = f_*(\boldsymbol{x}) + \varsigma = \sum_{i=Q}^P \frac{c_i}{i!}\text{He}_i(\langle \boldsymbol{x}, \boldsymbol{\beta} \rangle) + \varsigma$ where $c_i$ and $\boldsymbol{\beta}$ are drawn according to Assumption 1, and $\varsigma \sim \text{Unif}\{\pm\tau\}$. Then, for any fixed $\{c_i\}$ and $\boldsymbol{\beta}$ on the support of their distributions, it holds that*

$$\mathbb{P}(|y| \geq t + \tau) \leq 2\exp\left(-\frac{1}{C_P M^2}\min_{2 \leq k \leq P}\left(\frac{t}{R_c}\right)^{2/k}\right),$$

*where $M$ is an absolute constant and $C_P$ is a constant depending only on $P$.*

**Proof.** Note that from Corollary 9, $\mathbb{E}_{\boldsymbol{x}}[f_*(\boldsymbol{x})] = 0, \mathbb{E}_{\boldsymbol{x}}[\nabla f(\boldsymbol{x})] = \mathbf{0}$ and $\mathbb{E}_{\boldsymbol{x}}[\nabla^i f_*(\boldsymbol{x})] = c_i(\boldsymbol{\beta})^{\otimes i}$; by (2.2), Lemma 6 and $\|\boldsymbol{\beta}\| = 1$, $\|\mathbb{E}_{\boldsymbol{x}}[\nabla^i f_*(\boldsymbol{x})]\|_F \leq R_c$ holds. Lemma 15 yields the result. $\qquad\square$

**Corollary 17.** *Fix any $\{c_i\}$ and $\boldsymbol{\beta}$ on the support of their distributions. Then, $|f_*(\boldsymbol{x})|, |y| \lesssim (\log d)^{P/2}$ holds with high probability over the distribution of $\boldsymbol{x}$ and $\varsigma$.*

## B   Proofs for the MLP Layer

This section analyzes the behavior of MLP weight $\boldsymbol{w}$ after one gradient descent step (line 4 in Algorithm 1), as sketched in Section 3.2.1.

We set $\lambda_1 = \eta_1^{-1}$. From line 4 of Algorithm 1, for each $j \in [m]$,

$$\boldsymbol{w}_j^{(1)} = 2\eta_1 \frac{1}{T_1} \sum_{t=1}^{T_1} y^t \nabla_{\boldsymbol{w}_j^{(0)}} f(\boldsymbol{X}^t, \boldsymbol{y}^t, \boldsymbol{x}^t; \boldsymbol{W}^{(0)}, \boldsymbol{\Gamma}^{(0)}, \boldsymbol{b}^{(0)})$$

$$- 2\eta_1 \frac{1}{T_1} \sum_{t=1}^{T_1} f(\boldsymbol{X}^t, \boldsymbol{y}^t, \boldsymbol{x}^t; \boldsymbol{W}^{(0)}, \boldsymbol{\Gamma}^{(0)}, \boldsymbol{b}^{(0)}) \nabla_{\boldsymbol{w}_j^{(0)}} f(\boldsymbol{X}^t, \boldsymbol{y}^t, \boldsymbol{x}^t; \boldsymbol{W}^{(0)}, \boldsymbol{\Gamma}^{(0)}, \boldsymbol{b}^{(0)})$$

$$= 2\eta_1 \Gamma_{j,j}^{(0)} \left( \sum_{t=1}^{T_1} \frac{1}{T_1} y^t \sigma_{b_j}(\boldsymbol{w}_j^{(0)\top} \boldsymbol{x}^t) \frac{1}{N_1} \sum_{i=1}^{N_1} y_i^t \sigma_{b_j}'(\boldsymbol{w}_j^{(0)\top} \boldsymbol{x}_i^t) \boldsymbol{x}_i^t \right.$$

$$\left. + \sum_{t=1}^{T_1} \frac{1}{T_1} y^t \sigma_{b_j}'(\boldsymbol{w}_j^{(0)\top} \boldsymbol{x}^t) \boldsymbol{x}^t \frac{1}{N_1} \sum_{i=1}^{N_1} y_i^t \sigma_{b_j}(\boldsymbol{w}_j^{(0)\top} \boldsymbol{x}_i^t) \right) \tag{B.1}$$

$$- 2\eta_1 \frac{1}{T_1} \sum_{t=1}^{T_1} f(\boldsymbol{X}^t, \boldsymbol{y}^t, \boldsymbol{x}^t; \boldsymbol{W}^{(0)}, \boldsymbol{\Gamma}^{(0)}, \boldsymbol{b}^{(0)}) \nabla_{\boldsymbol{w}_j^{(0)}} f(\boldsymbol{X}^t, \boldsymbol{y}^t, \boldsymbol{x}^t; \boldsymbol{W}^{(0)}, \boldsymbol{\Gamma}^{(0)}, \boldsymbol{b}^{(0)}).$$

holds where $\sigma_{b_j}(z) := \sigma(z + b_j)$. Here, $\boldsymbol{x}_i^t$ and $y_i^t$ is the input and output of $i$-th context at $t$-th task, respectively. First we show that the final term is sufficiently small, if we set $\gamma = |\Gamma_{j,j}^{(0)}|$ sufficiently small.

**Lemma 18.** *With high probability over the distribution of $\{(\boldsymbol{X}^t, \boldsymbol{y}^t, \boldsymbol{x}^t, y^t)\}_{t=1}^{T_1}$ and $\{\boldsymbol{w}_j^{(0)}\}_{j=1}^m$,*

$$\left\| 2\eta_1 \frac{1}{T_1} \sum_{t=1}^{T_1} f(\boldsymbol{X}^t, \boldsymbol{y}^t, \boldsymbol{x}^t; \boldsymbol{W}^{(0)}, \boldsymbol{\Gamma}^{(0)}, \boldsymbol{b}^{(0)}) \nabla_{\boldsymbol{w}_j^{(0)}} f(\boldsymbol{X}^t, \boldsymbol{y}^t, \boldsymbol{x}^t; \boldsymbol{W}^{(0)}, \boldsymbol{\Gamma}^{(0)}, \boldsymbol{b}^{(0)}) \right\|$$
$$= \tilde{O}\left( \eta_1 \gamma^2 m \sqrt{d} \right).$$

**Proof.** Recall that

$$f(\boldsymbol{X}^t, \boldsymbol{y}^t, \boldsymbol{x}^t; \boldsymbol{W}^{(0)}, \boldsymbol{\Gamma}^{(0)}, \boldsymbol{b}^{(0)}) = \sum_{j=1}^m \Gamma_{j,j}^{(0)} \left( \frac{1}{N_1} \sum_{i=1}^{N_1} y_i^t \sigma_{b_j}(\boldsymbol{w}_j^{(0)\top} \boldsymbol{x}_i^t) \right) \sigma_{b_j}(\boldsymbol{w}_j^{(0)\top} \boldsymbol{x}^t),$$

$$\nabla_{\boldsymbol{w}_j^{(0)}} f(\boldsymbol{X}^t, \boldsymbol{y}^t, \boldsymbol{x}^t; \boldsymbol{W}^{(0)}, \boldsymbol{\Gamma}^{(0)}, \boldsymbol{b}^{(0)}) = \Gamma_{j,j}^{(0)} \left( \frac{1}{N_1} \sum_{i=1}^{N_1} y_i^t \sigma_{b_j}'(\boldsymbol{w}_j^{(0)\top} \boldsymbol{x}_i^t) \boldsymbol{x}_i^t \right) \sigma_{b_j}(\boldsymbol{w}_j^{(0)\top} \boldsymbol{x}^t)$$

$$+ \Gamma_{j,j}^{(0)} \left( \frac{1}{N_1} \sum_{i=1}^{N_1} y_i^t \sigma_{b_j}(\boldsymbol{w}_j^{(0)\top} \boldsymbol{x}_i^t) \right) \sigma_{b_j}'(\boldsymbol{w}_j^{(0)\top} \boldsymbol{x}^t) \boldsymbol{x}^t.$$

Fix $i, t, j$ and consider the inner product between $\boldsymbol{w}_j^{(0)} \sim \mathrm{Unif}(\mathbb{S}^{d-1})$ and $\boldsymbol{x}_i^t \sim \mathcal{N}(0, I_d)$. From the rotational invariance, without loss of generality, we can assume that $\boldsymbol{w}_j^{(0)} = [1, 0, \ldots, 0]^\top$ and $\boldsymbol{x}_i^t \sim \mathcal{N}(0, I_d)$. Therefore, $\left\langle \boldsymbol{w}_j^{(0)}, \boldsymbol{x}_i^t \right\rangle \sim \mathcal{N}(0, 1)$. From Lemma 4, $\left\langle \boldsymbol{w}_j^{(0)}, \boldsymbol{x}_i^t \right\rangle^2 \lesssim \log d$ holds with high probability. Moreover, from Lemma 10 and Corollary 17, we know that $|y_i^t|, |y^t| \lesssim (\log d)^{P/2}$

and $\|\boldsymbol{x}_i^t\|, \|\boldsymbol{x}^t\| \lesssim \sqrt{d}$ holds. We can take the union bound with respect to all the $i, t, j$, retaining the high probability bound (see Appendix A). Therefore, noting that $|b_j| \leq 1$, we obtain

$$|f(\boldsymbol{X}^t, \boldsymbol{y}^t, \boldsymbol{x}^t; \boldsymbol{W}^{(0)}, \boldsymbol{\Gamma}^{(0)}, \boldsymbol{b}^{(0)})| \lesssim m\gamma(\log d)^{P/2+1}$$

and

$$\|\nabla_{\boldsymbol{w}_j^{(0)}} f(\boldsymbol{X}^t, \boldsymbol{y}^t, \boldsymbol{x}^t; \boldsymbol{W}^{(0)}, \boldsymbol{\Gamma}^{(0)}, \boldsymbol{b}^{(0)})\| \lesssim \gamma(\log d)^{P/2+1/2}\sqrt{d}.$$

Thus we obtain the assertion. $\qquad\square$

From now on we focus on analyzing (B.1), which expresses the correlation between the output label $y$ and the gradient of model output $\nabla_{\boldsymbol{w}} f$. Let

$$\boldsymbol{g}_{T_1, N_1}(\boldsymbol{w}, b, \{(\boldsymbol{X}^t, \boldsymbol{y}^t, \boldsymbol{x}^t, y^t)\}_{t=1}^{T_1})$$

$$:= \sum_{i=1}^{T_1} \frac{1}{T_1} y^t \sigma_b(\boldsymbol{w}^\top \boldsymbol{x}^t) \frac{1}{N_1} \sum_{i=1}^{N_1} y_i^t \sigma_b'(\boldsymbol{w}^\top \boldsymbol{x}_i^t) \boldsymbol{x}_i^t$$

$$+ \sum_{i=1}^{T_1} \frac{1}{T_1} y^t \sigma_b'(\boldsymbol{w}^\top \boldsymbol{x}^t) \boldsymbol{x}^t \frac{1}{N_1} \sum_{i=1}^{N_1} y_i^t \sigma_b(\boldsymbol{w}^\top \boldsymbol{x}_i^t), \tag{B.2}$$

and $\boldsymbol{g}(\boldsymbol{w}, b) = \mathbb{E}[\boldsymbol{g}_{T_1, N_1}(\boldsymbol{w}, b, \{(\boldsymbol{X}^t, \boldsymbol{y}^t, \boldsymbol{x}^t, y^t)\}_{t=1}^{T_1})]$, where the expectation is taken with respect to the data $\{(\boldsymbol{X}^t, \boldsymbol{y}^t, \boldsymbol{x}^t, y^t)\}_{t=1}^{T_1}$. Note that

$$\boldsymbol{w}_j^{(1)} = 2\eta_1 \Gamma_{j,j}^{(0)} \boldsymbol{g}_{T_1, N_1}(\boldsymbol{w}_j^{(0)}, b_j^{(0)}, \{(\boldsymbol{X}^t, \boldsymbol{y}^t, \boldsymbol{x}^t, y^t)\}_{t=1}^{T_1}) + \tilde{O}\left(\eta_1 \gamma^2 m\sqrt{d}\right)$$

from Lemma 18.

In Appendix B.1, we analyze the explicit form of $\boldsymbol{g}(\boldsymbol{w}, b)$ via the Hermite expansion: we can check that its main term aligns with the low-dimensional subspace $\mathcal{S}$. In Appendix B.2, we show the concentration of $\boldsymbol{g}_{T_1, N_1}$ around $\boldsymbol{g}$.

## B.1 Calculation of Population Gradient

### B.1.1 Hermite Expansion

First, note that

$$\boldsymbol{g}(\boldsymbol{w}, b) = \mathbb{E}\big[y\sigma_b(\boldsymbol{w}^\top \boldsymbol{x})y_i\sigma_b'(\boldsymbol{w}^\top \boldsymbol{x}_i)\boldsymbol{x}_i\big] + \mathbb{E}\big[y\sigma_b'(\boldsymbol{w}^\top \boldsymbol{x})\boldsymbol{x}y_i\sigma_b(\boldsymbol{w}^\top \boldsymbol{x}_i)\big]$$

$$= \mathbb{E}_{f_*}\big[\mathbb{E}_{\{\boldsymbol{x}_i\}, \boldsymbol{x}, \{\varsigma_i\}, \varsigma}\big[(f_*(\boldsymbol{x}) + \varsigma)\sigma_b(\boldsymbol{w}^\top \boldsymbol{x})(f_*(\boldsymbol{x}_i) + \varsigma_i)\sigma_b'(\boldsymbol{w}^\top \boldsymbol{x}_i)\boldsymbol{x}_i\big]\big]$$

$$+ \mathbb{E}_{f_*}\big[\mathbb{E}_{\{\boldsymbol{x}_i\}, \boldsymbol{x}, \{\varsigma_i\}, \varsigma}\big[(f_*(\boldsymbol{x}) + \varsigma)\sigma_b'(\boldsymbol{w}^\top \boldsymbol{x})\boldsymbol{x}(f_*(\boldsymbol{x}_i) + \varsigma_i)\sigma_b(\boldsymbol{w}^\top \boldsymbol{x}_i)\big]\big]$$

$$= \mathbb{E}_{f_*}\big[\mathbb{E}_{\boldsymbol{x}, \varsigma}\big[(f_*(\boldsymbol{x}) + \varsigma)\sigma_b(\boldsymbol{w}^\top \boldsymbol{x})\big]\mathbb{E}_{\{\boldsymbol{x}_i\}, \{\varsigma_i\}}\big[(f_*(\boldsymbol{x}_i) + \varsigma_i)\sigma_b'(\boldsymbol{w}^\top \boldsymbol{x}_i)\boldsymbol{x}_i\big]\big]$$

$$+ \mathbb{E}_{f_*}\big[\mathbb{E}_{\boldsymbol{x}, \varsigma}\big[(f_*(\boldsymbol{x}) + \varsigma)\sigma_b'(\boldsymbol{w}^\top \boldsymbol{x})\boldsymbol{x}\big]\mathbb{E}_{\{\boldsymbol{x}_i\}, \{\varsigma_i\}}\big[(f_*(\boldsymbol{x}_i) + \varsigma_i)\sigma_b(\boldsymbol{w}^\top \boldsymbol{x}_i)\big]\big]$$

$$= 2\mathbb{E}_{f_*}[\mathbb{E}_{\boldsymbol{x}}[f_*(\boldsymbol{x})\sigma_b'(\boldsymbol{w}^\top \boldsymbol{x})\boldsymbol{x}]\mathbb{E}_{\boldsymbol{x}}[f_*(\boldsymbol{x})\sigma_b(\boldsymbol{w}^\top \boldsymbol{x})]].$$

Here $\mathbb{E}_{f_*}$ denotes the expectation with respect to the distribution of the true function $f_*$ (i.e., the distribution of $\{c_i\}_{i=Q}^P$ and $\boldsymbol{\beta}$) specified in Assumption 1. Now let $\sigma_b(z) = \sum_{i\geq 0} \frac{a_i(b)}{i!} \text{He}_i(z)$ be the Hermite expansion of student activation $\sigma_b(z) = \text{ReLU}(z + b)$.

**Lemma 19.** *It holds that*

$$\mathbb{E}_{\boldsymbol{x}}[f_*(\boldsymbol{x})\sigma_b'(\boldsymbol{w}^\top \boldsymbol{x})\boldsymbol{x}]$$

$$= \sum_{i=Q-1}^{P-1} \frac{a_{i+1}(b)\mathbb{E}_{\boldsymbol{x}}[\nabla^{i+1} f_*(\boldsymbol{x})](\boldsymbol{w}^{\otimes i})}{i!} + \boldsymbol{w} \sum_{i=Q}^P \frac{a_{i+2}(b)\mathbb{E}_{\boldsymbol{x}}[\nabla^i f_*(\boldsymbol{x})](\boldsymbol{w}^{\otimes i})}{i!}, \tag{B.3}$$

$$\mathbb{E}_{\boldsymbol{x}}[f_*(\boldsymbol{x})\sigma_b(\boldsymbol{w}^\top \boldsymbol{x})] = \sum_{i=Q}^P \frac{a_i(b)\mathbb{E}_{\boldsymbol{x}}[\nabla^i f_*(\boldsymbol{x})](\boldsymbol{w}^{\otimes i})}{i!}.$$

**Proof.** Consider a function $g(\boldsymbol{w}^\top \boldsymbol{x}) = \sum_{i \geq 0} \frac{\tilde{a}_i}{i!} \mathrm{He}_i(\boldsymbol{w}^\top \boldsymbol{x})$. Here $\mathbb{E}_{\boldsymbol{x}}[f_*(\boldsymbol{x})g(\boldsymbol{w}^\top \boldsymbol{x})]$ can be calculated as

$$\mathbb{E}_{\boldsymbol{x}}[f_*(\boldsymbol{x})g(\boldsymbol{w}^\top \boldsymbol{x})] = \sum_{i \geq 0} \frac{\tilde{a}_i}{i!} \mathbb{E}_{\boldsymbol{x}}[f_*(\boldsymbol{x})\mathrm{He}_i(\boldsymbol{w}^\top \boldsymbol{x})]$$

$$= \sum_{i \geq 0} \sum_{\substack{i_1 \geq 0, \ldots, i_d \geq 0}}^{i_1 + \cdots + i_d = i} \tilde{a}_i \frac{w_1^{i_1} \cdots w_d^{i_d}}{i_1! \cdots i_d!} \mathbb{E}_{\boldsymbol{x}}[f_*(\boldsymbol{x})\mathrm{He}_{i_1}(x_1) \cdots \mathrm{He}_{i_d}(x_d)] \ (\because \text{Lemma } 8)$$

$$= \sum_{i \geq 0} \sum_{\substack{i_1 \geq 0, \ldots, i_d \geq 0}}^{i_1 + \cdots + i_d = i} \tilde{a}_i \frac{w_1^{i_1} \cdots w_d^{i_d}}{i_1! \cdots i_d!} \mathbb{E}_{\boldsymbol{x}}\left[ \frac{\partial^{i_1}}{\partial x_1^{i_1}} \cdots \frac{\partial^{i_d}}{\partial x_r^{i_d}} f_*(\boldsymbol{x}) \right]$$

$$= \sum_{i \geq 0} \frac{\tilde{a}_i \mathbb{E}_{\boldsymbol{x}}[\nabla^i f_*(\boldsymbol{x})](\boldsymbol{w}^{\otimes i})}{i!}.$$

Where the final line is obtained noting that the tensor $\nabla^i f_*(\boldsymbol{x})$ has $\frac{i!}{i_1! \cdots i_d!}$ entries whose value is $\frac{\partial^{i_1}}{\partial x_1^{i_1}} \cdots \frac{\partial^{i_d}}{\partial x_r^{i_d}} f_*(\boldsymbol{x})$.

Note that $\mathbb{E}_{\boldsymbol{x}}[\nabla^i f_*(\boldsymbol{x})] = \mathbb{E}_{\boldsymbol{x}}\left[\nabla^i \sum_{j=Q}^{P} \frac{c_j}{j!} \mathrm{He}_j(\langle \boldsymbol{x}, \boldsymbol{\beta} \rangle)\right]$ is a nonzero tensor only when $Q \leq i \leq P$, from Lemma 8. We can also show (B.3) as

$$\mathbb{E}_{\boldsymbol{x}}[f_*(\boldsymbol{x})\sigma_b'(\boldsymbol{w}^\top \boldsymbol{x})\boldsymbol{x}]$$

$$= \mathbb{E}_{\boldsymbol{x}}[\nabla f_*(\boldsymbol{x})\sigma_b'(\boldsymbol{w}^\top \boldsymbol{x})] + \mathbb{E}_{\boldsymbol{x}}[f_*(\boldsymbol{x})\sigma_b''(\boldsymbol{w}^\top \boldsymbol{x})]\boldsymbol{w} \ (\because \text{Stein's lemma})$$

$$= \sum_{i \geq 0} \frac{a_{i+1}(b)\mathbb{E}_{\boldsymbol{x}}[\nabla^{i+1} f_*(\boldsymbol{x})](\boldsymbol{w}^{\otimes i})}{i!} + \boldsymbol{w} \sum_{i \geq 0} \frac{a_{i+2}(b)\mathbb{E}_{\boldsymbol{x}}[\nabla^i f_*(\boldsymbol{x})](\boldsymbol{w}^{\otimes i})}{i!}$$

$$= \sum_{i=Q-1}^{P-1} \frac{a_{i+1}(b)\mathbb{E}_{\boldsymbol{x}}[\nabla^{i+1} f_*(\boldsymbol{x})](\boldsymbol{w}^{\otimes i})}{i!} + \boldsymbol{w} \sum_{i=Q}^{P} \frac{a_{i+2}(b)\mathbb{E}_{\boldsymbol{x}}[\nabla^i f_*(\boldsymbol{x})](\boldsymbol{w}^{\otimes i})}{i!}$$

$\square$

Using Lemma 19, $\boldsymbol{g}(\boldsymbol{w}, b)$ can be expanded as

$$\boldsymbol{g}(\boldsymbol{w}, b) = \mathbb{E}_{f_*}\left[ 2\left( \sum_{i=Q-1}^{P-1} \frac{a_{i+1}(b)\mathbb{E}_{\boldsymbol{x}}[\nabla^{i+1} f_*(\boldsymbol{x})](\boldsymbol{w}^{\otimes i})}{i!} \right.\right.$$

$$\left.\left. + \boldsymbol{w} \sum_{i=Q}^{P} \frac{a_{i+2}(b)\mathbb{E}_{\boldsymbol{x}}[\nabla^i f_*(\boldsymbol{x})](\boldsymbol{w}^{\otimes i})}{i!} \right) \left( \sum_{i=Q}^{P} \frac{a_i(b)\mathbb{E}_{\boldsymbol{x}}[\nabla^i f_*(\boldsymbol{x})](\boldsymbol{w}^{\otimes i})}{i!} \right) \right]$$

$$=: \mathbb{E}_{f_*}\left[ \frac{2a_Q(b)}{(Q-1)!} \mathbb{E}_{\boldsymbol{x}}[\nabla^Q f_*(\boldsymbol{x})](\boldsymbol{w}^{\otimes(Q-1)}) \frac{a_Q(b)\mathbb{E}_{\boldsymbol{x}}[\nabla^Q f_*(\boldsymbol{x})](\boldsymbol{w}^{\otimes Q})}{Q!} \right] + \boldsymbol{s}(\boldsymbol{w}, b)$$

$$= \frac{2a_Q(b)^2}{Q!(Q-1)!} \mathbb{E}_{f_*}[\mathbb{E}_{\boldsymbol{x}}[\nabla^Q f_*(\boldsymbol{x})](\boldsymbol{w}^{\otimes(Q-1)}) \cdot \mathbb{E}_{\boldsymbol{x}}[\nabla^Q f_*(\boldsymbol{x})](\boldsymbol{w}^{\otimes Q})] + \boldsymbol{s}(\boldsymbol{w}, b), \quad \text{(B.4)}$$

where

$$\boldsymbol{s}(\boldsymbol{w}, b)$$

$$:= 2\mathbb{E}_{f_*}\left[ \left( \sum_{i=Q-1}^{P-1} \frac{a_{i+1}(b)\mathbb{E}_{\boldsymbol{x}}[\nabla^{i+1} f_*(\boldsymbol{x})](\boldsymbol{w}^{\otimes i})}{i!} + \boldsymbol{w} \sum_{i=Q}^{P} \frac{a_{i+2}(b)\mathbb{E}_{\boldsymbol{x}}[\nabla^i f_*(\boldsymbol{x})](\boldsymbol{w}^{\otimes i})}{i!} \right) \right.$$

$$\left. \cdot \left( \sum_{i=Q}^{P} \frac{a_i(b)\mathbb{E}_{\boldsymbol{x}}[\nabla^i f_*(\boldsymbol{x})](\boldsymbol{w}^{\otimes i})}{i!} \right) - \frac{a_Q(b)^2}{(Q-1)!} \mathbb{E}_{\boldsymbol{x}}[\nabla^Q f_*(\boldsymbol{x})](\boldsymbol{w}^{\otimes(Q-1)}) \frac{\mathbb{E}_{\boldsymbol{x}}[\nabla^Q f_*(\boldsymbol{x})](\boldsymbol{w}^{\otimes Q})}{Q!} \right]$$

expresses terms in the asymptotic expansion which are not the leading term.

Note that, in this explicit formulation, $a_i(b) = \mathbb{E}_{z \sim \mathcal{N}(0,1)}[\sigma_b(z)\mathrm{He}_i(z)]$ appears. It will be helpful to show that this coefficient is well-controlled.

**Lemma 20.** *There are constants $C_H$ and $C_L$ depending only on $Q$ and $P$ satisfying the following condition. Let $\sigma_b(z) = \sigma(z + b)$ and $b \sim \mathrm{Unif}([-1,1])$. Then,*

$$\left|\mathbb{E}_{z \sim \mathcal{N}(0,1)}[\sigma_b(z)\mathrm{He}_i(z)]\right| \leq C_H \ (2 \leq i \leq P + 2) \ and \ \left|\mathbb{E}_{z \sim \mathcal{N}(0,1)}[\sigma_b(z)\mathrm{He}_Q(z)]\right| \geq C_L \quad \text{(B.5)}$$

*holds with probability 1/2. Especially, $\left|\mathbb{E}_{z \sim \mathcal{N}(0,1)}[\sigma_b(z)\mathrm{He}_i(z)]\right| \leq C_H \ (2 \leq i \leq P + 2)$ holds with probability 1.*

**Proof.** From Lemma 15 in [BES$^+$23], we know that for $i \geq 2$, $\left|\mathbb{E}_{z \sim \mathcal{N}(0,1)}[\sigma_b(z)\mathrm{He}_i(z)]\right| = \frac{e^{-b^2/2}}{\sqrt{2\pi}}|\mathrm{He}_{i-2}(b)|$ holds. First, as $b \sim \mathrm{Unif}([-1,1])$, by setting sufficiently large $C_H$, $|\mathrm{He}_i(b)| \leq C_H$ holds for all $0 \leq i \leq P$. Therefore, as $\left|\mathbb{E}_{z \sim \mathcal{N}(0,1)}[\sigma_b(z)\mathrm{He}_i(z)]\right| \leq |\mathrm{He}_{i-2}(b)|$, $\left|\mathbb{E}_{z \sim \mathcal{N}(0,1)}[\sigma_b(z)\mathrm{He}_i(z)]\right| \leq C_H$ holds for all $2 \leq i \leq P + 2$ with probability one.

From continuity we know that there exists $C_L$ such that $\left|\mathbb{E}_{z \sim \mathcal{N}(0,1)}[\sigma_b(z)\mathrm{He}_Q(z)]\right| \geq C_L$ with probability 1/2. $\qquad \square$

In the following, we first calculate the main term and then upper bound the residual terms.

### B.1.2 Calculation of the Main Term

Let us calculate the main term in (B.4) explicitly.

**Lemma 21.**
$$\frac{2a_Q(b)^2}{Q!(Q-1)!}\mathbb{E}_{f_*}[\mathbb{E}_{\boldsymbol{x}}[\nabla^Q f_*(\boldsymbol{x})](\boldsymbol{w}^{\otimes(Q-1)}) \cdot \mathbb{E}_{\boldsymbol{x}}[\nabla^Q f_*(\boldsymbol{x})](\boldsymbol{w}^{\otimes Q})]$$

$$= \frac{2a_Q(b)^2 \mathbb{E}_{c_Q}[c_Q^2]}{Q!(Q-1)!} \frac{(2Q-1)!!}{\mathbb{E}_{z \sim \chi_r}[z^{2Q}]} \|\boldsymbol{w}_{1:r}\|^{2Q-2} \begin{bmatrix} \boldsymbol{w}_{1:r} \\ \boldsymbol{0}_{d-r} \end{bmatrix}$$

*holds.*

**Proof.** From Corollary 9, it holds that $\mathbb{E}_{\boldsymbol{x}}[\nabla^Q f_*(\boldsymbol{x})] = c_Q(\boldsymbol{\beta})^{\otimes Q}$. Then, we obtain

$$\mathbb{E}_{f_*}[\mathbb{E}_{\boldsymbol{x}}[\nabla^Q f_*(\boldsymbol{x})](\boldsymbol{w}^{\otimes(Q-1)}) \cdot \mathbb{E}_{\boldsymbol{x}}[\nabla^Q f_*(\boldsymbol{x})](\boldsymbol{w}^{\otimes Q})]$$
$$= \mathbb{E}_{c_Q}[c_Q^2]\mathbb{E}_{\boldsymbol{\beta}}[(\boldsymbol{\beta}^{\otimes Q}\boldsymbol{w}^{\otimes(Q-1)}) \cdot (\boldsymbol{\beta}^{\otimes Q} \circ \boldsymbol{w}^{\otimes Q})]$$
$$= \mathbb{E}_{c_Q}[c_Q^2]\mathbb{E}_{\boldsymbol{\beta}}[\boldsymbol{\beta}^{\otimes 2Q}](\boldsymbol{w}^{\otimes(2Q-1)}).$$

Next, let us show

$$\mathbb{E}_{\boldsymbol{\beta}}[\boldsymbol{\beta}^{\otimes 2Q}](\boldsymbol{w}^{\otimes(2Q-1)}) = \frac{(2Q-1)!!}{\mathbb{E}_{z \sim \chi_r}[z^{2Q}]}\|\boldsymbol{w}_{1:r}\|^{2Q-2} \begin{bmatrix} \boldsymbol{w}_{1:r} \\ \boldsymbol{0}_{d-r} \end{bmatrix}.$$

First, note that by letting $\boldsymbol{\beta}' \sim \mathcal{N}(0, \boldsymbol{\Sigma_\beta})$ where

$$\boldsymbol{\Sigma_\beta} = \begin{bmatrix} \boldsymbol{I}_r & \boldsymbol{0}_r \\ \boldsymbol{0}_r^\top & \boldsymbol{O}_{d-r} \end{bmatrix},$$

and $z \sim \chi_r$ independent from $\boldsymbol{\beta}$, $\boldsymbol{\beta}' \sim \boldsymbol{\beta} z$ holds (recall that we assumed that $\boldsymbol{\beta} \sim \mathrm{Unif}(\{(\beta_1, \ldots, \beta_r, 0, \ldots, 0)^\top \mid \beta_1^2 + \cdots + \beta_r^2 = 1\})$ – see Appendix A). Then,

$$\mathbb{E}_{\boldsymbol{\beta}}[\boldsymbol{\beta}^{\otimes 2Q}] = \frac{1}{\mathbb{E}_{z \sim \chi_r}[z^{2Q}]}\mathbb{E}_{\boldsymbol{v}'_{1:r} \sim \mathcal{N}(0, I_r), \boldsymbol{v}'_{r+1:d} = \boldsymbol{0}}[\boldsymbol{v}'^{\otimes 2Q}].$$

It suffices to show that $\mathbb{E}_{\boldsymbol{v} \sim \mathcal{N}(0, I_r)}[\boldsymbol{v}^{\otimes 2Q}](\boldsymbol{z}^{\otimes(2Q-1)}) = (2Q-1)!!\boldsymbol{z}\|\boldsymbol{z}\|^{2Q-2}$ for $\boldsymbol{z} \in \mathbb{R}^r$. As an example, we show that the first entry of $\mathbb{E}_{\boldsymbol{v} \sim \mathcal{N}(0, I_r)}[\boldsymbol{v}^{\otimes 2Q}](\boldsymbol{z}^{\otimes(2Q-1)})$ equals to $(2Q-1)!!z_1\|\boldsymbol{z}\|^{2Q-2}$: the other components can be calculated similarly. First, we expand $(\mathbb{E}_{\boldsymbol{v} \sim \mathcal{N}(0, I_r)}[\boldsymbol{v}^{\otimes 2Q}](\boldsymbol{z}^{\otimes(2Q-1)}))_1$ as

$$(\mathbb{E}_{\boldsymbol{v} \sim \mathcal{N}(0, I_r)}[\boldsymbol{v}^{\otimes 2Q}](\boldsymbol{z}^{\otimes(2Q-1)}))_1 = \sum_{i_1, \ldots, i_{2Q-1} \in [r]} \mathbb{E}[v_1 v_{i_1} \cdots v_{i_{2Q-1}}]z_{i_1} \cdots z_{i_{2Q-1}}.$$

Note that $\mathbb{E}[v_1 v_{i_1} \cdots v_{i_{2Q-1}}] \neq 0$ if and only if the degree of $v_1 v_{i_1} \cdots v_{i_{2Q-1}}$ with respect to $v_j$ is even, for all $j \in [r]$. Then, we obtain

$$(\mathbb{E}_{\boldsymbol{v} \sim \mathcal{N}(0, I_r)}[\boldsymbol{v}^{\otimes 2Q}](\boldsymbol{z}^{\otimes(2Q-1)}))_1$$

$$= \sum_{\substack{q_1+\cdots+q_r=Q \\ q_1\geq 1, q_2,\ldots,q_r\geq 0}} c_{q_1,\ldots,q_r}\mathbb{E}[v_1^{2q_1}v_2^{2q_2}\cdots v_r^{2q_r}]z_1^{2q_1-1}z_2^{2q_2}\cdots z_r^{2q_r}$$

$$=z_1 \sum_{\substack{q_1+\cdots+q_r=Q \\ q_1\geq 1, q_2,\ldots,q_r\geq 0}} c_{q_1,\ldots,q_r}\mathbb{E}[v_1^{2q_1}v_2^{2q_2}\cdots v_r^{2q_r}]z_1^{2q_1-2}z_2^{2q_2}\cdots z_r^{2q_r},$$

where $c_{q_1,\ldots,q_r} = \frac{(2Q-1)!}{(2q_1-1)!(2q_2)!\cdots(2q_r)!}$ is the number of ways to assign $(2q_1-1)$ "1"s, $(2q_2)$ "2"s, ..., among $i_1$ to $i_{2Q-1}$.

Note that $\mathbb{E}_{v\sim\mathcal{N}(0,1)}[v^{2q}] = (2q-1)!!$. Therefore, we obtain

$$(\mathbb{E}_{\boldsymbol{v}\sim\mathcal{N}(0,I_r)}[\boldsymbol{v}^{\otimes 2Q}](\boldsymbol{z}^{\otimes(2Q-1)}))_1$$

$$=z_1 \sum_{\substack{q_1+\cdots+q_r=Q \\ q_1\geq 1, q_2,\ldots,q_r\geq 0}} \frac{(2Q-1)!}{(2q_1-1)!(2q_2)!\cdots(2q_r)!}(2q_1-1)!!\cdots(2q_r-1)!!z_1^{2q_1-2}z_2^{2q_2}\cdots z_r^{2q_r}$$

$$=z_1 \sum_{\substack{q_1+\cdots+q_r=Q \\ q_1\geq 1, q_2,\ldots,q_r\geq 0}} \frac{(Q-1)!}{(q_1-1)!(q_2)!\cdots(q_r)!}(2Q-1)!!(z_1^2)^{q_1-1}(z_2^2)^{q_2}\cdots(z_r^2)^{q_r}$$

$$=(2Q-1)!!z_1(z_1^2+\cdots+z_r^2)^{Q-1} \qquad (\because \text{ multinomial theorem})$$

which concludes the assertion. $\qquad\square$

By Lemma 21, we observe that the main term of the expected one-step gradient aligns to the $r$-dimensional subspace $\mathcal{S}$. In other words, the MLP layer captures the information of $\mathcal{S}$ via pretraining, which will be useful in the later stage.

### B.1.3 Bounding Residual Terms

First, similarly to the Lemma 21, we can obtain the explicit formulation of the entire $\boldsymbol{g}(\boldsymbol{w}, b)$.

**Lemma 22.**

$$\boldsymbol{g}(\boldsymbol{w}, b)$$

$$= \sum_{\substack{Q-1\leq i\leq P-1, \\ Q\leq j\leq P, \\ i+j \text{ is odd}}} \frac{2a_{i+1}(b)a_j(b)\mathbb{E}[c_{i+1}c_j]}{i!j!}\frac{(i+j)!!}{\mathbb{E}_{z\sim\chi_r}[z^{i+j+1}]}\|\boldsymbol{w}_{1:r}\|^{i+j-1}\begin{bmatrix}\boldsymbol{w}_{1:r} \\ \boldsymbol{0}_{d-r}\end{bmatrix}$$

$$+ \boldsymbol{w} \sum_{\substack{Q\leq i\leq P, \\ Q\leq j\leq P, \\ i+j \text{ is even}}} \frac{2a_{i+2}(b)a_j(b)\mathbb{E}[c_ic_j]}{i!j!}\frac{(i+j-1)!!}{\mathbb{E}_{z\sim\chi_r}[z^{i+j}]}\|\boldsymbol{w}_{1:r}\|^{i+j} \qquad\qquad (\text{B.6})$$

holds.

**Proof.** Note that

$$\mathbb{E}_{\boldsymbol{\beta}}[\boldsymbol{\beta}^{\otimes n}](\boldsymbol{w}^{\otimes(n-1)}) = \begin{cases} \frac{(n-1)!!}{\mathbb{E}_{z\sim\chi_r}[z^n]}\|\boldsymbol{w}_{1:r}\|^{n-2}\begin{bmatrix}\boldsymbol{w}_{1:r} \\ \boldsymbol{0}_{d-r}\end{bmatrix} & (n \text{ is even}) \\ 0 \ (n \text{ is odd}) \end{cases}$$

and

$$\mathbb{E}_{\boldsymbol{\beta}}[\boldsymbol{\beta}^{\otimes n}](\boldsymbol{w}^{\otimes n}) = \begin{cases} \frac{(n-1)!!}{\mathbb{E}_{z\sim\chi_r}[z^n]}\|\boldsymbol{w}_{1:r}\|^n & (n \text{ is even}) \\ 0 \ (n \text{ is odd}) \end{cases},$$

which can be obtained similarly to the proof of Lemma 21. $\qquad\square$

Let us upper bound the non-leading terms; we need to show that the moment of first $r$ components in residual term $\boldsymbol{s}(\boldsymbol{w}, b)$ are sufficiently small.

**Lemma 23.** *Let $j = 4k$ where $k \in [P]$. then,*

$$\mathbb{E}_{\boldsymbol{w} \sim \mathbb{S}^{d-1}}[\|\boldsymbol{s}(\boldsymbol{w}, b)_{1:r}\|^j]^{1/j} = \tilde{O}_{d,r}\left(\sqrt{\frac{r}{d^{2Q+1}}}\right) \tag{B.7}$$

*holds uniformly over all $b \in [-1, 1]$.*

**Proof.** Note that $\|\boldsymbol{s}(\boldsymbol{w}, b)_{1:r}\|^2$ is a polynomial in $\boldsymbol{w}$; from Lemma 24 in [DLS22], it suffices to show the case $k = 1$.

$$\boldsymbol{s}(\boldsymbol{w}, b)_{1:r}$$

$$= \sum_{\substack{Q-1 \le i \le P-1, \\ Q \le j \le P, \\ i+j \text{ is odd}, \\ i,j \neq (Q-1,Q)}} \frac{2a_{i+1}(b)a_j(b)\mathbb{E}[c_{i+1}c_j]}{i!j!} \frac{(i+j)!!}{\mathbb{E}_{z \sim \chi_r}[z^{i+j+1}]} \|\boldsymbol{w}_{1:r}\|^{i+j-1}\boldsymbol{w}_{1:r}$$

$$+ \sum_{\substack{Q \le i \le P, \\ Q \le j \le P, \\ i+j \text{ is even}}} \frac{2a_{i+2}(b)a_j(b)\mathbb{E}[c_ic_j]}{i!j!} \frac{(i+j-1)!!}{\mathbb{E}_{z \sim \chi_r}[z^{i+j}]} \|\boldsymbol{w}_{1:r}\|^{i+j}\boldsymbol{w}_{1:r}.$$

By Minkowski's inequality $\mathbb{E}[\|\boldsymbol{x} + \boldsymbol{y}\|^4]^{1/4} \le \mathbb{E}[\|\boldsymbol{x}\|^4]^{1/4} + \mathbb{E}[\|\boldsymbol{y}\|^4]^{1/4}$, it suffices to show that (B.7) holds for each term. Note that $\mathbb{E}[c_{i+1}c_j] \le R_c^2$ and $\mathbb{E}[c_ic_j] \le R_c^2$ from Assumption 1, $|a_i(b)| \le C_H$ from Lemma 20, and $\mathbb{E}_{\boldsymbol{w}}[\|\boldsymbol{w}_{1:r}\|^{4k}] = O((r/d)^{2k})$ from Lemma 12. Moreover, it is known that $\mathbb{E}_{z \sim \chi_r}[z^{2l}] = \Theta(r^l)$. Putting these things together yields the assertion. $\qquad\square$

It is also needed in later stages to give a high probability upper bound on the $\|\boldsymbol{g}(\boldsymbol{w}, b)\|$;

**Lemma 24.**

$$\sup_{b \in [-1,1]} \|\boldsymbol{g}(\boldsymbol{w}, b)\| = \tilde{O}_{d,r}\left(\sqrt{\frac{1}{rd^{2Q-1}}}\right)$$

*holds with high probability over $\boldsymbol{w} \sim \mathrm{Unif}(\mathbb{S}^{d-1})$.*

**Proof.** This can be shown similarly to Lemma 23 – we use Lemma 11 instead of Lemma 12 to obtain a high probability bound (note that this bound includes not only $\boldsymbol{s}(\boldsymbol{w}, b)$ but also the main term, and it is for $\boldsymbol{g}(\boldsymbol{w}, b)$, instead of $\boldsymbol{g}(\boldsymbol{w}, b)_{1:r}$). $\qquad\square$

## B.2 Concentration of Empirical Gradient

Now we control the deviation of $\boldsymbol{g}_{T_1,N_1}(\boldsymbol{w}, b, \{(\boldsymbol{X}^t, \boldsymbol{y}^t, \boldsymbol{x}^t, y^t)\}_{t=1}^{T_1})$ from $\boldsymbol{g}(\boldsymbol{w}, b)$.

**Lemma 25.** *Under Assumption 1,*

$$\sup_{f_*, b \in [-1,1]} \|\mathbb{E}_{\boldsymbol{x}}[f_*(\boldsymbol{x})\sigma_b'(\boldsymbol{w}^\top \boldsymbol{x})\boldsymbol{x}]\| \lesssim \sqrt{\left(\frac{r}{d}\log d\right)^{Q-1}}, \tag{B.8}$$

$$\sup_{f_*, b \in [-1,1]} \|\mathbb{E}_{\boldsymbol{x}}[f_*(\boldsymbol{x})\sigma_b'(\boldsymbol{w}^\top \boldsymbol{x})\boldsymbol{x}]_{1:r}\| \lesssim \sqrt{\left(\frac{r}{d}\log d\right)^{Q-1}}, \tag{B.9}$$

$$\sup_{f_*, b \in [-1,1]} |\mathbb{E}_{\boldsymbol{x}}[f_*(\boldsymbol{x})\sigma_b(\boldsymbol{w}^\top \boldsymbol{x})]| \lesssim \sqrt{\left(\frac{r}{d}\log d\right)^Q} \tag{B.10}$$

*holds with high probability over $\boldsymbol{w} \sim \mathrm{Unif}(\mathbb{S}^{d-1})$, respectively. Here $\sup_{f_*}$ denotes the supremum over the support of $f_*$ whose distribution is specified in Assumption 1.*

**Proof.** We bound the leading term $\frac{a_Q(b)\mathbb{E}_{\boldsymbol{x}}[\nabla^Q f_*(\boldsymbol{x})](\boldsymbol{w}^{\otimes(Q-1)})}{(Q-1)!}$ in (B.3) of Lemma 19 to show (B.8). From Lemmas 5 and 6, we have

$$\left\|\frac{a_Q(b)\mathbb{E}_{\boldsymbol{x}}[\nabla^Q f_*(\boldsymbol{x})](\boldsymbol{w}^{\otimes(Q-1)})}{(Q-1)!}\right\|^2 = \left\|\frac{a_Q(b)c_Q}{(Q-1)!}(\boldsymbol{\beta})^{\otimes Q}(\boldsymbol{w}^{\otimes(Q-1)})\right\|^2$$

$$= \left\|\frac{a_Q(b)c_Q}{(Q-1)!}(\boldsymbol{\beta})^{\otimes Q}(\boldsymbol{w}_{1:r}^{\otimes(Q-1)})\right\|^2$$

$$\leq \left|\frac{a_Q(b)c_Q}{(Q-1)!}\right|^2 \|\boldsymbol{\beta}\|^{2Q}\|\boldsymbol{w}_{1:r}\|^{2(Q-1)}$$

$$= \left|\frac{a_Q(b)c_Q}{(Q-1)!}\right|^2 \|\boldsymbol{w}_{1:r}\|^{2(Q-1)}.$$

Moreover, Lemma 20 tells us that $|a_Q(b)| \leq C_H$ always holds when $b \in [-1,1]$, and Assumption 1 ensures that $|c_Q| \leq R_c$. Therefore, from Lemma 11, we can show that $\left\|\frac{a_Q(b)\mathbb{E}_{\boldsymbol{x}}[\nabla^Q f_*(\boldsymbol{x})](\boldsymbol{w}^{\otimes(Q-1)})}{(Q-1)!}\right\|^2 \leq \left(\frac{C_H R_c}{(Q-1)!}\right)^2 \left(\frac{4Cr}{d}\log d\right)^{Q-1}$ with probability at least $1 - 2\exp(-d/32) - 2rd^{-C}$. (B.9) and (B.10) can be obtained similarly. $\qquad\square$

The following lemma is parallel to Lemma 19 in [DLS22].

**Lemma 26.** *Fix any $t \in [T_1]$, $b \in [-1,1]$ and $\boldsymbol{w} \in \mathbb{S}^{d-1}$ satisfying the conditions* (B.8) *to* (B.10). *Then, with high probability over the distribution of $(\boldsymbol{X}^t, \boldsymbol{y}^t, \boldsymbol{x}^t, y^t)$, it holds that*

$$\left\|\frac{1}{N_1}\sum_{i=1}^{N_1} y_i^t \sigma_b'(\boldsymbol{w}^\top \boldsymbol{x}_i^t)\boldsymbol{x}_i^t - \mathbb{E}_{\boldsymbol{x}}[f_*^t(\boldsymbol{x})\sigma_b'(\boldsymbol{w}^\top \boldsymbol{x})\boldsymbol{x}]\right\| \leq \tilde{O}\left(\sqrt{\frac{d}{N_1}}\right),$$

$$\left\|\left(\frac{1}{N_1}\sum_{i=1}^{N_1} y_i^t \sigma_b'(\boldsymbol{w}^\top \boldsymbol{x}_i^t)\boldsymbol{x}_i^t - \mathbb{E}_{\boldsymbol{x}}[f_*^t(\boldsymbol{x})\sigma_b'(\boldsymbol{w}^\top \boldsymbol{x})\boldsymbol{x}]\right)_{1:r}\right\| \leq \tilde{O}\left(\sqrt{\frac{r}{N_1}}\right),$$

$$\left|\frac{1}{N_1}\sum_{i=1}^{N_1} y_i^t \sigma_b(\boldsymbol{w}^\top \boldsymbol{x}_i^t) - \mathbb{E}_{\boldsymbol{x}}[f_*^t(\boldsymbol{x})\sigma_b(\boldsymbol{w}^\top \boldsymbol{x})]\right| \leq \tilde{O}\left(\sqrt{\frac{1}{N_1}}\right).$$

*Here the right-hand sides do not depend on $t$.*

**Proof.** We only present the proof of the first inequality, as other bounds can be attained similarly. From Corollary 17, $|f_*^t(\boldsymbol{x})| \leq R$ where $R \lesssim (\log d)^{P/2}$ holds with high probability for all $t \in [T_1]$. Conditioned on this,

$$\left\|\frac{1}{N_1}\sum_{i=1}^{N_1} f_*^t(\boldsymbol{x}_i^t)\sigma_b'(\boldsymbol{w}^\top \boldsymbol{x}_i^t)\boldsymbol{x}_i^t - \mathbb{E}_{\boldsymbol{x}}[f_*^t(\boldsymbol{x})\sigma_b'(\boldsymbol{w}^\top \boldsymbol{x})\boldsymbol{x}]\right\|$$

$$\leq \left\|\frac{1}{N_1}\sum_{i=1}^{N_1} f_*^t(\boldsymbol{x}_i^t)\mathbb{I}_{f_*^t(\boldsymbol{x}_i^t)\leq R}\sigma_b'(\boldsymbol{w}^\top \boldsymbol{x}_i^t)\boldsymbol{x}_i^t - \mathbb{E}_{\boldsymbol{x}}[f_*^t(\boldsymbol{x})\mathbb{I}_{f_*^t(\boldsymbol{x})\leq R}\sigma_b'(\boldsymbol{w}^\top \boldsymbol{x})\boldsymbol{x}]\right\|$$

$$+ \left\|\mathbb{E}_{\boldsymbol{x}}[f_*^t(\boldsymbol{x})\mathbb{I}_{f_*^t(\boldsymbol{x})>R}\sigma_b'(\boldsymbol{w}^\top \boldsymbol{x})\boldsymbol{x}]\right\|$$

Note that $f_*^t(\boldsymbol{x}_i^t)\mathbb{I}_{f_*^t(\boldsymbol{x}_i^t)\leq R}\sigma_b'(\boldsymbol{w}^\top \boldsymbol{x}_i^t)\langle \boldsymbol{x}_i^t, \boldsymbol{u}\rangle - \mathbb{E}_{\boldsymbol{x}}[f_*^t(\boldsymbol{x})\mathbb{I}_{f_*^t(\boldsymbol{x})\leq R}\sigma_b'(\boldsymbol{w}^\top \boldsymbol{x})\langle \boldsymbol{x}, \boldsymbol{u}\rangle]$ is $R^2$-sub Gaussian for any $\|\boldsymbol{u}\| = 1$ from Lemma 14. Hence the first term can be upper bounded by $\tilde{O}\left(R\sqrt{\frac{d}{N_1}}\right)$ with high probability from Lemma 13. For the second term, we observe that

$$\left\|\mathbb{E}_{\boldsymbol{x}}[f_*^t(\boldsymbol{x})\mathbb{I}_{f_*^t(\boldsymbol{x})>R}\sigma_b'(\boldsymbol{w}^\top \boldsymbol{x})\boldsymbol{x}]\right\| \leq \mathbb{E}_{\boldsymbol{x}}[\|f_*^t(\boldsymbol{x})\mathbb{I}_{f_*^t(\boldsymbol{x})>R}\sigma_b'(\boldsymbol{w}^\top \boldsymbol{x})\boldsymbol{x}\|]$$

$$\leq \mathbb{E}_{\boldsymbol{x}}[f_*^t(\boldsymbol{x})^2]^{1/2}\mathbb{E}_{\boldsymbol{x}}[(\sigma_b'(\boldsymbol{w}^\top \boldsymbol{x})\|\boldsymbol{x}\|)^2]^{1/2}\mathbb{E}_{\boldsymbol{x}}[(\mathbb{I}_{f_*^t(\boldsymbol{x})>R})^4]^{1/4}$$

$$\leq \mathbb{E}_{\boldsymbol{x}}[f_*^t(\boldsymbol{x})^2]^{1/2}\mathbb{E}_{\boldsymbol{x}}[\|\boldsymbol{x}\|^2]^{1/2}\mathbb{E}_{\boldsymbol{x}}[(\mathbb{I}_{f_*^t(\boldsymbol{x})>R})]^{1/4}$$

$$\leq O(\sqrt{d}\cdot d^{-C/4})$$

for sufficiently large $C$; hence this term can be ignored. Finally, $\varsigma_i \sigma_b'(\boldsymbol{w}^\top \boldsymbol{x}_i^t)\langle \boldsymbol{x}_i^t, \boldsymbol{u}\rangle$ is $\tau^2$-sub Gaussian for any $\|\boldsymbol{u}\| = 1$ and then $\|\frac{1}{N_1}\sum_{i=1}^{N_1}\varsigma_i\sigma_b'(\boldsymbol{w}^\top \boldsymbol{x}_i^t)\boldsymbol{x}_i^t\| = \tilde{O}(\sqrt{d/N_1})$ with high probability. $\qquad\square$

By similar procedure, we can obtain a bound of $\|\boldsymbol{g}_{T_1,N_1}(\boldsymbol{w}, b, \{(\boldsymbol{X}^t, \boldsymbol{y}^t, \boldsymbol{x}^t, y^t)\}_{t=1}^{T_1}) - \boldsymbol{g}(\boldsymbol{w}, b)\|$.

**Lemma 27.** *Fix any $b \in [-1, 1]$ and $\boldsymbol{w} \in \mathbb{S}^{d-1}$ satisfying the conditions* (B.8)-(B.10). *Then, with high probability over the distribution of* $\{(\boldsymbol{X}^t, \boldsymbol{y}^t, \boldsymbol{x}^t, y^t)\}_{t=1}^{T_1}$,

$$\|\boldsymbol{g}_{T_1, N_1}(\boldsymbol{w}, b, \{(\boldsymbol{X}^t, \boldsymbol{y}^t, \boldsymbol{x}^t, y^t)\}_{t=1}^{T_1}) - \boldsymbol{g}(\boldsymbol{w}, b)\| = \tilde{O}\left(\sqrt{\frac{r^{Q-1}}{d^{Q-1}}}\sqrt{\frac{d}{T_1}} + \sqrt{\frac{d^2}{N_1 T_1}}\right),$$

$$\|(\boldsymbol{g}_{T_1, N_1}(\boldsymbol{w}, b, \{(\boldsymbol{X}^t, \boldsymbol{y}^t, \boldsymbol{x}^t, y^t)\}_{t=1}^{T_1}) - \boldsymbol{g}(\boldsymbol{w}, b))_{1:r}\| = \tilde{O}\left(\sqrt{\frac{r^{Q-1}}{d^{Q-1}}}\sqrt{\frac{r}{T_1}} + \sqrt{\frac{r^2}{N_1 T_1}}\right),$$

*holds.*

**Proof.** In this proof, the term "with high probability" indicates the probability with respect to the distribution of $\{(\boldsymbol{X}^t, \boldsymbol{y}^t, \boldsymbol{x}^t, y^t)\}_{t=1}^{T_1}$. Let

$$\boldsymbol{z}_1(\boldsymbol{w}, b, \{(\boldsymbol{X}^t, \boldsymbol{y}^t, \boldsymbol{x}^t, y^t)\}_{t=1}^{T_1}) = \sum_{t=1}^{T_1} \frac{1}{T_1} y^t \sigma_b(\boldsymbol{w}^\top \boldsymbol{x}^t) \frac{1}{N_1} \sum_{i=1}^{N_1} y_i^t \sigma_b'(\boldsymbol{w}^\top \boldsymbol{x}_i^t) \boldsymbol{x}_i^t$$

and

$$\boldsymbol{z}_2(\boldsymbol{w}, b, \{(\boldsymbol{X}^t, \boldsymbol{y}^t, \boldsymbol{x}^t, y^t)\}_{t=1}^{T_1}) = \sum_{t=1}^{T_1} \frac{1}{T_1} y^t \sigma_b'(\boldsymbol{w}^\top \boldsymbol{x}^t) \boldsymbol{x}^t \frac{1}{N_1} \sum_{i=1}^{N_1} y_i^t \sigma_b(\boldsymbol{w}^\top \boldsymbol{x}_i^t).$$

Note that $\boldsymbol{g}_{T_1, N_1} = \boldsymbol{z}_1 + \boldsymbol{z}_2$ holds.

Let us consider $\boldsymbol{z}_1(\boldsymbol{w}, b, \{(\boldsymbol{X}^t, \boldsymbol{y}^t, \boldsymbol{x}^t, y^t)\}_{t=1}^{T_1})$. Since we assume (B.8)-(B.10) hold, by taking union bound for Lemma 26 with respect to $t \in [T_1]$, with high probability $\|\frac{1}{N_1} \sum_{i=1}^{N_1} y_i^t \sigma_b'(\boldsymbol{w}^\top \boldsymbol{x}_i^t) \boldsymbol{x}_i^t\| \leq R = \tilde{O}\left(\sqrt{\frac{d}{N_1}} + \sqrt{\frac{r^{Q-1}}{d^{Q-1}}}\right)$ for each $t$. Moreover, from Corollary 17, $|y^t| \leq R' = \tilde{O}(1)$ for each $t$. Here, we notice that the procedure in the proof of Lemma 26 can be applied, yielding

$$\|\boldsymbol{z}_1(\boldsymbol{w}, b, \{(\boldsymbol{X}^t, \boldsymbol{y}^t, \boldsymbol{x}^t, y^t)\}_{t=1}^{T_1}) - \mathbb{E}[\boldsymbol{z}_1(\boldsymbol{w}, b, \{(\boldsymbol{X}^t, \boldsymbol{y}^t, \boldsymbol{x}^t, y^t)\}_{t=1}^{T_1})]\|$$
$$= \tilde{O}\left(R R' \sqrt{d/T_1}\right)$$
$$= \tilde{O}\left(\left(\sqrt{\frac{r^{Q-1}}{d^{Q-1}}} + \sqrt{\frac{d}{N_1}}\right)\sqrt{\frac{d}{T_1}}\right)$$

with high probability. Similarly we can obtain

- $\|\boldsymbol{z}_2 - \mathbb{E}[\boldsymbol{z}_2]\| = \tilde{O}\left(\left(\sqrt{\frac{r^Q}{d^Q}} + \sqrt{\frac{1}{N_1}}\right)\sqrt{\frac{d}{T_1}}\right)$,

- $\|(\boldsymbol{z}_1 - \mathbb{E}[\boldsymbol{z}_1])_{1:r}\| = \tilde{O}\left(\left(\sqrt{\frac{r^{Q-1}}{d^{Q-1}}} + \sqrt{\frac{r}{N_1}}\right)\sqrt{\frac{r}{T_1}}\right)$, and

- $\|(\boldsymbol{z}_2 - \mathbb{E}[\boldsymbol{z}_2])_{1:r}\| = \tilde{O}\left(\left(\sqrt{\frac{r^Q}{d^Q}} + \sqrt{\frac{1}{N_1}}\right)\sqrt{\frac{r}{T_1}}\right)$,

which concludes the proof. $\square$

We need to obtain a bound uniformly over $b \in [-1, 1]$. We first introduce the following definition.

**Definition 28.** *Fix any $\boldsymbol{w} \in \mathbb{S}^{d-1}$ and $X = \bigcup_{t=1}^{T_1} \{\boldsymbol{x}_1^t, \ldots, \boldsymbol{x}_{N_1}^t, \boldsymbol{x}^t\} \subset \mathbb{R}^d$. Then, define a finite disjoint partition $\mathcal{B}(\boldsymbol{w}, X) = \{B_1, \ldots, B_{N(\boldsymbol{w}, X)}\}$ of $[-1, 1]$, i.e., $\cup_i B_i = [-1, 1]$ and $B_i \cap B_j = \emptyset$ $(i \neq j)$ as follows: $b$ and $b'$ belong to the same $B_i$ if and only if $\mathrm{sign}(\langle \boldsymbol{w}, \boldsymbol{x}\rangle + b) = \mathrm{sign}(\langle \boldsymbol{w}, \boldsymbol{x}\rangle + b')$ for all $\boldsymbol{x} \in X$.*

It is clear that $|\mathcal{B}(\boldsymbol{w}, X)| \lesssim |X| \leq (N_1 + 1)T_1$ holds because it suffices to divide $[-1, 1]$ at the point satisfying $\langle \boldsymbol{w}, \boldsymbol{x}\rangle + b = 0$ for each $\boldsymbol{x} \in X$.

**Lemma 29.** *Fix any $\boldsymbol{w} \in \mathbb{S}^{d-1}$. Then, with high probability over the distribution of $\{(\boldsymbol{X}^t, \boldsymbol{y}^t, \boldsymbol{x}^t, y^t)\}_{t=1}^{T_1}$, the following holds:*

- $g(\boldsymbol{w}, b)$ is $L_1$-Lipschitz continuous with respect to $b$ in the interval $[-1, 1]$.

- Let $X = \bigcup_{t=1}^{T_1} \{\boldsymbol{x}_1^t, \ldots, \boldsymbol{x}_{N_1}^t, \boldsymbol{x}^t\}$. Then, $g_{T_1,N_1}(\boldsymbol{w}, b, \{(\boldsymbol{X}^t, \boldsymbol{y}^t, \boldsymbol{x}^t, y^t)\}_{t=1}^{T_1})$ is $L_2$-Lipschitz continuous with respect to $b$ in each interval $B_i \in \mathcal{B}(\boldsymbol{w}, X)$.

Here $L_1 = O(1)$ and $L_2 = \tilde{O}(\sqrt{d})$ do not depend on $\boldsymbol{w}$.

**Proof.** By the explicit form of $g(\boldsymbol{w}, b)$ (B.6), we notice that each term is made by taking the product between the term as $a_j(b)a_k(b)$ and a vector independent of $b$ whose norm is $O(1)$. Thus, it suffices to show that $a_j(b)a_k(b)$ is $O(1)$-Lipschitz. This follows from the fact $a_j = \pm \frac{e^{-b^2/2}}{\sqrt{2\pi}} \mathrm{He}_{j-2}(b)$ (see the proof of Lemma 20) and $\mathrm{He}_{j-2}(b)e^{-b^2/2}$ is $O(1)$-Lipschitz continuous on the bounded set $[-1, 1]$.

Next, let us see $\boldsymbol{g}_{T_1,N_1}(\boldsymbol{w}, b, \{(\boldsymbol{X}^t, \boldsymbol{y}^t, \boldsymbol{x}^t, y^t)\}_{t=1}^{T_1})$. Recall that

$$\boldsymbol{g}_{T_1,N_1}(\boldsymbol{w}, b, \{(\boldsymbol{X}^t, \boldsymbol{y}^t, \boldsymbol{x}^t, y^t)\}_{t=1}^{T_1})$$
$$:= \sum_{i=1}^{T_1} \frac{1}{T_1} y^t \sigma_b(\boldsymbol{w}^\top \boldsymbol{x}^t) \frac{1}{N_1} \sum_{i=1}^{N_1} y_i^t \sigma_b'(\boldsymbol{w}^\top \boldsymbol{x}_i^t) \boldsymbol{x}_i^t + \sum_{i=1}^{T_1} \frac{1}{T_1} y^t \sigma_b'(\boldsymbol{w}^\top \boldsymbol{x}^t) \boldsymbol{x}^t \frac{1}{N_1} \sum_{i=1}^{N_1} y_i^t \sigma_b(\boldsymbol{w}^\top \boldsymbol{x}_i^t).$$

From Lemma 10 and Corollary 17, we know that $|y_i^t|, |y^t| \lesssim (\log d)^{P/2}$ and $\|\boldsymbol{x}_i^t\|, \|\boldsymbol{x}^t\| \lesssim \sqrt{d}$ holds for all $i, t$ with high probability. Assuming this and noting that $\sigma$ is 1-Lipschitz, in each interval in $\mathcal{B}(\boldsymbol{w}, X)$, $\boldsymbol{g}_{T_1,N_1}(\boldsymbol{w}, b, \{(\boldsymbol{X}^t, \boldsymbol{y}^t, \boldsymbol{x}^t, y^t)\}_{t=1}^{T_1})$ is $L_2 = \tilde{O}(\sqrt{d})$-Lipschitz. $\qquad \square$

**Corollary 30.** *With high probability over the distribution of $\boldsymbol{w}$ and $\{(\boldsymbol{X}^t, \boldsymbol{y}^t, \boldsymbol{x}^t, y^t)\}_{t=1}^{T_1}$,*

$$\sup_{b \in [-1,1]} \|\boldsymbol{g}_{T_1,N_1}(\boldsymbol{w}, b, \{(\boldsymbol{X}^t, \boldsymbol{y}^t, \boldsymbol{x}^t, y^t)\}_{t=1}^{T_1}) - \boldsymbol{g}(\boldsymbol{w}, b)\| = \tilde{O}\left(\sqrt{\frac{r^{Q-1}}{d^{Q-1}}} \sqrt{\frac{d}{T_1}} + \sqrt{\frac{d^2}{N_1 T_1}}\right),$$

$$\sup_{b \in [-1,1]} \|(\boldsymbol{g}_{T_1,N_1}(\boldsymbol{w}, b, \{(\boldsymbol{X}^t, \boldsymbol{y}^t, \boldsymbol{x}^t, y^t)\}_{t=1}^{T_1}) - \boldsymbol{g}(\boldsymbol{w}, b))_{1:r}\| = \tilde{O}\left(\sqrt{\frac{r^{Q-1}}{d^{Q-1}}} \sqrt{\frac{r}{T_1}} + \sqrt{\frac{r^2}{N_1 T_1}}\right)$$
(B.11)

*holds.*

**Proof.** Let $X = \bigcup_{t=1}^{T_1} \{\boldsymbol{x}_1^t, \ldots, \boldsymbol{x}_{N_1}^t, \boldsymbol{x}^t\}$ and consider $\mathcal{B}(\boldsymbol{w}, X) = \{B_1, \ldots, B_{N(\boldsymbol{w},X)}\}$. Then, let $B$ be the union of $\tilde{O}\left(\left(\sqrt{\frac{r^{Q-1}}{d^{Q-1}}}\sqrt{\frac{r}{T_1}} + \sqrt{\frac{r^2}{N_1 T_1}}\right)/\sqrt{d}\right)$-coverings of $B_i \in \mathcal{B}(\boldsymbol{w}, X)$. Since $|B|$ is polynomial in $d$, Lemma 27 holds with high probability uniformly over $b \in B$.

Moreover, from the construction and Lemma 29, for $b \in [-1, 1] \setminus B$, there exists $\pi(b) \in B$ such that

$$\bigl|\|\boldsymbol{g}_{T_1,N_1}(\boldsymbol{w}, b, \{(\boldsymbol{X}^t, \boldsymbol{y}^t, \boldsymbol{x}^t, y^t)\}_{t=1}^{T_1}) - \boldsymbol{g}(\boldsymbol{w}, b)\|$$
$$- \|\boldsymbol{g}_{T_1,N_1}(\boldsymbol{w}, \pi(b), \{(\boldsymbol{X}^t, \boldsymbol{y}^t, \boldsymbol{x}^t, y^t)\}_{t=1}^{T_1}) - \boldsymbol{g}(\boldsymbol{w}, \pi(b))\|\bigr|$$
$$\leq \tilde{O}\left(\sqrt{\frac{r^{Q-1}}{d^{Q-1}}} \sqrt{\frac{r}{T_1}} + \sqrt{\frac{r^2}{N_1 T_1}}\right)$$

holds. $\qquad \square$

**Lemma 31.** *With high probability over the distribution of $\{(\boldsymbol{X}^t, \boldsymbol{y}^t, \boldsymbol{x}^t, y^t)\}_{t=1}^{T_1}$,*

- *It holds that*

$$\sup_{b \in [-1,1]} \|\boldsymbol{g}(\boldsymbol{w}, b) - \boldsymbol{g}_{T_1,N_1}(\boldsymbol{w}, b, \{(\boldsymbol{X}^t, \boldsymbol{y}^t, \boldsymbol{x}^t, y^t)\}_{t=1}^{T_1})\|$$

$$= \tilde{O}\left(\sqrt{\frac{r^{Q-1}}{d^{Q-1}}} \sqrt{\frac{d}{T_1}} + \sqrt{\frac{d^2}{N_1 T_1}}\right)$$
(B.12)

*with high probability over $\boldsymbol{w} \sim \mathrm{Unif}(\mathbb{S}^{d-1})$.*

- *It holds that*

$$\sup_{b\in[-1,1]} \mathbb{E}_{\boldsymbol{w}\sim\mathrm{Unif}(\mathbb{S}^{d-1})}[\|\boldsymbol{g}(\boldsymbol{w},b) - \boldsymbol{g}_{T_1,N_1}(\boldsymbol{w},b,\{(\boldsymbol{X}^t,\boldsymbol{y}^t,\boldsymbol{x}^t,y^t)\}_{t=1}^{T_1})\|^j]^{1/j}$$

$$=\tilde{O}\left(\sqrt{\frac{r^{Q-1}}{d^{Q-1}}}\sqrt{\frac{r}{T_1}} + \sqrt{\frac{r^2}{N_1 T_1}}\right) \tag{B.13}$$

*for $j \in [4P]$.*

**Proof.** If an event $A$ occurs with probability at least $1 - d^{-C}$ over the simultaneous distribution of $(\boldsymbol{w}, \{(\boldsymbol{X}^t,\boldsymbol{y}^t,\boldsymbol{x}^t,y^t)\}_{t=1}^{T_1})$, then we can say that

- with probability at least $1 - d^{C/2}$ over $\{(\boldsymbol{X}^t,\boldsymbol{y}^t,\boldsymbol{x}^t,y^t)\}_{t=1}^{T_1}$, $\mathbb{P}_{\boldsymbol{w}}\{A \mid \{(\boldsymbol{X}^t,\boldsymbol{y}^t,\boldsymbol{x}^t,y^t)\}_{t=1}^{T_1})\} \geq 1 - d^{-C/2}$ holds.

(B.12) follows from this remark and Corollary 30.

For (B.13), with high probability over the distribution of $\{(\boldsymbol{X}^t,\boldsymbol{y}^t,\boldsymbol{x}^t,y^t)\}_{t=1}^{T_1}$, $\mathbb{P}_{\boldsymbol{w}}\{$(B.11) $\mid \{(\boldsymbol{X}^t,\boldsymbol{y}^t,\boldsymbol{x}^t,y^t)\}_{t=1}^{T_1}\} \geq 1 - d^{-\tilde{C}}$ is satisfied for some $\tilde{C}$. Also, from Lemma 10 and Corollary 17, with high probability over the distribution of $\{(\boldsymbol{X}^t,\boldsymbol{y}^t,\boldsymbol{x}^t,y^t)\}_{t=1}^{T_1}$, we have $\|\boldsymbol{x}_i^t\|, \|\boldsymbol{x}^t\| = O(\sqrt{d})$ and $|y^t| = \tilde{O}(1)$. From the definition of $\boldsymbol{g}_{T_1,N_1}(\boldsymbol{w},b,\{(\boldsymbol{X}^t,\boldsymbol{y}^t,\boldsymbol{x}^t,y^t)\}_{t=1}^{T_1})$ (B.2) and expansion of $\boldsymbol{g}(\boldsymbol{w},b)$ (B.6), $\sup_{\boldsymbol{w}} \|\boldsymbol{g}(\boldsymbol{w},b) - \boldsymbol{g}_{T_1,N_1}(\boldsymbol{w},b,\{(\boldsymbol{X}^t,\boldsymbol{y}^t,\boldsymbol{x}^t,y^t)\}_{t=1}^{T_1})\| \leq \sup_{\boldsymbol{w}} \|\boldsymbol{g}(\boldsymbol{w},b)\| + \sup_{\boldsymbol{w}} \|\boldsymbol{g}_{T_1,N_1}(\boldsymbol{w},b,\{(\boldsymbol{X}^t,\boldsymbol{y}^t,\boldsymbol{x}^t,y^t)\}_{t=1}^{T_1})\|$ is upper bounded by $\tilde{O}(d)$. Therefore, with high probability

$$\sup_{b\in[-1,1]} \mathbb{E}_{\boldsymbol{w}\sim\mathrm{Unif}(\mathbb{S}^{d-1})}[\|\boldsymbol{g}(\boldsymbol{w},b) - \boldsymbol{g}_{T_1,N_1}(\boldsymbol{w},b,\{(\boldsymbol{X}^t,\boldsymbol{y}^t,\boldsymbol{x}^t,y^t)\}_{t=1}^{T_1})\|^j]^{1/j}$$

$$\leq \mathbb{E}_{\boldsymbol{w}\sim\mathrm{Unif}(\mathbb{S}^{d-1})}\left[\left(\sup_{b\in[-1,1]} \|\boldsymbol{g}(\boldsymbol{w},b) - \boldsymbol{g}_{T_1,N_1}(\boldsymbol{w},b,\{(\boldsymbol{X}^t,\boldsymbol{y}^t,\boldsymbol{x}^t,y^t)\}_{t=1}^{T_1})\|\right)^j\right]^{1/j}$$

$$=\tilde{O}\left(\left(\left(1 - d^{-\tilde{C}}\right)\left(\sqrt{\frac{r^{Q-1}}{d^{Q-1}}}\sqrt{\frac{r}{T_1}} + \sqrt{\frac{r^2}{N_1 T_1}}\right)^j + \left(d^{-\tilde{C}}\right)d^j\right)^{1/j}\right).$$

By setting sufficiently large $\tilde{C}$, we arrive at the claim. $\qquad\square$

**Summary.** We combine the obtained results and show that $\|\boldsymbol{g}_{T_1,N_1}(\boldsymbol{w},b,\{(\boldsymbol{X}^t,\boldsymbol{y}^t,\boldsymbol{x}^t,y^t)\}_{t=1}^{T_1})\|$ and the moment of the residual terms are bounded with high probability.

**Corollary 32.** *With high probability over the distribution of $\{(\boldsymbol{X}^t,\boldsymbol{y}^t,\boldsymbol{x}^t,y^t)\}_{t=1}^{T_1}$,*

(I) $\boldsymbol{g}_{T_1,N_1}(\boldsymbol{w},b,\{(\boldsymbol{X}^t,\boldsymbol{y}^t,\boldsymbol{x}^t,y^t)\}_{t=1}^{T_1}) = \boldsymbol{m}(\boldsymbol{w},b) + \boldsymbol{r}(\boldsymbol{w},b,\{(\boldsymbol{X}^t,\boldsymbol{y}^t,\boldsymbol{x}^t,y^t)\}_{t=1}^{T_1})$ *holds where*

$$\boldsymbol{m}(\boldsymbol{w},b) := \frac{2a_Q(b)^2\mathbb{E}_{c_Q}[c_Q^2]}{Q!(Q-1)!}\frac{(2Q-1)!!}{\mathbb{E}_{z\sim\chi_r}[z^{2Q}]}\|\boldsymbol{w}_{1:r}\|^{2Q-2}\begin{bmatrix}\boldsymbol{w}_{1:r}\\\boldsymbol{0}_{d-r}\end{bmatrix},$$

*and*

$$\sup_{b\in[-1,1]} \mathbb{E}_{\boldsymbol{w}\sim\mathrm{Unif}(\mathbb{S}^{d-1})}\left[\left\|\left(\boldsymbol{r}(\boldsymbol{w},b,\{(\boldsymbol{X}^t,\boldsymbol{y}^t,\boldsymbol{x}^t,y^t)\}_{t=1}^{T_1})\right)_{1:r}\right\|^j\right]^{1/j}$$

$$=\tilde{O}_{d,r}\left(\sqrt{\frac{r^{Q-1}}{d^{Q-1}}}\sqrt{\frac{r}{T_1}} + \sqrt{\frac{r^2}{N_1 T_1}} + \sqrt{\frac{r}{d^{2Q+1}}}\right).$$

*(II) If $\eta_1\gamma \leq \sqrt{rd^{2Q-1}}/C_1(\log d)^{C_2}$ for sufficiently large $C_1$ and $C_2$, $T_1 \gtrsim r^Q d^{Q+1}$, and $N_1 T_1 \gtrsim rd^{2Q+1}$, then*

$$\sup_{b\in[-1,1]} \|2\eta_1\gamma\boldsymbol{g}_{T_1,N_1}(\boldsymbol{w},b,\{(\boldsymbol{X}^t,\boldsymbol{y}^t,\boldsymbol{x}^t,y^t)\}_{t=1}^{T_1})\| \leq 1$$

*holds with high probability over $\boldsymbol{w} \sim \mathrm{Unif}(\mathbb{S}^{d-1})$. Moreover,*

$$\sup_{b\in[-1,1]} \left| \left\langle 2\eta_1\gamma\boldsymbol{g}_{T_1,N_1}(\boldsymbol{w},b,\{(\boldsymbol{X}^t,\boldsymbol{y}^t,\boldsymbol{x}^t,y^t)\}_{t=1}^{T_1}),\boldsymbol{z}\right\rangle \right| \leq \log d$$

*holds with high probability over $\boldsymbol{w} \sim \mathrm{Unif}(\mathbb{S}^{d-1})$ and the training data for the second stage $\{(\boldsymbol{X}^t,\boldsymbol{y}^t,\boldsymbol{x}^t,y^t)\}_{t=T_1+1}^{T_2}$, where $\boldsymbol{z} \in \bigcup_{t=T_1+1}^{T_2}\{\boldsymbol{x}_1^t,\ldots,\boldsymbol{x}_{N_2}^t,\boldsymbol{x}^t\}$.*

**Proof.** Combining Lemmas 23 and 31 yields (I). For (II), by the condition for $T_1$ and $N_1$, we obtain

$$\sup_{b\in[-1,1]} \|\boldsymbol{g}_{T_1,N_1}(\boldsymbol{w},b,\{(\boldsymbol{X}^t,\boldsymbol{y}^t,\boldsymbol{x}^t,y^t)\}_{t=1}^{T_1})\| \leq \tilde{O}_{d,r}\left( \sqrt{\frac{r^{Q-1}}{d^{Q-1}}}\sqrt{\frac{d}{T_1}} + \sqrt{\frac{d^2}{N_1 T_1}} + \sqrt{\frac{1}{rd^{2Q-1}}} \right)$$

$$= \tilde{O}_{d,r}\left( \sqrt{\frac{1}{rd^{2Q-1}}} \right)$$

from Lemmas 24 and 31. Then, as $\eta_1\gamma \lesssim \sqrt{rd^{2Q-1}}/C_1(\log d)^{C_2}$, $\sup_{b\in[-1,1]} \|2\eta_1\gamma\boldsymbol{g}_{T_1,N_1}(\boldsymbol{w},b,\{(\boldsymbol{X}^t,\boldsymbol{y}^t,\boldsymbol{x}^t,y^t)\}_{t=1}^{T_1})\| \leq 1$ is achieved.

By an argument similar to the one in the proof of Lemma 18, we obtain that with high probability, $\left|\left\langle 2\eta_1\gamma\boldsymbol{g}_{T_1,N_1}(\boldsymbol{w},b,\{(\boldsymbol{X}^t,\boldsymbol{y}^t,\boldsymbol{x}^t,y^t)\}_{t=1}^{T_1}),\boldsymbol{z}\right\rangle\right| \lesssim \log d$ for $\boldsymbol{z} \in \bigcup_{t=T_1+1}^{T_2}\{\boldsymbol{x}_1^t,\ldots,\boldsymbol{x}_{N_2}^t,\boldsymbol{x}^t\}$. To obtain the supremum bound over $b \in [-1,1]$, it suffices to utilize Lemma 29 again. $\qquad\square$

In the following sections, we assume that these two events (I) and (II) occur.

## C   Construction of Attention Matrix

We construct an attention matrix $\bar{\boldsymbol{\Gamma}}$ with a good approximation property: see Section 3.2.2 for the proof outline.

In this section, a set of basis functions $\mathcal{H}$ is defined as

$$\mathcal{H} = \left\{ \boldsymbol{x} \mapsto \prod_{j=1}^{r} \frac{1}{\sqrt{j!}}\mathrm{He}_{p_j}(x_j) \middle| Q \leq p_1 + \cdots + p_r \leq P, p_1 \geq 0,\ldots,p_r \geq 0 \right\}.$$

We let $B_P := |\mathcal{H}|$ and introduce a numbering $\mathcal{H} = \{h_1,\ldots,h_{B_P}\}$. Note that

$$B_P = \sum_{i=Q}^{P}\binom{r+i-1}{i} = \Theta(r^P).$$

### C.1   Approximation Error of Two-layer Neural Network

As sketched in Section 3.2.2, we need to show that there is a set of two-layer neural networks which can approximate basis functions well. The goal of this section is to show the following proposition.

**Proposition 33.** *Assume $T_1 = \tilde{\Omega}(d^{Q+1}r^Q)$ and $N_1 T_1 = \tilde{\Omega}(d^{2Q+1}r)$. Let $\{\boldsymbol{w}_j^{(1)}\}_j$ be neurons obtained by line 4 of Algorithm 1 with $\gamma \asymp \frac{1}{m^{\frac{3}{2}}r^{\frac{1}{2}}d^Q}$, $\eta_1 \asymp m^{\frac{3}{2}}rd^{2Q-\frac{1}{2}}\cdot(\log d)^{-C_\eta}$ for constant $C_\eta$. Assume $\boldsymbol{b}$ is re-initialized in Line 5 of Algorithm 1. Then, with high probability over the distribution of training data and the initialization of model parameters, there exist vectors $\boldsymbol{a}^1,\ldots,\boldsymbol{a}^{B_P} \in \mathbb{R}^m$ such that for each $n \in [B_P]$,*

$$\sup_{i\in[N_2],T_1+1\leq t\leq T_2} \left( \sum_{j=1}^{m} a_j^n\sigma(\boldsymbol{w}_j^{(1)\top}\boldsymbol{x}_i^t + b_j) - h_n(\boldsymbol{x}_i^t) \right)^2 = \tilde{O}\left( \frac{r^P}{m} + \frac{1}{N_2} \right),$$

$$\sup_{T_1+1\leq t\leq T_2}\left(\sum_{j=1}^m a_j^n\sigma(\boldsymbol{w}_j^{(1)\top}\boldsymbol{x}^t+b_j)-h_n(\boldsymbol{x}^t)\right)^2=\tilde{O}\left(\frac{r^P}{m}+\frac{1}{N_2}\right)$$

holds. Moreover, $\sup_{n\in[B_P]}\|\boldsymbol{a}^n\|^2=\tilde{O}\left(\frac{r^P}{m}\right)$.

This proposition states that there exists a two-layer neural network $\sum_{j=1}^m a_j^n\sigma\left(\langle\boldsymbol{w}_j^{(1)},\cdot\rangle+b_j\right)$ which can approximate the basis function $h_n$, and its second-layer magnitude $\|\boldsymbol{a}^n\|^2$ only scales with $r$. Note that the condition of $T_1$ and $N_1T_1$ comes from Corollary 32, which implies that the noise terms of the one-step gradient is well controlled.

### C.1.1 Alignment to $\mathcal{S}$ and Efficient Approximation

From now on, we suppose that $\boldsymbol{g}_{T_1,N_1}(\boldsymbol{w},b)$ satisfies the conditions in Corollary 32 (to simplify the notation, we drop the dependency on $\{(\boldsymbol{X}^t,\boldsymbol{y}^t,\boldsymbol{x}^t,y^t)\}_{t=1}^{T_1}$). An important previous result [DLS22] is that, if $\boldsymbol{g}_{T_1,N_1}(\boldsymbol{w},b)$ aligns with the $r$-dimensional subspace $\mathcal{S}$, then it can be used to approximate polynomials on $\mathcal{S}$ efficiently.

**Lemma 34** (Lemma 21 in [DLS22]). *There exists a set of symmetric $r$-dimensional tensors $\{\boldsymbol{T}_k^n\}_{0\leq k\leq P,n\in[B_P]}$ such that*

$$h_n(\boldsymbol{x})=\sum_{0\leq k\leq P}\langle\boldsymbol{T}_k^n,\boldsymbol{x}_{1:r}^{\otimes k}\rangle$$

*and $\|\boldsymbol{T}_k^n\|_F\lesssim r^{\frac{P-k}{4}}$.*

**Definition 35** ($f(r,d)$-alignment). *We call that $\boldsymbol{g}_{T_1,N_1}(\boldsymbol{w},b)$ has $f(r,d)$-alignment with $\mathcal{S}$, if the following holds:*

- *For any $0\leq k\leq P$-symmetric $r$-dimensional tensor $\boldsymbol{T}$,*

$$\mathbb{E}_{\boldsymbol{w}\sim\mathrm{Unif}(\mathbb{S}^{d-1})}[\langle\boldsymbol{T},\boldsymbol{g}_{T_1,N_1}(\boldsymbol{w},b)_{1:r}^{\otimes k}\rangle^2]\gtrsim f(r,d)^k\|\boldsymbol{T}\|_F^2,$$

  *and the hidden constant in "$\gtrsim$" needs to be uniformly lower bounded.*

**Lemma 36** (Corollary 4 in [DLS22], adapted). *Suppose that $\boldsymbol{g}_{T_1,N_1}(\boldsymbol{w},b)$ has $f(r,d)$-alignment with $\mathcal{S}$. Let $\boldsymbol{T}$ be a $0\leq k\leq P$-symmetric $r$-dimensional tensor. Then, there exists $\psi_{\boldsymbol{T}}(\boldsymbol{w},b)$ such that*

$$\mathbb{E}_{\boldsymbol{w}\sim\mathrm{Unif}(\mathbb{S}^{d-1})}[\psi_{\boldsymbol{T}}(\boldsymbol{w},b)\langle\boldsymbol{g}_{T_1,N_1}(\boldsymbol{w},b),\boldsymbol{x}\rangle^k]=\langle\boldsymbol{T},\boldsymbol{x}_{1:r}^{\otimes k}\rangle,$$
$$\mathbb{E}_{\boldsymbol{w}\sim\mathrm{Unif}(\mathbb{S}^{d-1})}[\psi_{\boldsymbol{T}}(\boldsymbol{w},b)^2]\lesssim f(r,d)^{-k}\|\boldsymbol{T}\|_F^2,$$
$$|\psi_{\boldsymbol{T}}(\boldsymbol{w},b)|\lesssim f(r,d)^{-k}\|\boldsymbol{T}\|_F\|\boldsymbol{g}_{T_1,N_1}(\boldsymbol{w},b)\|^k.$$

Intuitively speaking, $f(r,d)$-alignment ensures that the MLP gradient $\boldsymbol{g}_{T_1,N_1}(\boldsymbol{w},b)$ has sufficiently large component on $\mathcal{S}$. If so, the lemma states that, an infinite-width neural network $\mathbb{E}_{\boldsymbol{w}}[\psi_{\boldsymbol{T}}(\boldsymbol{w},b)\langle\boldsymbol{g}_{T_1,N_1}(\boldsymbol{w},b),\cdot\rangle^k]$, whose hidden layer weight is $\boldsymbol{g}_{T_1,N_1}(\boldsymbol{w},b)$ and the activation is $z\mapsto z^k$, can express the functions $\langle\boldsymbol{T},\boldsymbol{x}_{1:r}^{\otimes k}\rangle$ with small output layer $\psi_{\boldsymbol{T}}(\boldsymbol{w},b)$.

Let us show that our $\boldsymbol{g}_{T_1,N_1}(\boldsymbol{w},b)$ has sufficient alignment if $b$ satisfies (B.5).

**Lemma 37** (Tensor expectation lower bound). *Let $\boldsymbol{T}$ be a $k\leq P$-symmetric $r$-dimensional tensor and $\bar{\boldsymbol{w}}=\boldsymbol{w}_{1:r}\|\boldsymbol{w}_{1:r}\|^{2Q-2}$. Then*

$$\mathbb{E}_{\boldsymbol{w}\sim\mathrm{Unif}(\mathbb{S}^{d-1})}[\boldsymbol{T}(\bar{\boldsymbol{w}}^{\otimes k})^2]\gtrsim\frac{r^{Qi}}{d^{(Q+1)i}}\mathbb{E}[\|\boldsymbol{T}(\bar{\boldsymbol{w}}^{\otimes k-i})\|_F^2]$$

*holds for all $0\leq i\leq k$. Here $\boldsymbol{T}(\bar{\boldsymbol{w}}^{\otimes 0}):=\boldsymbol{T}$.*

**Proof.** Let $\boldsymbol{u}\sim\mathcal{N}(0,I_d)$, $z\sim\chi(d)$ and $\bar{\boldsymbol{u}}=\boldsymbol{u}_{1:r}\|\boldsymbol{u}_{1:r}\|^{2Q-2}$. We can decompose $\bar{\boldsymbol{u}}$ as $\bar{\boldsymbol{u}}=z^{2Q-1}\bar{\boldsymbol{w}}$. Therefore

$$\mathbb{E}[\boldsymbol{T}(\bar{\boldsymbol{u}}^{\otimes k})^2]=\mathbb{E}[z^{(4Q-2)k}]\mathbb{E}[\boldsymbol{T}(\bar{\boldsymbol{w}}^{\otimes k})^2]$$

holds. On the other hand, let $\boldsymbol{x} \sim \mathbb{S}^{r-1}$ and $z' \sim \chi(r)$. Then we can decompose $\bar{\boldsymbol{u}}$ as $\bar{\boldsymbol{u}} = (z')^{2Q-1}\boldsymbol{x}$. Thus

$$\mathbb{E}[\boldsymbol{T}(\bar{\boldsymbol{u}}^{\otimes k})^2] = \mathbb{E}[(z')^{(4Q-2)k}]\mathbb{E}[\boldsymbol{T}(\boldsymbol{x}^{\otimes k})^2]$$

holds. It implies that

$$\mathbb{E}[\boldsymbol{T}(\bar{\boldsymbol{w}}^{\otimes k})^2] = \frac{\mathbb{E}[(z')^{(4Q-2)k}]}{\mathbb{E}[z^{(4Q-2)k}]}\mathbb{E}[\boldsymbol{T}(\boldsymbol{x}^{\otimes k})^2].$$

Similarly,

$$\mathbb{E}[\|\boldsymbol{T}(\bar{\boldsymbol{w}}^{\otimes k-i})\|_F^2] = \frac{\mathbb{E}[(z')^{(4Q-2)(k-i)}]}{\mathbb{E}[z^{(4Q-2)(k-i)}]}\mathbb{E}[\|\boldsymbol{T}(\boldsymbol{x}^{\otimes(k-i)})\|_F^2]$$

is satisfied. Now from Corollary 13 in [DLS22],

$$\mathbb{E}[\|\boldsymbol{T}(\boldsymbol{x}^{\otimes(k-i)})\|_F^2] \lesssim r^i \mathbb{E}[\boldsymbol{T}(\boldsymbol{x}^{\otimes k})^2]$$

holds. Therefore we obtain

$$\begin{aligned}
\mathbb{E}[\boldsymbol{T}(\bar{\boldsymbol{w}}^{\otimes k})^2] &= \frac{\mathbb{E}[(z')^{(4Q-2)k}]}{\mathbb{E}[z^{(4Q-2)k}]}\mathbb{E}[\boldsymbol{T}(\boldsymbol{x}^{\otimes k})^2]\\
&\gtrsim r^{-i}\frac{\mathbb{E}[(z')^{(4Q-2)k}]}{\mathbb{E}[z^{(4Q-2)k}]}\mathbb{E}[\|\boldsymbol{T}(\boldsymbol{x}^{\otimes(k-i)})\|_F^2]\\
&= r^{-i}\frac{\mathbb{E}[(z')^{(4Q-2)k}]}{\mathbb{E}[z^{(4Q-2)k}]}\frac{\mathbb{E}[z^{(4Q-2)(k-i)}]}{\mathbb{E}[(z')^{(4Q-2)(k-i)}]}\mathbb{E}[\|\boldsymbol{T}(\bar{\boldsymbol{w}}^{\otimes k-i})\|_F^2]\\
&= \frac{r^{(2Q-2)i}}{d^{(2Q-1)i}}\mathbb{E}[\|\boldsymbol{T}(\bar{\boldsymbol{w}}^{\otimes k-i})\|_F^2].
\end{aligned}$$

$\square$

**Corollary 38.** *Let $\boldsymbol{T}$ be a $k \leq P$-symmetric $r$-dimensional tensor and let $\boldsymbol{m}(\boldsymbol{w}, b)_{1:r} = \frac{2a_Q(b)^2}{Q!(Q-1)!}\frac{\mathbb{E}_{c_Q}[c_Q^2](2Q-1)!!}{\mathbb{E}_{z\sim\chi_r}[z^{2Q}]}\|\boldsymbol{w}_{1:r}\|^{2Q-2}\boldsymbol{w}_{1:r}$. Moreover, assume that (B.5) holds. Then,*

$$\mathbb{E}_{\boldsymbol{w}\sim\mathrm{Unif}(\mathbb{S}^{d-1})}[\langle\boldsymbol{T}, \boldsymbol{m}(\boldsymbol{w}, b)_{1:r}^{\otimes k}\rangle^2] \gtrsim \frac{1}{d^{(2Q-1)i}r^{2i}}\mathbb{E}[\|\boldsymbol{T}(\boldsymbol{m}(\boldsymbol{w}, b)_{1:r}^{\otimes k-i})\|_F^2]$$

*holds for all $0 \leq i \leq k$.*

**Proof.** This is obtained by calibrating the coefficients from the previous lemma. $\square$

**Lemma 39.** *Let $\boldsymbol{T}$ be a $0 \leq k \leq P$-symmetric $r$-dimensional tensor. Assume that (B.5) holds. Furthermore, suppose $r \lesssim d^{\frac{1}{2}}$, $T_1 = \tilde{\Omega}(d^{Q+1}r^Q)$ and $N_1 T_1 = \tilde{\Omega}(d^{2Q+1}r)$. Then,*

$$\mathbb{E}_{\boldsymbol{w}\sim\mathrm{Unif}(\mathbb{S}^{d-1})}[\langle\boldsymbol{T}, \boldsymbol{g}_{T_1,N_1}(\boldsymbol{w}, b)_{1:r}^{\otimes k}\rangle^2] \gtrsim \frac{1}{d^{(2Q-1)k}r^{2k}}\|\boldsymbol{T}\|_F^2$$

*holds for all $0 \leq k \leq P$. Therefore, $\boldsymbol{g}_{T_1,N_1}(\boldsymbol{w}, b)$ has $d^{-2Q+1}r^{-2}$-alignment with $\mathcal{S}$.*

**Proof.** Decompose $\boldsymbol{g}_{T_1,N_1}(\boldsymbol{w}, b) = \boldsymbol{m}(\boldsymbol{w}, b) + \boldsymbol{r}(\boldsymbol{w}, b)$ as defined in Corollary 32. From the proof of Lemma 11 in [DLS22], we can show that

$$\begin{aligned}
&\mathbb{E}_{\boldsymbol{w}\sim\mathrm{Unif}(\mathbb{S}^{d-1})}[\langle\boldsymbol{T}, \boldsymbol{g}_{T_1,N_1}(\boldsymbol{w}, b)_{1:r}^{\otimes k}\rangle^2]\\
&\geq \mathbb{E}_{\boldsymbol{w}\sim\mathrm{Unif}(\mathbb{S}^{d-1})}\left[\frac{1}{2}\langle\boldsymbol{T}, \boldsymbol{m}(\boldsymbol{w}, b)_{1:r}^{\otimes k}\rangle^2\right] - \mathbb{E}_{\boldsymbol{w}\sim\mathrm{Unif}(\mathbb{S}^{d-1})}[\delta(\boldsymbol{w}, b)^2] \quad\text{(C.1)}
\end{aligned}$$

for some $\delta(\boldsymbol{w}, b)$ satisfying

$$\begin{aligned}
&\mathbb{E}_{\boldsymbol{w}\sim\mathrm{Unif}(\mathbb{S}^{d-1})}[\delta(\boldsymbol{w}, b)^2]\\
&\lesssim \sum_{i=1}^{k}\mathbb{E}_{\boldsymbol{w}\sim\mathrm{Unif}(\mathbb{S}^{d-1})}[\|\boldsymbol{T}(\boldsymbol{m}(\boldsymbol{w}, b)_{1:r}^{\otimes k-i})\|_F^2]\sqrt{\mathbb{E}_{\boldsymbol{w}\sim\mathrm{Unif}(\mathbb{S}^{d-1})}[\|\boldsymbol{r}(\boldsymbol{w}, b)_{1:r}\|^{4i}]}.
\end{aligned}$$

Here, from Corollaries 32 and 38,

$$\mathbb{E}_{\boldsymbol{w}\sim\mathrm{Unif}(\mathbb{S}^{d-1})}[\delta(\boldsymbol{w},b)^2]$$

$$\lesssim \sum_{i=1}^k d^{(2Q-1)i}r^{2i}\mathbb{E}_{\boldsymbol{w}\sim\mathrm{Unif}(\mathbb{S}^{d-1})}[\langle\boldsymbol{T},\boldsymbol{m}(\boldsymbol{w},b)_{1:r}^{\otimes k}\rangle^2]\tilde{O}_{d,r}\left(\left(\frac{r^{Q-1}}{d^{Q-1}}\frac{r}{T_1}+\frac{r^2}{N_1 T_1}+\frac{r}{d^{2Q+1}}\right)^i\right)$$

holds. Now, from the condition of $N_1$ and $T_1$, and $r\lesssim\sqrt{d}$, we know

$$d^{(2Q-1)}r^2\left(\frac{r^{Q-1}}{d^{Q-1}}\frac{r}{T_1}+\frac{r^2}{N_1 T_1}+\frac{r}{d^{2Q+1}}\right)\lesssim 1.$$

Therefore we can achieve $\mathbb{E}_{\boldsymbol{w}\sim\mathrm{Unif}(\mathbb{S}^{d-1})}[\delta(\boldsymbol{w},b)^2]\leq\frac{1}{4}\mathbb{E}_{\boldsymbol{w}\sim\mathrm{Unif}(\mathbb{S}^{d-1})}[\langle\boldsymbol{T},\boldsymbol{m}(\boldsymbol{w},b)_{1:r}^{\otimes k}\rangle^2]$. Applying this and Corollary 38 ($i=k$) to (C.1) yields the result. $\qquad\square$

### C.1.2  Approximation Power of Infinite-width Neural Network

No we can show that some "infinite-width" neural network can approximate the functions $h\in\mathcal{H}$.

**Lemma 40** (Lemma 9 in [DLS22], adapted). *Let $a\sim\mathrm{Unif}(\{-1,1\})$ and $b\sim\mathrm{Unif}([-\log d,\log d])$. Then, for any $k\geq 0$ there exists $v_k(a,b)$ such that for $|x|\leq\log d$,*

$$\mathbb{E}_{a,b}[v_k(a,b)\sigma(ax+b)]=x^k,\sup_{a,b}|v_k(a,b)|=\tilde{O}_d(1).$$

**Lemma 41** (Lemma 12 and Corollary 5 in [DLS22], adapted). *There exists $\phi_n(a,\boldsymbol{w},b_1,b_2)$ such that if*

$$f_{\phi_n}(\boldsymbol{x}):=\mathbb{E}_{a,\boldsymbol{w},b_1,b_2}[\phi_n(a,\boldsymbol{w},b_1,b_2)\sigma(2\eta_1 a\boldsymbol{g}_{T_1,N_1}(\boldsymbol{w},b_1)+b_2)]$$

*where*

$$a\sim\mathrm{Unif}\{\pm\gamma\},\boldsymbol{w}\sim\mathrm{Unif}(\mathbb{S}^{d-1}),b_1\sim\mathrm{Unif}([-1,1]),b_2\sim\mathrm{Unif}([-\log d,\log d]),$$

*it holds that*

$$\sup_{i\in[N_2],T_1+1\leq t\leq T_2}\left(f_{\phi_n}(\boldsymbol{x}_i^t)-h_n(\boldsymbol{x}_i^t)\right)^2\lesssim\frac{1}{N_2},$$

$$\sup_{T_1+1\leq t\leq T_2}\left(f_{\phi_n}(\boldsymbol{x}^t)-h_n(\boldsymbol{x}^t)\right)^2\lesssim\frac{1}{N_2},$$

$$\mathbb{E}_{a,\boldsymbol{w},b_1,b_2}[\phi_n(a,\boldsymbol{w},b_1,b_2)^2]=\tilde{O}(r^P),$$

$$\sup_{a,\boldsymbol{w},b_1,b_2}|\phi_n(a,\boldsymbol{w},b_1,b_2)|=\tilde{O}(r^P)$$

*for each $n\in[B_P]$.*

**Proof.** We first construct $\phi_{\boldsymbol{T}_k^n}(a,\boldsymbol{w},b_1,b_2)$ for basis tensors defined in Lemma 34, such that if

$$f_{\phi_{\boldsymbol{T}_k^n}}(\boldsymbol{x}):=\mathbb{E}_{a,\boldsymbol{w},b_1,b_2}[\phi_{\boldsymbol{T}_k^n}(a,\boldsymbol{w},b_1,b_2)\sigma(\langle 2\eta_1 a\boldsymbol{g}_{T_1,N_1}(\boldsymbol{w},b_1),\boldsymbol{x}\rangle+b_2)],$$

it holds that

$$\sup_{i\in[N_2],T_1+1\leq t\leq T_2}\left(f_{\phi_{\boldsymbol{T}_k^n}}(\boldsymbol{x}_i^t)-\langle\boldsymbol{T}_k^n,(\boldsymbol{x}_i^t)_{1:r}^{\otimes k}\rangle\right)^2\lesssim\frac{1}{N_2},\qquad(\text{C.2})$$

$$\sup_{T_1+1\leq t\leq T_2}\left(f_{\phi_{\boldsymbol{T}_k^n}}(\boldsymbol{x}^t)-\langle\boldsymbol{T}_k^n,(\boldsymbol{x}^t)_{1:r}^{\otimes k}\rangle\right)^2\lesssim\frac{1}{N_2}.\qquad(\text{C.3})$$

Let $\phi_{\boldsymbol{T}_k^n}(a,\boldsymbol{w},b_1,b_2)=\frac{2v_k(a,b_2)\psi_{\boldsymbol{T}_k^n}(\boldsymbol{w},b_1)}{(2\eta_1)^k\gamma^k}\mathbb{I}_{A(\boldsymbol{w})}\mathbb{I}_{B(b_1)}$ where $v_k$ and $\psi_{\boldsymbol{T}_k^n}$ are constructed in Lemma 36 and the events $A(\boldsymbol{w})$ and $B(b_1)$ are defined as

$$A(\boldsymbol{w}):=\left\{\sup_{b_1\in[-1,1]}2\eta_1\gamma|\langle\boldsymbol{g}_{T_1,N_1}(\boldsymbol{w},b_1),\boldsymbol{x}_i^t\rangle|,\sup_{b_1\in[-1,1]}2\eta_1\gamma|\langle\boldsymbol{g}_{T_1,N_1}(\boldsymbol{w},b_1),\boldsymbol{x}^t\rangle|\leq\log d\right.$$

$$\text{for } i \in [N_2], T_1 + 1 \le t \le T_2 \Big\}$$

$$\cap \left\{ \sup_{b_1 \in [-1,1]} \|2\eta_1 \gamma \boldsymbol{g}_{T_1,N_1}(\boldsymbol{w}, b_1)\| \le 1 \right\}$$

$$B(b_1) := \left\{ \left| \mathbb{E}_{z \sim \mathcal{N}(0,1)}[\sigma_{b_1}(z)\mathrm{He}_i(z)] \right| \le C_H \ (2 \le i \le P+2) \right\}$$
$$\cap \left\{ \left| \mathbb{E}_{z \sim \mathcal{N}(0,1)}[\sigma_{b_1}(z)\mathrm{He}_Q(z)] \right| \ge C_L \right\}.$$

Then,

$$f_{\phi_{\boldsymbol{T}_k^n}}(\boldsymbol{x})$$

$$= \mathbb{E}_{a,\boldsymbol{w},b_1,b_2}\left[ \frac{2v_k(a,b_2)\psi_{\boldsymbol{T}_k^n}(\boldsymbol{w},b_1)}{(2\eta_1)^k \gamma^k} \mathbb{I}_{A(\boldsymbol{w})}\mathbb{I}_{B(b_1)}\sigma(\langle 2\eta_1 a\boldsymbol{g}_{T_1,N_1}(\boldsymbol{w},b_1),\boldsymbol{x}\rangle + b_2) \right]$$

$$= \mathbb{E}_{b_1}\left[ 2 \cdot \mathbb{I}_{B(b_1)}\mathbb{E}_{\boldsymbol{w}}\left[ \frac{\psi_{\boldsymbol{T}_k^n}(\boldsymbol{w},b_1)}{(2\eta_1)^k \gamma^k} \mathbb{I}_{A(\boldsymbol{w})}\mathbb{E}_{a',b_2}[v_k(a,b_2)\sigma(\langle 2\eta_1 a'\gamma\boldsymbol{g}_{T_1,N_1}(\boldsymbol{w},b_1),\boldsymbol{x}\rangle + b_2)] \right] \right],$$

where $a' \sim \mathrm{Unif}(\{\pm 1\})$. Here, if $\boldsymbol{x} \in \bigcup_{t=T_1+1}^{T_2}\{\boldsymbol{x}_1^t, \ldots, \boldsymbol{x}_{N_2}^t, \boldsymbol{x}^t\}$,

$$\mathbb{E}_{\boldsymbol{w}}\left[ \frac{\psi_{\boldsymbol{T}_k^n}(\boldsymbol{w},b_1)}{(2\eta_1)^k \gamma^k} \mathbb{I}_{A(\boldsymbol{w})}\mathbb{E}_{a'\sim\mathrm{Unif}(\{\pm 1\}),b_2}[v_k(a,b_2)\sigma(\langle 2\eta_1 a'\gamma\boldsymbol{g}_{T_1,N_1}(\boldsymbol{w},b_1),\boldsymbol{x}\rangle + b_2)] \right]$$

$$= \mathbb{E}_{\boldsymbol{w}}\left[ \psi_{\boldsymbol{T}_k^n}(\boldsymbol{w},b_1)\langle \boldsymbol{g}_{T_1,N_1}(\boldsymbol{w},b_1),\boldsymbol{x}\rangle^k \right] + \mathbb{E}_{\boldsymbol{w}}\left[ \psi_{\boldsymbol{T}_k^n}(\boldsymbol{w},b_1)(1 - \mathbb{I}_{A(\boldsymbol{w})})\langle \boldsymbol{g}_{T_1,N_1}(\boldsymbol{w},b_1),\boldsymbol{x}\rangle^k \right].$$

Note that the absolute value of second term above can be upper bounded by $\mathbb{E}_{\boldsymbol{w}}\left[ (1 - \mathbb{I}_{A(\boldsymbol{w})}) \right]^{1/2}\mathbb{E}_{\boldsymbol{w}}\left[ (\psi_{\boldsymbol{T}_k^n}(\boldsymbol{w},b_1)\langle \boldsymbol{g}_{T_1,N_1}(\boldsymbol{w},b_1),\boldsymbol{x}\rangle^k)^2 \right]^{1/2}$. Recall that $A(\boldsymbol{w})$ occurs with high probability from Corollary 32, and $\mathbb{E}_{\boldsymbol{w}}\left[ (\psi_{\boldsymbol{T}_k^n}(\boldsymbol{w},b_1)\langle \boldsymbol{g}_{T_1,N_1}(\boldsymbol{w},b_1),\boldsymbol{x}\rangle^k)^2 \right]^{1/2}$ is polynomial in $d$. Hence we can set the second term to be $O(1/\sqrt{N_2})$.

Moreover, as $\mathbb{P}[B(b_1)] = 1/2$ from Lemma 20, we have

$$\mathbb{E}_{b_1}\left[ 2 \cdot \mathbb{I}_{B(b_1)}\mathbb{E}_{\boldsymbol{w}}\left[ \psi_{\boldsymbol{T}_k^n}(\boldsymbol{w},b_1)\langle \boldsymbol{g}_{T_1,N_1}(\boldsymbol{w},b_1),\boldsymbol{x}\rangle^k \right] \right]$$

$$= \mathbb{E}_{b_1}\left[ 2 \cdot \langle \boldsymbol{T}_k^n, \boldsymbol{x}^{\otimes k}\rangle \right] + \mathbb{E}_{b_1}\left[ 2(1 - \mathbb{I}_{B(b_1)})\langle \boldsymbol{T}_k^n, \boldsymbol{x}^{\otimes k}\rangle \right] = \langle \boldsymbol{T}_k^n, \boldsymbol{x}^{\otimes k}\rangle.$$

Therefore, (C.2) and (C.3) holds. Noting that $\eta_1 \gamma \asymp \sqrt{rd^{2Q-1}}/C_1(\log d)^{C_2}$, We obtain

$$\mathbb{E}_{a,\boldsymbol{w},b_1,b_2}[\phi_{\boldsymbol{T}_k^n}(a,\boldsymbol{w},b_1,b_2)^2]$$

$$\le \frac{4}{(2\eta_1)^{2k}\gamma^{2k}}\mathbb{E}_{a,b_2}[v_k(a,b_2)^2]\mathbb{E}_{\boldsymbol{w},b_1}[\mathbb{I}_{B(b_1)}\psi_{\boldsymbol{T}_k^n}(\boldsymbol{w},b_1)^2]$$

$$\lesssim \frac{4}{(2\eta_1)^{2k}\gamma^{2k}}\mathbb{E}_{a,b_2}[v_k(a,b_2)^2](d^{(2Q-1)}r^2)^k\|\boldsymbol{T}_k^n\|_F^2$$

$$= \tilde{O}\left( r^k r^{\frac{P-k}{2}} \right)$$

and

$$\sup_{a,\boldsymbol{w},b_1,b_2} |\phi_{\boldsymbol{T}_k^n}(a,\boldsymbol{w},b_1,b_2)|$$

$$\le \frac{2}{(2\eta_1)^k\gamma^k}|v_k(a,b_2)|\mathbb{I}_{A(\boldsymbol{w})}\mathbb{I}_{B(b_1)}|\psi_{\boldsymbol{T}_k^n}(\boldsymbol{w},b_1)|$$

$$\lesssim \frac{2}{(2\eta_1)^{2k}\gamma^{2k}}(d^{(2Q-1)}r^2)^k\|\boldsymbol{T}_k^n\|_F\|(2\eta_1\gamma)\boldsymbol{g}_{T_1,N_1}(\boldsymbol{w},b_1)\|^k\mathbb{I}_{A(\boldsymbol{w})}\mathbb{I}_{B(b_1)}$$

$$\lesssim \tilde{O}\left( r^k r^{\frac{P-k}{4}} \right)$$

from Lemmas 36 and 39. To show the original claim, it suffices to set $\phi_n(a,\boldsymbol{w},b_1,b_2) = \sum_{k=0}^{P}\phi_{\boldsymbol{T}_k^n}(a,\boldsymbol{w},b_1,b_2)$, which yields the upper bounds on the magnitude of $\phi_n$ because $\sum_{k=0}^{P}r^k r^{\frac{P-k}{2}} \lesssim r^P$ and $\sum_{k=0}^{P}r^k r^{\frac{P-k}{4}} \lesssim r^P$. $\qquad \square$

### C.1.3  Proof of Proposition 33

We discretize Lemma 41 to establish Proposition 33.

**Proof.** [Proof of Proposition 33] First, in the same discretization manner as the proof of Lemma 13 in [DLS22], we can show that there exists $\boldsymbol{a}^1, \ldots, \boldsymbol{a}^{B_P}$ such that for each $n \in [B_P]$,

$$
\sup_{i \in [N_2], T_1+1 \leq t \leq T_2} \left( \sum_{j=1}^m a_j^n \sigma\left( \left\langle 2\eta_1 \Gamma_{j,j}^{(0)} \boldsymbol{g}_{T_1,N_1}(\boldsymbol{w}_j^{(0)}, b_j^{(0)}), \boldsymbol{x}_i^t \right\rangle + b_j \right) - h_n(\boldsymbol{x}_i^t) \right)^2
$$

$$
= \tilde{O}\left( \frac{r^P}{m} + \frac{1}{N_2} \right),
$$

$$
\sup_{T_1+1 \leq t \leq T_2} \left( \sum_{j=1}^m a_j^n \sigma\left( \left\langle 2\eta_1 \Gamma_{j,j}^{(0)} \boldsymbol{g}_{T_1,N_1}(\boldsymbol{w}_j^{(0)}, b_j^{(0)}), \boldsymbol{x}^t \right\rangle + b_j \right) - h_n(\boldsymbol{x}^t) \right)^2
$$

$$
= \tilde{O}\left( \frac{r^P}{m} + \frac{1}{N_2} \right)
$$

and $\sup_{n \in [B_P]} \|\boldsymbol{a}^n\|^2 = \tilde{O}\left( \frac{r^P}{m} \right)$ with high probability ($b_j^{(0)}$, $b_j$ denotes the biases at the first and second initialization, respectively). Recall that $\boldsymbol{w}_j^{(1)} = 2\eta_1 \Gamma_{j,j}^{(0)} \boldsymbol{g}_{T_1,N_1}(\boldsymbol{w}_j^{(0)}, b_j^{(0)}) - 2\eta_1 \frac{1}{T_1} \sum_{t=1}^{T_1} f(\boldsymbol{X}^t, \boldsymbol{y}^t, \boldsymbol{x}^t; \boldsymbol{W}^{(0)}, \boldsymbol{\Gamma}^{(0)}, \boldsymbol{b}^{(0)}) \nabla_{\boldsymbol{w}_j^{(0)}} f(\boldsymbol{X}^t, \boldsymbol{y}^t, \boldsymbol{x}^t; \boldsymbol{W}^{(0)}, \boldsymbol{\Gamma}^{(0)}, \boldsymbol{b}^{(0)})$. It remains to bound the effect of the second term. By Lemma 18, we know that the norm of second term is $\tilde{O}\left( \eta_1 \gamma^2 m \sqrt{d} \right)$. Similar to the argument in the proof of Lemma 18, the inner product between the second term and $\boldsymbol{x}^t, \boldsymbol{x}_i^t$ ($i \in [N_2], T_1 + 1 \leq t \leq T_2$) is also $\tilde{O}\left( \eta_1 \gamma^2 m \sqrt{d} \right)$, with high probability. From the Cauchy-Schwarz inequality and the Lipschitz continuity of $\sigma$,

$$
\left| \sum_{j=1}^m a_j^n \sigma\left( \left\langle 2\eta_1 \Gamma_{j,j}^{(0)} \boldsymbol{g}_{T_1,N_1}(\boldsymbol{w}_j^{(0)}, b_j^{(0)}), \boldsymbol{x} \right\rangle + b_j \right) - \sum_{j=1}^m a_j^n \sigma\left( \left\langle \boldsymbol{w}_j^{(1)}, \boldsymbol{x} \right\rangle + b_j \right) \right|
$$

$$
= \tilde{O}\left( \sqrt{\frac{r^P}{m}} \eta_1 \gamma^2 m \sqrt{d} \sqrt{m} \right) = \tilde{O}\left( \sqrt{\frac{r^P}{m}} \right).
$$

We obtain the assertion. $\qquad\square$

### C.2  Concentration of Correlation

Next, we evaluate the discrepancy between the empirical correlation $\frac{1}{N_2} \sum_{j=1}^{N_2} y_j h(\boldsymbol{x}_j)$ and its expectation for each $h \in \mathcal{H}$, which decides the approximation error in (b) of (3.1).

Recall that $h \in \mathcal{H}$ can be written in the form as

$$
h(\boldsymbol{x}) = \prod_{i=1}^r \frac{1}{\sqrt{p_i!}} \mathrm{He}_{p_i}(x_i) \ (Q \leq p_1 + \cdots + p_r \leq P, p_1 \geq 0, \ldots, p_r \geq 0).
$$

Also note that

$$
\frac{1}{N_2} \sum_{j=1}^{N_2} y_j h(\boldsymbol{x}_j) - \mathbb{E}_{\boldsymbol{x}, \varsigma}[y h(\boldsymbol{x})]
$$

$$
= \frac{1}{N_2} \sum_{j=1}^{N_2} \varsigma_j h(\boldsymbol{x}_j) + \frac{1}{N_2} \sum_{j=1}^{N_2} \sigma_*(\langle \boldsymbol{x}_j, \boldsymbol{\beta} \rangle) h(\boldsymbol{x}_j) - \mathbb{E}_{\boldsymbol{x}}[\sigma_*(\langle \boldsymbol{x}, \boldsymbol{\beta} \rangle) h(\boldsymbol{x})],
$$

where $\sigma_*(z) = \sum_{i=Q}^P \frac{c_i}{i!} \mathrm{He}_i(z)$. First we give an upper bound for $\left| \frac{1}{N_2} \sum_{j=1}^{N_2} \sigma_*(\langle \boldsymbol{x}_j, \boldsymbol{\beta} \rangle) h(\boldsymbol{x}_j) - \mathbb{E}_{\boldsymbol{x}}[\sigma_*(\langle \boldsymbol{x}, \boldsymbol{\beta} \rangle) h(\boldsymbol{x})] \right|$.

**Lemma 42.** *Let $D \leq 2P$ be the degree of $\sigma_*(\langle \boldsymbol{x}, \boldsymbol{\beta} \rangle)h(\boldsymbol{x})$, when we regard it as an $r$-variable polynomial with respect to $(x_1, \ldots, x_r)$. For all $t > 0$ there exist absolute constants $M$ and $C_D$ depending only on $D$ such that*

$$\mathbb{P}\left( \left| \frac{1}{N_2} \sum_{j=1}^{N_2} \sigma_*(\langle \boldsymbol{x}_j, \boldsymbol{\beta} \rangle)h(\boldsymbol{x}_j) - \mathbb{E}_{\boldsymbol{x}}[\sigma_*(\langle \boldsymbol{x}, \boldsymbol{\beta} \rangle)h(\boldsymbol{x})] \right| \geq t \right)$$

$$\leq 2 \exp\left( -\frac{1}{C_D M^2} \min_{1 \leq k \leq D} \left( \frac{t\sqrt{N_2}}{\|\mathbb{E}[\nabla_{\boldsymbol{x}}^k(\sigma_*(\langle \boldsymbol{x}, \boldsymbol{\beta} \rangle)h(\boldsymbol{x}))]\|_F} \right)^{2/k} \right)$$

*holds, where $C_D$ and $M$ are constants.*

**Proof.** Let $F(\boldsymbol{x}_1, \ldots, \boldsymbol{x}_N) = \frac{1}{N_2} \sum_{j=1}^{N_2} \sigma_*(\langle \boldsymbol{x}_j, \boldsymbol{\beta} \rangle)h(\boldsymbol{x}_j)$. It is a degree-$D$ polynomial for $x_{1,1}, \ldots, x_{1,r}, \ldots, x_{N_2,r}$, which are standard Gaussian variables. Also,

$$\|\mathbb{E}[\nabla^k F(\boldsymbol{x}_1, \ldots, \boldsymbol{x}_N)]\|_F = \sqrt{N_2}^{-1} \|\mathbb{E}[\nabla^k(\sigma_*(\langle \boldsymbol{x}, \boldsymbol{\beta} \rangle)h(\boldsymbol{x}))]\|_F$$

is satisfied. Then, applying Lemma 15 to $F$ yields the desired result. $\qquad\square$

Then, our aim is to bound $\|\mathbb{E}[\nabla^k(\sigma_*(\langle \boldsymbol{x}, \boldsymbol{\beta} \rangle)h(\boldsymbol{x}))]\|_F$.

**Lemma 43.** *Let $h(\boldsymbol{x}) = \prod_{i=1}^r \frac{1}{\sqrt{p_i}} \mathrm{He}_{p_i}(x_i)$ and $\tilde{P} = p_1 + \cdots + p_r$. For $Q + \tilde{P} \leq k \leq P + \tilde{P}$,*

$$\|\mathbb{E}[\nabla^k(\sigma_*(\langle \boldsymbol{x}, \boldsymbol{\beta} \rangle)h(\boldsymbol{x}))]\|_F^2 \leq R_c^2 \bar{C}_k$$

*holds where $\bar{C}_k$ is a constant depending only on $k$. For other $k$, $\|\mathbb{E}[\nabla^k(\sigma_*(\langle \boldsymbol{x}, \boldsymbol{\beta} \rangle)h(\boldsymbol{x}))]\|_F = 0$.*

**Proof.** First,

$$\|\mathbb{E}[\nabla^k(\sigma_*(\langle \boldsymbol{x}, \boldsymbol{\beta} \rangle)h(\boldsymbol{x}))]\|_F^2 = \left\| \sum_{i=Q}^{P} \frac{c_i}{i!} \mathbb{E}[\nabla^k(\mathrm{He}_i(\langle \boldsymbol{x}, \boldsymbol{\beta} \rangle)h(\boldsymbol{x}))] \right\|_F^2$$

holds. To calculate $\mathbb{E}[\nabla^k(\mathrm{He}_i(\langle \boldsymbol{x}, \boldsymbol{\beta} \rangle)h(\boldsymbol{x}))]$, note that from Lemma 8,

$$\mathrm{He}_i(\langle \boldsymbol{x}, \boldsymbol{\beta} \rangle)h(\boldsymbol{x})$$
$$= \left( \sum_{\substack{q_1 + \cdots + q_r = i \\ q_j \geq 0}} \frac{i!}{q_1! \cdots q_r!} \beta_1^{q_1} \cdots \beta_r^{q_r} \mathrm{He}_{q_1}(x_1) \cdots \mathrm{He}_{q_r}(x_r) \right) \frac{\mathrm{He}_{p_1}(x_1)}{\sqrt{p_1!}} \cdots \frac{\mathrm{He}_{p_r}(x_r)}{\sqrt{p_r!}}. \quad \text{(C.4)}$$

Also, from the discussion in Appendix A.3, $\mathbb{E}_{\boldsymbol{x}}\left[ \frac{\partial^{s_1}}{\partial x_1^{s_1}} \cdots \frac{\partial^{s_r}}{\partial z_r^{s_r}} \mathrm{He}_i(\langle \boldsymbol{x}, \boldsymbol{\beta} \rangle)h(\boldsymbol{x}) \right]$ is equal to the coefficient of (C.4) with respect to $\frac{1}{s_1! \cdots s_r!} \mathrm{He}_{s_1}(x_1) \cdots \mathrm{He}_{s_r}(x_r)$. Therefore, $\mathbb{E}[\nabla^k(\mathrm{He}_i(\langle \boldsymbol{x}, \boldsymbol{\beta} \rangle)h(\boldsymbol{x}))] \neq \boldsymbol{O}$ if and only if $i + \tilde{P} = k$. Then, $\left\| \sum_{i=Q}^{P} \frac{c_i}{i!} \mathbb{E}[\nabla^k(\mathrm{He}_i(\langle \boldsymbol{x}, \boldsymbol{\beta} \rangle)h(\boldsymbol{x}))] \right\|_F^2 = \left\| \frac{c_{k-\tilde{P}}}{(k-\tilde{P})!} \mathbb{E}[\nabla^k(\mathrm{He}_{k-\tilde{P}}(\langle \boldsymbol{x}, \boldsymbol{\beta} \rangle)h(\boldsymbol{x}))] \right\|_F^2$.

Here, we can observe that the $(j_1, \ldots, j_k)$-th entry of the tensor $\frac{c_{k-\tilde{P}}}{(k-\tilde{P})!} \mathbb{E}[\nabla^k(\mathrm{He}_{k-\tilde{P}}(\langle \boldsymbol{x}, \boldsymbol{\beta} \rangle)h(\boldsymbol{x}))]$ is equal to

$$c_{k-\tilde{P}} \frac{s_1! \cdots s_r!}{(s_1 - p_1)! \cdots (s_r - p_r)!} \cdot \beta_1^{s_1 - p_1} \cdots \beta_r^{s_r - p_r} \frac{1}{\sqrt{p_1!} \cdots \sqrt{p_r!}} \mathbb{I}_{s_1 - p_1 \geq 0, \ldots, s_r - p_r \geq 0}$$

where $s_i = \sum_{k'=1}^k \mathbb{I}_{j_{k'} = i}$ $(i \in [r])$. There are $\frac{k!}{s_1! \cdots s_r!}$ index sets $(j_1, \ldots, j_k)$ having the same entry, and thus

$$\left\| \frac{c_{k-\tilde{P}}}{(k-\tilde{P})!} \mathbb{E}[\nabla^k(\mathrm{He}_{k-\tilde{P}}(\langle \boldsymbol{x}, \boldsymbol{\beta} \rangle)h(\boldsymbol{x}))] \right\|_F^2$$
$$= \sum_{s_1 + \cdots + s_r = k} c_{k-\tilde{P}}^2 \frac{1}{p_1! \cdots p_r!} \frac{k!}{s_1! \cdots s_r!} \left( \frac{s_1! \cdots s_r!}{(s_1 - p_1)! \cdots (s_r - p_r)!} \right)^2$$

$$\cdot (\beta_1^{s_1-p_1}\cdots\beta_r^{s_r-p_r})^2 \mathbb{I}_{s_1-p_1\geq 0,\ldots,s_r-p_r\geq 0}$$

$$= \sum_{q_1+\cdots+q_r=k-\tilde{P},q_j\geq 0} c_{k-\tilde{P}}^2 \frac{1}{p_1!\cdots p_r!} \frac{k!}{(q_1+p_1)!\cdots(q_r+p_r)!}\left(\frac{(q_1+p_1)!\cdots(q_r+p_r)!}{q_1!\cdots q_r!}\right)^2$$

$$\cdot (\beta_1^{q_1}\cdots\beta_r^{q_r})^2$$

$$= \sum_{q_1+\cdots+q_r=k-\tilde{P},q_j\geq 0} c_{k-\tilde{P}}^2 \frac{(k-\tilde{P})!}{q_1!\cdots q_r!} \frac{k!}{(k-\tilde{P})!}\left(\frac{(q_1+p_1)!\cdots(q_r+p_r)!}{p_1!\cdots p_r!q_1!\cdots q_r!}\right)$$

$$\cdot (\beta_1^{q_1}\cdots\beta_r^{q_r})^2. \tag{C.5}$$

We can verify that there exists a constant $\bar{C}_k$ depending only on $P$ such that $\frac{k!}{(k-\tilde{P})!}\left(\frac{(q_1+p_1)!\cdots(q_r+p_r)!}{p_1!\cdots p_r!q_1!\cdots q_r!}\right) \leq \bar{C}_k$ holds uniformly; for example, one can naively obtain

$$\frac{k!}{(k-\tilde{P})!}\left(\frac{(q_1+p_1)!\cdots(q_r+p_r)!}{p_1!\cdots p_r!q_1!\cdots q_r!}\right) \leq k!(q_1+p_1)!\cdots(q_r+p_r)! \leq k!k^k.$$

Here we used $(q_1+p_1)+\cdots+(q_r+p_r)=k$.

Moreover, $\sum_{q_1+\cdots+q_r=k-\tilde{P},q_i\geq 0}\frac{(k-\tilde{P})!}{q_1!\cdots q_r!}(\beta_1^{q_1}\cdots\beta_r^{q_r})^2 = \|\boldsymbol{\beta}^{\otimes k-\tilde{P}}\|_F^2 = 1$ holds. Hence (C.5) is upper bounded by $c_{k-\tilde{P}}^2\bar{C}_k \leq R_c^2\bar{C}_k$. $\qquad\square$

**Corollary 44.** *With high probability,*

$$\left|\frac{1}{N_2}\sum_{j=1}^{N_2} y_j h(\boldsymbol{x}_j) - \mathbb{E}_{\boldsymbol{x}}[\sigma_*(\langle\boldsymbol{x},\boldsymbol{\beta}\rangle)h(\boldsymbol{x})]\right| = \tilde{O}\left(\frac{1}{\sqrt{N_2}}\right)$$

*holds.*

**Proof.** Plugging the results above and $t = \Theta(\frac{1}{\sqrt{N_2}}(\log d)^P)$ into Lemma 42 yields

$$\left|\frac{1}{N_2}\sum_{j=1}^{N_2}\sigma_*(\langle\boldsymbol{x}_j,\boldsymbol{\beta}\rangle)h(\boldsymbol{x}_j) - \mathbb{E}_{\boldsymbol{x}}[\sigma_*(\langle\boldsymbol{x},\boldsymbol{\beta}\rangle)h(\boldsymbol{x})]\right| \lesssim \frac{1}{\sqrt{N_2}}(\log d)^P$$

with high probability. Moreover, we can easily verify $\left|\frac{1}{N_2}\sum_{j=1}^{N_2}\varsigma_j h(\boldsymbol{x}_j)\right| = \tilde{O}\left(\frac{1}{\sqrt{N_2}}\right)$ by the method used in the proof of Lemma 26. $\qquad\square$

### C.3 Proof of Proposition 2

We prove the following Proposition 2 using the preparations above.

**Proposition 2.** *Let $\boldsymbol{X}^t = (\boldsymbol{x}_1^t,\ldots,\boldsymbol{x}_{N_2}^t)$ and $\boldsymbol{y}^t = (y_1^t,\ldots,y_{N_2}^t)^\top$ where $T_1+1\leq t\leq T_1+T_2$. With high probability over the data distribution and random initialization of the parameters, there exists $\bar{\boldsymbol{\Gamma}}$ such that*

$$\left|f(\boldsymbol{X}^t,\boldsymbol{y}^t,\boldsymbol{x}^t;\boldsymbol{W}^{(1)},\bar{\boldsymbol{\Gamma}},\boldsymbol{b}) - y^t\right| - \tau = \tilde{O}\left(\sqrt{\frac{r^{3P}}{m} + \frac{r^{2P}}{N_2}}\right)$$

*holds for all $t \in \{T_1+1,\ldots,T_2\}$, where $\boldsymbol{W}^{(1)}$ is obtained by line 4 of Algorithm 1 and $\boldsymbol{b}$ is re-initialized in Line 5 of Algorithm 1, where the conditions on $\eta_1,\lambda_1,m,\gamma,N_1,T_1$ are identical to that in Theorem 1. Moreover, $\|\bar{\boldsymbol{\Gamma}}\|_F = \tilde{O}(r^{2P}/m)$ is satisfied.*

**Proof.** It suffices to show the proposition for each $T_1+1\leq t\leq T_1+T_2$, and then take the union bound. We drop the superscript $t$ for simplicity.

Note that, from Lemma 8 and $\mathbb{E}_{\boldsymbol{x}}[h(\boldsymbol{x})h'(\boldsymbol{x})] = \mathbb{I}_{h=h'}$ for $h,h' \in \mathcal{H}$, we can expand $f_*(\boldsymbol{x})$ as $f_*(\boldsymbol{x}) = \sum_{n=1}^{B_P}\mathbb{E}_{\boldsymbol{x}\sim\mathcal{N}(0,I_d)}[f_*(\boldsymbol{x})h_n(\boldsymbol{x})]h_n(\boldsymbol{x})$. Now let $\boldsymbol{A} = \begin{bmatrix}\boldsymbol{a}^1 & \cdots & \boldsymbol{a}^{B_P}\end{bmatrix} \in \mathbb{R}^{m\times B_P}$, where

$\{a^n\}_{n=1}^{B_P}$ is constructed in Proposition 33. Then, by setting $\bar{\Gamma} = AA^\top$,

$$\left\langle \frac{AA^\top \sigma(W^{(1)}X + b\mathbf{1}_N^\top)y}{N_2}, \begin{bmatrix} \sigma(w_1^{(1)\top} x + b_1) \\ \vdots \\ \sigma(w_m^{(1)\top} x + b_m) \end{bmatrix} \right\rangle - y$$

$$= \sum_{n=1}^{B_P} (h_n(x) + \epsilon_n(x)) \left( \mathbb{E}[f_* h_n] + \left( \frac{1}{N_2} \sum_{j=1}^{N_2} y_j h_n(x_j) - \mathbb{E}[f_* h_n] \right) + \frac{1}{N_2} \sum_{j=1}^{N_2} y_j \epsilon_n(x_j) \right)$$

$$\quad - f_*(x) - \varsigma$$

$$= \sum_{n=1}^{B_P} (h_n(x) + \epsilon_n(x)) \left( \mathbb{E}[f_* h_n] + \left( \frac{1}{N_2} \sum_{j=1}^{N_2} y_j h_n(x_j) - \mathbb{E}[f_* h_n] \right) + \frac{1}{N_2} \sum_{j=1}^{N_2} y_j \epsilon_n(x_j) \right)$$

$$\quad - \sum_{n=1}^{B_P} h_n(x) \mathbb{E}[f_* h_n] - \varsigma$$

holds, where $\epsilon_n(x) := \sum_{j=1}^m a_j^n \sigma(w_j^{(1)\top} x + b_j) - h_n(x)$ satisfits $|\epsilon_n(x)| = \tilde{O}\left( \sqrt{r^P/m + 1/N_2} \right)$ from Proposition 33.

To bound the error, we note that the following holds for each $n$:

- As $h_n(x)$ is a polynomial whose degree is at most $P$, and $\mathbb{E}_x[\nabla^k h_n(x)] = O(1)$, $|h_n(x)| \lesssim (\log d)^{P/2}$ holds with high probability from Lemma 15.
- From lemma 8, $|\mathbb{E}[f_*(x) h_n(x)]| = O(1)$ holds.
- From Corollary 17, $|y_j| \lesssim (\log d)^{P/2}$ holds with high probability.

Hence, for each $n \in [B_P]$, we obtain

$$\left| \left( \underbrace{h_n(x)}_{\tilde{O}(1)} + \underbrace{\epsilon_n(x)}_{\tilde{O}(\sqrt{\frac{r^P}{m} + \frac{1}{N_2}})} \right) \left( \underbrace{\mathbb{E}[f_* h_n]}_{O(1)} + \underbrace{\frac{1}{N_2} \sum_{j=1}^{N_2} y_j h_n(x_j) - \mathbb{E}[f_* h_n]}_{\tilde{O}(\sqrt{\frac{1}{N_2}})} + \underbrace{\frac{1}{N_2} \sum_{j=1}^{N_2} y_j \epsilon_n(x_j)}_{\tilde{O}(\sqrt{\frac{r^P}{m} + \frac{1}{N_2}})} \right) \right.$$

$$\left. - h_n(x) \mathbb{E}[f_* h_n] \right|$$

$$= \tilde{O}\left( \sqrt{\frac{r^P}{m} + \frac{1}{N_2}} + \left( \frac{r^P}{m} + \frac{1}{N_2} \right) \right).$$

Combining this with $B_P = \Theta(r^P)$ and $m = \tilde{\Omega}(r^P)$, we arrive at the upper bound.

Finally for $\|\bar{\Gamma}\|_F$, we have

$$\|\bar{\Gamma}\|_F \le \sum_{n=1}^{B_P} \|a^n (a^n)^\top\|_F = \sum_{n=1}^{B_P} \|a^n\|^2 = \tilde{O}(r^{2P}/m).$$

$\square$

# D  Generalization Error Analysis and Proof of Theorem 1

Note that after one-step gradient descent, $\|w_j^{(1)}\| = \tilde{O}_{d,r}(1)$ is ensured with high probability from Lemma 18 and Corollary 32.

### D.1 Rademacher Complexity Bound

Let $\boldsymbol{W} = \boldsymbol{W}^{(1)}$ and suppose $b_1, \ldots, b_m$ are biases obtained in line 5 of Algorithm 1. Let

$$\mathcal{F}_{N,G} = \left\{ (\boldsymbol{X}, \boldsymbol{y}, \boldsymbol{x}) \mapsto \left\langle \frac{\boldsymbol{\Gamma}\sigma(\boldsymbol{W}^{(1)}\boldsymbol{X} + \boldsymbol{b}\mathbf{1}_N^\top)\boldsymbol{y}}{N}, \begin{bmatrix} \sigma((\boldsymbol{w}_1^{(1)})^\top \boldsymbol{x} + b_1) \\ \vdots \\ \sigma((\boldsymbol{w}_m^{(1)})^\top \boldsymbol{x} + b_m) \end{bmatrix} \right\rangle \Bigg| \|\boldsymbol{\Gamma}\|_F \leq G \right\}$$

be the set of transformers where the norm of the attention matrix is constrained, and let

$$\mathrm{Rad}_T(\mathcal{F}_{N,G}) = \mathbb{E}_{\{\boldsymbol{X}^t\}, \{\boldsymbol{y}^t\}, \{\boldsymbol{x}^t\}, \{\epsilon^t\}} \left[ \sup_{f \in \mathcal{F}_{N,G}} \frac{1}{T} \sum_{t=1}^T \epsilon^t f(\boldsymbol{X}^t, \boldsymbol{y}^t, \boldsymbol{x}^t) \right]$$

be its Rademacher complexity, where $\epsilon^t \sim \mathrm{Unif}(\{\pm 1\})$. Here contexts $\boldsymbol{X} \in \mathbb{R}^{d \times N}$ and $\boldsymbol{y} \in \mathbb{R}^N$ are of length $N$. We can evaluate the Rademacher complexity as follows.

**Proposition 45.**

$$\mathrm{Rad}_T(\mathcal{F}_{N,G}) = \tilde{O}_{d,r}\left(\frac{Gm}{\sqrt{T}}\right)$$

holds, when $|b_j| = \tilde{O}_{d,r}(1)$ and $\|\boldsymbol{w}_j^{(1)}\| = \tilde{O}_{d,r}(1)$ for all $j \in [m]$.

**Proof.** First, it holds that

$$\mathrm{Rad}_T(\mathcal{F}_{N,G})$$

$$= \mathbb{E}_{\{\boldsymbol{X}^t\}, \{\boldsymbol{y}^t\}, \{\boldsymbol{x}^t\}, \{\epsilon^t\}} \left[ \sup_{f \in \mathcal{F}_{N,G}} \frac{1}{T} \sum_{t=1}^T \epsilon^t f(\boldsymbol{X}^t, \boldsymbol{y}^t, \boldsymbol{x}^t) \right]$$

$$= \mathbb{E}\left[ \sup_{\|\boldsymbol{\Gamma}\|_F \leq G} \frac{1}{T} \sum_{t=1}^T \epsilon^t \left\langle \frac{\boldsymbol{\Gamma}\sigma(\boldsymbol{W}^{(1)}\boldsymbol{X}^t + \boldsymbol{b}\mathbf{1}_N^\top)\boldsymbol{y}^t}{N}, \begin{bmatrix} \sigma((\boldsymbol{w}_1^{(1)})^\top \boldsymbol{x}^t + b_1) \\ \vdots \\ \sigma((\boldsymbol{w}_m^{(1)})^\top \boldsymbol{x}^t + b_m) \end{bmatrix} \right\rangle \right]$$

$$\overset{(a)}{\leq} \mathbb{E}\left[ \frac{1}{T} \sup_{\|\boldsymbol{\Gamma}\|_F \leq G} \|\boldsymbol{\Gamma}\|_F \left\| \sum_{t=1}^T \epsilon^t \frac{\sigma(\boldsymbol{W}^{(1)}\boldsymbol{X}^t + \boldsymbol{b}\mathbf{1}_N^\top)\boldsymbol{y}^t}{N} \begin{bmatrix} \sigma((\boldsymbol{w}_1^{(1)})^\top \boldsymbol{x}^t + b_1) \\ \vdots \\ \sigma((\boldsymbol{w}_m^{(1)})^\top \boldsymbol{x}^t + b_m) \end{bmatrix}^\top \right\|_F \right]$$

$$= \frac{G}{T} \mathbb{E}\left[ \left\| \sum_{t=1}^T \epsilon^t \frac{\sigma(\boldsymbol{W}^{(1)}\boldsymbol{X}^t + \boldsymbol{b}\mathbf{1}_N^\top)\boldsymbol{y}^t}{N} \begin{bmatrix} \sigma((\boldsymbol{w}_1^{(1)})^\top \boldsymbol{x}^t + b_1) \\ \vdots \\ \sigma((\boldsymbol{w}_m^{(1)})^\top \boldsymbol{x}^t + b_m) \end{bmatrix}^\top \right\|_F \right]$$

$$\leq \frac{G}{T} \sqrt{\mathbb{E}\left[ \left\| \sum_{t=1}^T \epsilon^t \frac{\sigma(\boldsymbol{W}^{(1)}\boldsymbol{X}^t + \boldsymbol{b}\mathbf{1}_N^\top)\boldsymbol{y}^t}{N} \begin{bmatrix} \sigma((\boldsymbol{w}_1^{(1)})^\top \boldsymbol{x}^t + b_1) \\ \vdots \\ \sigma((\boldsymbol{w}_m^{(1)})^\top \boldsymbol{x}^t + b_m) \end{bmatrix}^\top \right\|_F^2 \right]}$$

$$\overset{(b)}{=} \frac{G}{T} \sqrt{T \mathbb{E}_{\boldsymbol{X}, \boldsymbol{y}, \boldsymbol{x}} \left[ \left\| \frac{\sigma(\boldsymbol{W}^{(1)}\boldsymbol{X} + \boldsymbol{b}\mathbf{1}_N^\top)\boldsymbol{y}}{N} \begin{bmatrix} \sigma((\boldsymbol{w}_1^{(1)})^\top \boldsymbol{x} + b_1) \\ \vdots \\ \sigma((\boldsymbol{w}_m^{(1)})^\top \boldsymbol{x} + b_m) \end{bmatrix}^\top \right\|_F^2 \right]}$$

$$= \frac{G}{\sqrt{T}} \sqrt{\sum_{k,l=1}^m \mathbb{E}_{\boldsymbol{X}, \boldsymbol{y}, \boldsymbol{x}} \left[ \left( \frac{1}{N} \sum_{j=1}^N y_j \sigma((\boldsymbol{w}_k^{(1)})^\top \boldsymbol{x}_j + b_k) \sigma((\boldsymbol{w}_l^{(1)})^\top \boldsymbol{x} + b_l) \right)^2 \right]}$$

$$\overset{(c)}{\le} \frac{G}{\sqrt{T}} \sqrt{\sum_{k,l=1}^{m} \mathbb{E}_{\boldsymbol{x}\sim\mathcal{N}(0,\boldsymbol{I}_d),\boldsymbol{x}'\sim\mathcal{N}(0,\boldsymbol{I}_d),f_*,\varsigma}\left[(f_*(\boldsymbol{x})+\varsigma)^2\sigma((\boldsymbol{w}_k^{(1)})^\top\boldsymbol{x}+b_k)^2\sigma((\boldsymbol{w}_l^{(1)})^\top\boldsymbol{x}'+b_l)^2\right]}$$

$$\overset{(d)}{\le} \frac{G}{\sqrt{T}} \sqrt{\sum_{k,l=1}^{m} \mathbb{E}_{\boldsymbol{x},f_*,\varsigma}\left[(2f_*(\boldsymbol{x})^2+2\varsigma^2)(2((\boldsymbol{w}_k^{(1)})^\top\boldsymbol{x})^2+2b_k^2)\right]\mathbb{E}_{\boldsymbol{x}'}\left[(2((\boldsymbol{w}_l^{(1)})^\top\boldsymbol{x}')^2+2b_l^2)\right]},$$

where the annotated equality/inequalities are obtained by (a) Cauchy–Schwarz inequality, (b) $\mathbb{E}_{\epsilon_t\sim\mathrm{Unif}\{\pm1\}}[\|\sum_t \epsilon_t \boldsymbol{A}_t\|_F^2] = \sum_t \|\boldsymbol{A}_t\|_F^2$, (c) convexity of $f(z) = z^2$, and (d)$(\sigma(\alpha+\beta))^2 \le (\alpha+\beta)^2 \le 2\alpha^2 + 2\beta^2$.

Let us further evaluate the last line: from the assumption and Lemma 7, we know that $\|w_k\|$, $|b_k|$ and $\mathbb{E}[f_*(\boldsymbol{x})^2]$ are all $\tilde{O}(1)$: in particular, we can easily obtain that $E_{\boldsymbol{x}\sim\mathcal{N}(0,\boldsymbol{I}_d)}\left[((\boldsymbol{w}_k^{(1)})^\top\boldsymbol{x})^2\right] = \tilde{O}_{d,r}(1)$. Consequently, the last line is $\frac{G}{\sqrt{T}}\sqrt{\sum_{k,l=1}^{m}\mathbb{E}_{\boldsymbol{x}\sim\mathcal{N}(0,\boldsymbol{I}_d),f_*}\left[(2f_*(\boldsymbol{x})^2)(2((\boldsymbol{w}_k^{(1)})^\top\boldsymbol{x})^2)\right]} + \tilde{O}_{d,r}(1)$, and remains to evaluate $\mathbb{E}_{\boldsymbol{x}\sim\mathcal{N}(0,\boldsymbol{I}_d),f_*}\left[f_*(\boldsymbol{x})^2((\boldsymbol{w}_k^{(1)})^\top\boldsymbol{x})^2\right]$. On this term, we obtain

$$\mathbb{E}_{\boldsymbol{x},f_*}\left[f_*(\boldsymbol{x})^2((\boldsymbol{w}_k^{(1)})^\top\boldsymbol{x})^2\right] \le \mathbb{E}_{\boldsymbol{x},f_*}\left[f_*(\boldsymbol{x})^4\right]^{1/2}\mathbb{E}_{\boldsymbol{x}}\left[((\boldsymbol{w}_k^{(1)})^\top\boldsymbol{x})^4\right]^{1/2}$$

$$= \tilde{O}(1) \ (\because \text{ Lemma } 7).$$

Hence we can conclude $\mathrm{Rad}_T(\mathcal{F}_{N,G}) = O\left(\frac{Gm}{\sqrt{T}}\right)$. $\square$

**Corollary 46.** *Let*

$$\tilde{\mathcal{F}}_{N,G} = \left\{(\boldsymbol{X},\boldsymbol{y},\boldsymbol{x},y) \mapsto \left\langle \frac{\boldsymbol{\Gamma}\sigma(\boldsymbol{W}^{(1)}\boldsymbol{X}+\boldsymbol{b}\boldsymbol{1}_N^\top)\boldsymbol{y}}{N}, \begin{bmatrix} \sigma((\boldsymbol{w}_1^{(1)})^\top\boldsymbol{x}+b_1) \\ \vdots \\ \sigma((\boldsymbol{w}_m^{(1)})^\top\boldsymbol{x}+b_m) \end{bmatrix} \right\rangle \middle| \|\boldsymbol{\Gamma}\|_F \le G \right\}$$

$$\cup \left\{(\boldsymbol{X},\boldsymbol{y},\boldsymbol{x},y) \mapsto y\right\}.$$

*Then,* $\mathrm{Rad}_T(\tilde{\mathcal{F}}_{N,G}) = \tilde{O}_{d,r}\left(\frac{Gm}{\sqrt{T}}\right)$ *holds when* $|b_j| = \tilde{O}_{d,r}(1)$ *and* $\|\boldsymbol{w}_j^{(1)}\| = \tilde{O}_{d,r}(1)$ *for all* $j \in [m]$.

**Proof.** Note that

$$\mathbb{E}_{\{\boldsymbol{X}^t\},\{\boldsymbol{y}^t\},\{\boldsymbol{x}^t\},\{y^t\},\{\epsilon^t\}}\left[\sup_{f\in\tilde{\mathcal{F}}_{N,G}} \frac{1}{T}\sum_{t=1}^{T}\epsilon^t f(\boldsymbol{X}^t,\boldsymbol{y}^t,\boldsymbol{x}^t,y^t)\right]$$

$$\le \mathbb{E}_{\{\boldsymbol{X}^t\},\{\boldsymbol{y}^t\},\{\boldsymbol{x}^t\},\{\epsilon^t\}}\left[\sup_{f\in\mathcal{F}_{N,G}} \frac{1}{T}\sum_{t=1}^{T}\epsilon^t f(\boldsymbol{X}^t,\boldsymbol{y}^t,\boldsymbol{x}^t)\right] + \mathbb{E}_{\{y^t\},\{\epsilon^t\}}\left[\left|\frac{1}{T}\sum_{t=1}^{T}\epsilon^t y^t\right|\right].$$

This follows from $\sup_{f\in\mathcal{F}_{N,G}} \frac{1}{T}\sum_{t=1}^{T}\epsilon^t f(\boldsymbol{X}^t,\boldsymbol{y}^t,\boldsymbol{x}^t) \ge 0$, as $\mathcal{F}_{N,G}$ contains zero function corresponding to $\boldsymbol{\Gamma} = \boldsymbol{O}$. The second term in the last line can be bounded as

$$\mathbb{E}_{\{y^t\},\{\epsilon^t\}}\left[\left|\frac{1}{T}\sum_{t=1}^{T}\epsilon^t y^t\right|\right] \le \sqrt{\mathbb{E}_{\{y^t\},\{\epsilon^t\}}\left[\left(\frac{1}{T}\sum_{t=1}^{T}\epsilon^t y^t\right)^2\right]}$$

$$= \frac{1}{\sqrt{T}}\mathbb{E}[(y^t)^2] = O\left(\frac{1}{\sqrt{T}}\right)$$

by Lemma 7. $\square$

## D.2 Prompt Length-free Generalization Bound

The ICL risk for prompt length $N$ was defined as

$$\mathcal{R}_N(f) = \mathbb{E}_{\boldsymbol{X},\boldsymbol{y},\boldsymbol{x},y}[|f(\boldsymbol{X}_{1:N}, \boldsymbol{y}_{1:N}, \boldsymbol{x}; \boldsymbol{W}, \boldsymbol{\Gamma}, \boldsymbol{b}) - y|],$$

where the length of $\boldsymbol{X}_{1:N}$ and $\boldsymbol{y}_{1:N}$ is fixed to $N$. In this section, we show that the ICL risk "converges" when the context length is sufficiently large.

**Proposition 47.** *Assume that $\|\boldsymbol{w}_j\| = \tilde{O}_{d,r}(1)$ and $|b_j| = \tilde{O}_{d,r}(1)$ for each $j \in [m]$. Then,*

$$|\mathcal{R}_N(f) - \mathcal{R}_M(f)| = \tilde{O}\Big(\|\boldsymbol{\Gamma}\|_F m \sqrt{1/N + 1/M}\Big)$$

*holds.*

**Proof.** Note that

$$
\begin{aligned}
&|\mathcal{R}_N(f) - \mathcal{R}_M(f)|\\
&\leq \mathbb{E}[|f(\boldsymbol{X}_{1:N}, \boldsymbol{y}_{1:N}, \boldsymbol{x}; \boldsymbol{W}, \boldsymbol{\Gamma}, \boldsymbol{b}) - f(\boldsymbol{X}_{1:M}, \boldsymbol{y}_{1:M}, \boldsymbol{x}; \boldsymbol{W}, \boldsymbol{\Gamma}, \boldsymbol{b})|]\\
&= \mathbb{E}\left[\left|\left\langle \boldsymbol{\Gamma}\left(\frac{\sigma(\boldsymbol{W}\boldsymbol{X}_{1:N} + \boldsymbol{b}\boldsymbol{1}_N^\top)\boldsymbol{y}_{1:N}}{N} - \frac{\sigma(\boldsymbol{W}\boldsymbol{X}_{1:M} + \boldsymbol{b}\boldsymbol{1}_M^\top)\boldsymbol{y}_{1:M}}{M}\right), \begin{bmatrix} \sigma(\boldsymbol{w}_1^\top \boldsymbol{x} + b_1) \\ \vdots \\ \sigma(\boldsymbol{w}_m^\top \boldsymbol{x} + b_m) \end{bmatrix}\right\rangle\right|\right]\\
&\overset{(a)}{\leq} \|\boldsymbol{\Gamma}\|_F \mathbb{E}\left[\left\|\left(\frac{\sigma(\boldsymbol{W}\boldsymbol{X}_{1:N} + \boldsymbol{b}\boldsymbol{1}_N^\top)\boldsymbol{y}_{1:N}}{N} - \frac{\sigma(\boldsymbol{W}\boldsymbol{X}_{1:M} + \boldsymbol{b}\boldsymbol{1}_M^\top)\boldsymbol{y}_{1:M}}{M}\right)\right\| \left\| \begin{bmatrix} \sigma(\boldsymbol{w}_1^\top \boldsymbol{x} + b_1) \\ \vdots \\ \sigma(\boldsymbol{w}_m^\top \boldsymbol{x} + b_m) \end{bmatrix}\right\|\right]\\
&\leq \|\boldsymbol{\Gamma}\|_F \mathbb{E}\left[\left\|\left(\frac{\sigma(\boldsymbol{W}\boldsymbol{X}_{1:N} + \boldsymbol{b}\boldsymbol{1}_N^\top)\boldsymbol{y}_{1:N}}{N} - \frac{\sigma(\boldsymbol{W}\boldsymbol{X}_{1:M} + \boldsymbol{b}\boldsymbol{1}_M^\top)\boldsymbol{y}_{1:M}}{M}\right)\right\|^2\right]^{1/2}\\
&\quad \cdot \mathbb{E}\left[\left\|\begin{bmatrix} \sigma(\boldsymbol{w}_1^\top \boldsymbol{x} + b_1) \\ \vdots \\ \sigma(\boldsymbol{w}_m^\top \boldsymbol{x} + b_m) \end{bmatrix}\right\|^2\right]^{1/2},
\end{aligned}
$$

where (a) holds because $\boldsymbol{a}^\top \boldsymbol{\Gamma} \boldsymbol{b} = \boldsymbol{\Gamma} \circ (\boldsymbol{a}\boldsymbol{b}^\top) \leq \|\boldsymbol{\Gamma}\|_F \|\boldsymbol{a}\boldsymbol{b}^\top\|_F = \|\boldsymbol{\Gamma}\|_F \|\boldsymbol{a}\|\|\boldsymbol{b}\|$.

Let us bound $\mathbb{E}\left[\left\|\left(\frac{\sigma(\boldsymbol{W}\boldsymbol{X}_{1:N}+\boldsymbol{b}\boldsymbol{1}_N^\top)\boldsymbol{y}_{1:N}}{N} - \frac{\sigma(\boldsymbol{W}\boldsymbol{X}_{1:M}+\boldsymbol{b}\boldsymbol{1}_M^\top)\boldsymbol{y}_{1:M}}{M}\right)\right\|^2\right]$, which is to $\sum_{j=1}^m \mathbb{E}_{f_*}\left[\mathbb{E}_{\boldsymbol{X},\boldsymbol{y}}\left[\left(\frac{1}{N}\sum_{i=1}^N \sigma(\boldsymbol{w}_j^\top \boldsymbol{x}_i + b_j)y_i - \frac{1}{M}\sum_{i=1}^M \sigma(\boldsymbol{w}_j^\top \boldsymbol{x}_i + b_j)y_i\right)^2\right]\right]$. For simplicity, we drop the subscript $j$ and obtain

$$
\begin{aligned}
&\mathbb{E}_{\boldsymbol{X},\boldsymbol{y}}\left[\left(\frac{1}{N}\sum_{i=1}^N \sigma(\boldsymbol{w}^\top \boldsymbol{x}_i + b)y_i - \frac{1}{M}\sum_{i=1}^M \sigma(\boldsymbol{w}^\top \boldsymbol{x}_i + b)y_i\right)^2\right]\\
&\leq 2\mathbb{E}\left[\left(\frac{1}{N}\sum_{i=1}^N \sigma(\boldsymbol{w}^\top \boldsymbol{x}_i + b)y_i - \mathbb{E}\big[\sigma(\boldsymbol{w}^\top \boldsymbol{x} + b)y\big]\right)^2\right]\\
&\quad + 2\mathbb{E}\left[\left(\frac{1}{M}\sum_{i=1}^M \sigma(\boldsymbol{w}^\top \boldsymbol{x}_i + b)y_i - \mathbb{E}\big[\sigma(\boldsymbol{w}^\top \boldsymbol{x} + b)y\big]\right)^2\right]\\
&= \frac{2}{N}\mathbb{V}\big[(\sigma(\boldsymbol{w}^\top \boldsymbol{x} + b)y)\big] + \frac{2}{M}\mathbb{V}\big[(\sigma(\boldsymbol{w}^\top \boldsymbol{x} + b)y)\big]\\
&\leq \left(\frac{2}{N} + \frac{2}{M}\right)\mathbb{E}\big[\sigma(\boldsymbol{w}^\top \boldsymbol{x} + b)^2(f_*(\boldsymbol{x}) + \varsigma)^2\big]\\
&\overset{(a)}{\leq} \left(\frac{2}{N} + \frac{2}{M}\right)\mathbb{E}\big[(2(\boldsymbol{w}^\top \boldsymbol{x})^2 + 2b^2)(2f_*(\boldsymbol{x})^2 + 2\varsigma^2)\big].
\end{aligned}
$$

Here (a) is obtained in the same vein as the derivation of (d) in the proof of Proposition 45. Moreover, in the same way as that proof, we can show that $\mathbb{E}\left[(2(\boldsymbol{w}^\top\boldsymbol{x})^2 + 2b^2)(2f_*(\boldsymbol{x})^2 + 2\varsigma^2)\right] = \tilde{O}_{d,r}(1)$. Consequently we get $\mathbb{E}\left[\left\|\left(\frac{\sigma(\boldsymbol{W}\boldsymbol{X}_{1:N}+\boldsymbol{b}\mathbf{1}_N^\top)\boldsymbol{y}_{1:N}}{N} - \frac{\sigma(\boldsymbol{W}\boldsymbol{X}_{1:M}+\boldsymbol{b}\mathbf{1}_M^\top)\boldsymbol{y}_{1:M}}{M}\right)\right\|^2\right] = \left(\frac{2}{N} + \frac{2}{M}\right)\tilde{O}_{d,r}(m)$.

Next, we observe that

$$\mathbb{E}\left[\left\|\begin{bmatrix}\sigma(\boldsymbol{w}_1^\top\boldsymbol{x}+b_1)\\ \vdots \\ \sigma(\boldsymbol{w}_m^\top\boldsymbol{x}+b_m)\end{bmatrix}\right\|^2\right] \leq 2\sum_{j=1}^m \mathbb{E}[(\boldsymbol{w}_j^\top\boldsymbol{x})^2 + b_j^2] = \tilde{O}_{d,r}(m).$$

Putting all things together, we arrive at

$$|\mathcal{R}_N(f) - \mathcal{R}_M(f)| = \tilde{O}\left(\|\boldsymbol{\Gamma}\|_F m\sqrt{1/N + 1/M}\right).$$

$\square$

### D.3 Proof of Theorem 1

Finally we are ready to prove our main theorem.

**Proof.** [Proof of Theorem 1] Let $\bar{\boldsymbol{\Gamma}}$ be the attention matrix constructed in Proposition 2 and let $\boldsymbol{\Gamma}^*$ be the minimizer of the ridge regression problem (line 8 in Algorithm 1). By the equivalence between optimization with $L_2$ regularization and norm-constrained optimization, there exists $\lambda_2$ such that

$$\|\boldsymbol{\Gamma}^*\|_F \leq \|\bar{\boldsymbol{\Gamma}}\|_F = \tilde{O}(r^{2P}/m),$$

$$\begin{aligned}&\left(\frac{1}{T_2}\sum_{t=T_1+1}^{T_1+T_2} |y^t - f(\boldsymbol{X}^t,\boldsymbol{y}^t,\boldsymbol{x}^t;\boldsymbol{W}^{(1)},\boldsymbol{\Gamma}^*,\boldsymbol{b})|\right)^2\\ \leq& \frac{1}{T_2}\sum_{t=T_1+1}^{T_1+T_2} (y^t - f(\boldsymbol{X}^t,\boldsymbol{y}^t,\boldsymbol{x}^t;\boldsymbol{W}^{(1)},\boldsymbol{\Gamma}^*,\boldsymbol{b}))^2\\ \leq& \frac{1}{T_2}\sum_{t=T_1+1}^{T_1+T_2} (y^t - f(\boldsymbol{X}^t,\boldsymbol{y}^t,\boldsymbol{x}^t;\boldsymbol{W}^{(1)},\bar{\boldsymbol{\Gamma}},\boldsymbol{b}))^2.\end{aligned}$$

Then, from Proposition 2,

$$\frac{1}{T_2}\sum_{t=T_1+1}^{T_1+T_2} |y^t - f(\boldsymbol{X}^t,\boldsymbol{y}^t,\boldsymbol{x}^t;\boldsymbol{W}^{(1)},\boldsymbol{\Gamma}^*,\boldsymbol{b})| - \tau = \tilde{O}\left(\sqrt{\frac{r^{3P}}{m} + \frac{r^{2P}}{N_2}}\right)$$

holds.

We evaluate $\mathcal{R}_{N_2}(f) - \tau$ where $f = f(\boldsymbol{X}^t,\boldsymbol{y}^t,\boldsymbol{x}^t;\boldsymbol{W}^{(1)},\boldsymbol{\Gamma}^*,\boldsymbol{b})$ using the Rademacher complexity.

$$\begin{aligned}&\mathcal{R}_{N_2}(f) - \tau\\ =& \frac{1}{T_2}\sum_{t=T_1+1}^{T_1+T_2} |y^t - f(\boldsymbol{X}^t,\boldsymbol{y}^t,\boldsymbol{x}^t;\boldsymbol{W}^{(1)},\boldsymbol{\Gamma}^*,\boldsymbol{b})|\\ &+ \left(\mathcal{R}_{N_2}(f) - \frac{1}{T_2}\sum_{t=T_1+1}^{T_1+T_2} |y^t - f(\boldsymbol{X}^t,\boldsymbol{y}^t,\boldsymbol{x}^t;\boldsymbol{W}^{(1)},\boldsymbol{\Gamma}^*,\boldsymbol{b})|\right) - \tau\\ \leq& \tilde{O}\left(\sqrt{\frac{r^{3P}}{m} + \frac{r^{2P}}{N_2}}\right)\\ &+ \sup_{f\in\tilde{\mathcal{F}}_{N,\|\bar{\boldsymbol{\Gamma}}\|_F}}\left(\mathcal{R}_{N_2}(f) - \frac{1}{T_2}\sum_{t=T_1+1}^{T_1+T_2} |y^t - f(\boldsymbol{X}^t,\boldsymbol{y}^t,\boldsymbol{x}^t,y^t)|\right)\end{aligned}$$

holds, where $\tilde{\mathcal{F}}$ is defined in Corollary 46. We can evaluate the expectation value of the second term of the last line as, using $\epsilon^t \overset{\text{i.i.d.}}{\sim} \text{Unif}\{\pm 1\}$,

$$\mathbb{E}_{\{\boldsymbol{X}^t\},\{\boldsymbol{y}^t\},\{\boldsymbol{x}^t\},\{y^t\}} \left[ \sup_{f \in \tilde{\mathcal{F}}_{N,\|\bar{\boldsymbol{\Gamma}}\|_F}} \left( \mathcal{R}_{N_2}(f) - \frac{1}{T_2} \sum_{t=T_1+1}^{T_1+T_2} |y^t - f(\boldsymbol{X}^t, \boldsymbol{y}^t, \boldsymbol{x}^t, y^t)| \right) \right]$$

$$\overset{(a)}{\le} 2\mathbb{E}_{\{\boldsymbol{X}^t\},\{\boldsymbol{y}^t\},\{\boldsymbol{x}^t\},\{y^t\},\{\epsilon^t\}} \left[ \sup_{f \in \tilde{\mathcal{F}}_{N,\|\bar{\boldsymbol{\Gamma}}\|_F}} \frac{1}{T_2} \sum_{t=T_1+1}^{T_1+T_2} \epsilon^t |y^t - f(\boldsymbol{X}^t, \boldsymbol{y}^t, \boldsymbol{x}^t, y^t)| \right]$$

$$\overset{(b)}{\lesssim} \text{Rad}_{T_2}(\tilde{\mathcal{F}}_{N_2,\|\bar{\boldsymbol{\Gamma}}\|_F}) + \mathbb{E}_{y,\epsilon} \left[ \frac{1}{T_2} \sum_{t=T_1+1}^{T_1+T_2} \epsilon^t y^t \right]$$

$$\le \text{Rad}_{T_2}(\tilde{\mathcal{F}}_{N_2,\|\bar{\boldsymbol{\Gamma}}\|_F}) + \mathbb{E}_{y,\epsilon} \left[ \left( \frac{1}{T_2} \sum_{t=T_1+1}^{T_1+T_2} \epsilon^t y^t \right)^2 \right]^{1/2}$$

$$\le \text{Rad}_{T_2}(\tilde{\mathcal{F}}_{N_2,\|\bar{\boldsymbol{\Gamma}}\|_F}) + \frac{1}{\sqrt{T_2}} \mathbb{E}[y^2]^{1/2} \overset{(c)}{=} \tilde{O}\left( \sqrt{\frac{r^{4P}}{T_2}} \right).$$

Here, (a) holds from the standard symmetrization argument (see eq. (4.17) in [Wai19] for example), (b) is derived using eq. (1) in [Mau16], noting that $(x,y)^\top \mapsto |x-y|$ is $\sqrt{2}$-Lipschitz continuous, and (c) follows from the bound of $\|\bar{\boldsymbol{\Gamma}}\|_F$ in Proposition 2, together with Lemma 7 and Proposition 45.

Note that

$$\sup_{f \in \tilde{\mathcal{F}}_{N,\|\bar{\boldsymbol{\Gamma}}\|_F}} \left( \mathcal{R}_{N_2}(f) - \frac{1}{T_2} \sum_{t=T_1+1}^{T_1+T_2} |y^t - f(\boldsymbol{X}^t, \boldsymbol{y}^t, \boldsymbol{x}^t, y^t)| \right) \tag{D.1}$$

is nonnegative because $f(\boldsymbol{X}, \boldsymbol{y}, \boldsymbol{x}, y) = y$ is included in $\tilde{\mathcal{F}}_{N,\|\bar{\boldsymbol{\Gamma}}\|_F}$. Hence from Markov's inequality, (D.1) is $\tilde{O}\left( \sqrt{\frac{r^{4P}}{T_2}} \right)$ with probability at least 0.995. Hence we obtain

$$\mathcal{R}_{N_2}(f) - \tau = \tilde{O}\left( \sqrt{\frac{r^{3P}}{m} + \frac{r^{2P}}{N_2}} + \sqrt{\frac{r^{4P}}{T_2}} \right).$$

Now we have done the upper bound for $\mathcal{R}_{N_2}(f)$. For $\mathcal{R}_{N^*}(f)$, we can use Proposition 47.

Note that all the desired events occur with high probability, except for the one described above, which occurs with probability 0.995. This completes the proof of Theorem 1. $\qquad\square$

# E   Derivation of Simplified Self-attention Module

We derive equation (2.6), following the same line of argument as [ZFB23]. As we assumed that $\boldsymbol{W}^{PV} \boldsymbol{W}^{KQ}$ are in the form as (2.4), the output of the attention layer is written as

$$\tilde{f}_{\text{Attn}}(\boldsymbol{E}; \boldsymbol{W}^{PV}, \boldsymbol{W}^{KQ}) = \boldsymbol{E} + \begin{bmatrix} * & * \\ \boldsymbol{0}_{1 \times m} & v \end{bmatrix} \boldsymbol{E} \cdot \frac{\boldsymbol{E}^\top \begin{bmatrix} \boldsymbol{K} & * \\ \boldsymbol{0}_{1 \times m} & * \end{bmatrix} \boldsymbol{E}}{N}. \tag{E.1}$$

Note that we adopt the right-bottom entry $\left( \tilde{f}_{\text{Attn}}(\boldsymbol{E}; \boldsymbol{W}^{PV}, \boldsymbol{W}^{KQ}) \right)_{m+1,N+1}$ as a prediction for output of the query. By (E.1). Hence we have

$$\left( \tilde{f}_{\text{Attn}}(\boldsymbol{E}; \boldsymbol{W}^{PV}, \boldsymbol{W}^{KQ}) \right)_{m+1,N+1}$$

$$
\begin{aligned}
&\quad \boldsymbol{E}^{\top}\begin{bmatrix}\boldsymbol{K} & * \\ \boldsymbol{0}_{1\times m} & *\end{bmatrix}\begin{bmatrix}\sigma(\boldsymbol{w}_1^{\top}\boldsymbol{x}+b_1)\\ \vdots \\ \sigma(\boldsymbol{w}_m^{\top}\boldsymbol{x}+b_m)\\ 0\end{bmatrix}\\
&=[\boldsymbol{0}_{1\times m}\quad v]\boldsymbol{E}\cdot\frac{}{N}\\[2mm]
&=v[y_1\ \cdots\ y_N\ \ 0]\cdot\frac{\boldsymbol{E}^{\top}\begin{bmatrix}\boldsymbol{K}\begin{bmatrix}\sigma(\boldsymbol{w}_1^{\top}\boldsymbol{x}+b_1)\\ \vdots \\ \sigma(\boldsymbol{w}_m^{\top}\boldsymbol{x}+b_m)\end{bmatrix}\\ 0\end{bmatrix}}{N}\\[2mm]
&=v[y_1\ \cdots\ y_N]\cdot\frac{\sigma(\boldsymbol{W}\boldsymbol{X}+\boldsymbol{b}\boldsymbol{1}_N^{\top})^{\top}\boldsymbol{K}\begin{bmatrix}\sigma(\boldsymbol{w}_1^{\top}\boldsymbol{x}+b_1)\\ \vdots \\ \sigma(\boldsymbol{w}_m^{\top}\boldsymbol{x}+b_m)\end{bmatrix}}{N}\\[2mm]
&=\left\langle\frac{v\boldsymbol{K}^{\top}\sigma(\boldsymbol{W}\boldsymbol{X}+\boldsymbol{b}\boldsymbol{1}_N^{\top})\boldsymbol{y}}{N},\begin{bmatrix}\sigma(\boldsymbol{w}_1^{\top}\boldsymbol{x}+b_1)\\ \vdots \\ \sigma(\boldsymbol{w}_m^{\top}\boldsymbol{x}+b_m)\end{bmatrix}\right\rangle.
\end{aligned}
$$

Setting $\boldsymbol{\Gamma}=v\boldsymbol{K}^{\top}$ yields equation (2.6).

# F    Details of Experiments

## F.1    Detailed Experimental Settings

For the experiments in Section 4, test loss was averaged over 128 independent tasks, with 2048 and 128 independent queries generated for each task in the experiment of Figure 1 and 2, respectively. During the validation of the experiment of Figure 1, the coefficients $\{c_i\}$ in the single-index model were fixed to be $(c_2,c_3)=(\sqrt{2\cdot 2!}/2,\sqrt{2\cdot 3!}/2)$ to reduce the variance in the baseline methods.

**GPT-2 model.**    The experiments in Section 4 were based on the architecture used in [GTLV22][1], whose backbone is the GPT-2 model [RWC$^+$19]. Given the prompt $\{(\boldsymbol{x}_i,y_i)\}_{i=1}^{N+1}$, the embedding $\boldsymbol{E}\in\mathbb{R}^{d\times(2N+2)}$ was constructed as $\boldsymbol{E}=[\boldsymbol{x}_1,\tilde{\boldsymbol{y}}_1,\ldots,\boldsymbol{x}_N,\tilde{\boldsymbol{y}}_N]$ where $\tilde{\boldsymbol{y}}_i=[y_i,0,\ldots,0]^{\top}$. This embedding was mapped to $\tilde{\boldsymbol{E}}\in\mathbb{R}^{256\times(2N+2)}$ through the read-in layer. The transformed embedding $\tilde{\boldsymbol{E}}$ was then passed through the 12-layer GPT-2 backbone with 8 heads, with dropout disabled. Finally, the output was converted into a $2N+2$-dimensional vector through the read-out layer, where the $(2k-1)$-th entry ($k\in[N+1]$) corresponds to the prediction $\hat{y}_k(\boldsymbol{x}_1^t,y_1^t\ldots,\boldsymbol{x}_{k-1}^t,y_{k-1}^t,\boldsymbol{x}_k^t)$ for $y_k$, that is, the answer for the query $\boldsymbol{x}_k^t$ reading the context of length $k-1$. We used the Adam optimizer [KB15] with a learning rate of 0.0001. The training loss at each step was set as $\frac{1}{B}\sum_{t=1}^{B}\sum_{k=1}^{N+1}\left(y_k^t-\hat{y}_k(\boldsymbol{x}_1^t,y_1^t\ldots,\boldsymbol{x}_{k-1}^t,y_{k-1}^t,\boldsymbol{x}_k^t)\right)^2$ where $B=8$ is the minibatch size.

**Baseline algorithms.**    Baseline algorithms used in the comparison experiment are as follows:

- Two-layer neural network $f(\boldsymbol{x})=\sum_{j=1}^m a_j\sigma(\boldsymbol{w}_j^{\top}\boldsymbol{x})$ where $\sigma=\mathrm{ReLU}$ and $m=256$. The initialization scale followed the mean-field regime, with $a_j\overset{\text{i.i.d.}}{\sim}\mathrm{Unif}(\{\pm 1/m\})$ and $\boldsymbol{w}_j\overset{\text{i.i.d.}}{\sim}\mathrm{Unif}(\mathbb{S}^{d-1})$. The architecture was trained using the Adam optimizer, with a learning rate of 0.1 for the first layer and $0.1/m$ for the second layer, along with a weight dacay rate of 0.0001. Early stopping was applied to prevent overfitting.

- Kernel ridge regression using the Gaussian RBF kernel. The ridge regression parameter $\lambda$ was set to $\lambda=0.01$.

Again, note that these baseline algorithms were trained from scratch for each validation task, whereas the parameters of the pretrained GPT-2 model remained unchanged across all validation tasks.

---

[1]https://github.com/dtsip/in-context-learning, MIT License.

## F.2 Experiment on Our Architecture and Algorithm 1

In addition to the GPT-2 experiments, we also conducted comparison between our Algorithm 1 on our simplified architecture and baseline algorithms. Data is generated as $y_i^t = \text{He}_2(\langle \boldsymbol{x}_i^t, \boldsymbol{\beta}^t \rangle)$, where $\boldsymbol{\beta}^t \overset{\text{i.i.d.}}{\sim}$ Unif$\{\boldsymbol{\beta} \in \mathbb{R}^d \mid \boldsymbol{\beta} = [\boldsymbol{\beta}_{1:r}, \boldsymbol{0}]^\top, \|\boldsymbol{\beta}\| = 1\}$ with $d = 32$ and $r = 2$. We pretrained our transformer architecture (2.6) with $m = 8,000$, using Algorithm 1 with $T_1 = 10^5, N_1 = 10^4, \eta_1 = 10^5$ (recall that $\eta_1 \asymp m^{\frac{3}{2}} r d^{2Q - \frac{1}{2}} \cdot (\log d)^{-C_\eta}$ should be large) and $T_2 = 3,000, N_2 = 512, \lambda_2 = 0.1$. We used mini-batch SGD to find an approximate optimizer $\boldsymbol{\Gamma}^*$ of the ridge regression. For validation, we altered the context length from $2^{4.5}$ to $2^9$ and compared its test error with one-step GD (training the first layer via one-step gradient descent, and then conducting ridge regression on the second layer) on a two-layer NN of width $8,000$ and kernel ridge regression with RBF kernel working directly on

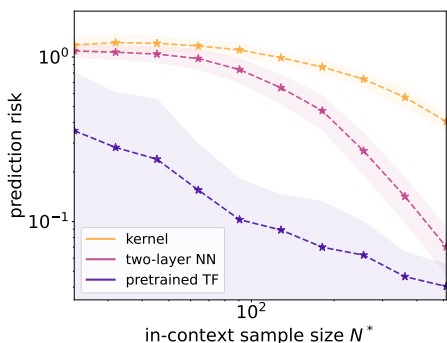

Figure 3: In-context generalization error of kernel ridge regression, neural network + one-step gradient descent, and Algorithm 1 on our Transformer (2.6).

the test prompt. In Figure 3 we see that our simplified transformer also outperforms neural network and kernel method, especially in short context regime.

