# OpenReview forum: "Pretrained Transformer Efficiently Learns Low-Dimensional Target Functions In-Context"
_NeurIPS.cc/2024/Conference — NeurIPS 2024 poster_

### Official Review · Reviewer_UaEF · 2024-07-11

**Soundness:** 3
**Presentation:** 3
**Contribution:** 2
**Rating:** 4
**Confidence:** 4

**Summary:**

This paper investigates the ICL abilities of transformers for learning nonlinear function classes. The authors study transformers with nonlinear MLP layers and demonstrate that such transformers can efficiently learn single-index target functions in-context with a prompt length dependent on the function's low-dimensional structure. The proposed method could significantly reduce the statistical complexity compared to direct learning methods.

**Strengths:**

1. The topic of this work is both interesting and important.

2. The analysis provided is rigorous and sound. I think the low-rank result is very encouraging that the risk bound depend on r not d, which could be very helpful for explaining many success of LLMs.

3. The presentation is clear and easy to read.

**Weaknesses:**

1. Although the ICL capabilities of transformers are indeed powerful and of great interest, the model and pre-training algorithm presented in this paper are not very realistic. While it is common to use simplified models for analyzing LLMs, the authors' pre-training algorithm involves setups (such as gradient descent and (re-)initialization) that deviate from standard LLM training approaches.

2. Given the unrealistic setup, the paper would benefit greatly from numerical experiments demonstrating the effectiveness of the proposed algorithm in learning nonlinear functions. If obtaining these results is challenging, conducting other experiments that could provide intuition or insights into understanding ICL in standard LLMs would also be helpful.

3. While the work seems very relevant to understanding ICLs, the paper lacks real evidence supporting this connection. If I am misunderstanding the relevance, I would appreciate it if the authors could provide more detailed explanations on how this work contributes to understanding the mechanisms of ICL and LLMs.

**Questions:**

Please see part 3 in the last section.

---

> ### Author Rebuttal · Authors · 2024-08-07
>
> We thank the reviewer for the helpful feedback. We address the technical concerns below.
>
> ---
>
> **"pre-training algorithm involves setups that deviate from standard LLM training approaches"**
>
> We agree that Algorithm 1 deviates from the most standard training procedure in practice. We make the following remarks on our architecture and optimization algorithm.
>
> - **MLP before attention.** This simplified theoretical setting reflects the scenario where lower layers of the trained transformer act as data representations on top of which the upper layers perform ICL.
> See Figure 1 in [[Guo et al. 2023]](https://openreview.net/pdf?id=ikwEDva1JZ): they considered a Transformer architecture with alternating MLP and attention layers and suggested that the earlier layers grasp the features of the true function, while the subsequent layers act as linear attention.Our architecture, with an MLP layer for feature learning followed by a linear attention module, idealizes such process occurring in more complicated Transformer.
> The same architecture has been analyzed in prior works on in-context learning of nonlinear features such as [[Kim and Suzuki 2024]](https://proceedings.mlr.press/v235/kim24af.html).
>
> - **Linear attention.** In the ICL theory literature, the Softmax attention is often simplified to linear attention to facilitate the optimization analysis. Such simplification can capture several prominent aspects of practical transformer training dynamics -- see [[Ahn et al. 2023]](https://openreview.net/pdf?id=0uI5415ry7) and references therein. Also, in our setting we already observe interesting in-context behavior using a linear attention. Extending our analysis to Softmax attention is an interesting future direction.
>
> - **Layer-wise training and one-step GD.**
> In the neural network theory literature, layer-wise training and reinitialized bias units are fairly standard algorithmic modifications when proving end-to-end sample complexity guarantees, as seen in many prior works on feature learning [[Damian et al. 2022]](https://proceedings.mlr.press/v178/damian22a/damian22a.pdf) [[Abbe et al. 2023]](https://proceedings.mlr.press/v195/abbe23a/abbe23a.pdf). Without such modifications, stronger target assumptions are generally needed, such as matching link function [[Ben Arous et al. 2020]](https://jmlr.csail.mit.edu/papers/volume22/20-1288/20-1288.pdf).
> In addition, training the representation for one gradient step (followed by SGD on the top layer) is a well-studied optimization procedure for two-layer neural networks (with the hope being that the "early phase" of representation learning already extracts useful features) -- see [[Ba et al. 2022]](https://openreview.net/pdf?id=akddwRG6EGi) [[Barak et al. 2022]](https://openreview.net/pdf?id=8XWP2ewX-im); but to our knowledge, we are the first to apply this dynamics to nonlinear transformers in the ICL setting and establish rigorous statistical guarantees.
>
> ---
>
> **"numerical experiments demonstrating the effectiveness of the proposed algorithm"**
>
> Following your suggestion, we conducted experiments that supports our theoretical claim on the statistical efficiency of ICL -- please see the .pdf document attached in our general response. Specifically, in Figure 1 we indeed observe that when $r\ll d$, there is a separation in the test loss between pretrained transformers and baseline algorithms that directly learn from the test prompt, including kernel ridge regression and two-layer neural network + gradient descent.
>
> ---
>
> **"how this work contributes to understanding the mechanisms of ICL and LLMs"**
>
> We believe that the the following high-level messages from our theoretical results are relevant:
>
> * Our analysis highlights how pretrained transformers can leverage prior information (low-dimensional structure in particular) of the function class and outperform simple baseline algorithms. Such a phenomenon is empirically known (e.g., on linear functions, it had been observed that Transformers adapt to the "sparsity of problem distribution" [[Garg et al. 2022]](https://openreview.net/pdf?id=flNZJ2eOet)), but existing theoretical analysis with end-to-end learning guarantees typically showed that ICL implements a simple *prior-agnostic* algorithm, such as one gradient descent step on the least squares objective for linear regression.
> In addition,  our finding that transformer can adapt to a low-rank target prior suggests, at a high level, a generalization error scaling that does not depend on the ambient dimensionality of the problem, but some notion of "intrinsic dimensionality" of the function class.
>
> * Our analysis makes use of a mechanism for ICL in shallow transformers: *feature extraction \& adaptivity to low-dimensionality* by MLP layer + *in-context function approximation* by attention layer.  We believe that similar mechanism also appears in modern pretrained transformers, which can provide insight to mechanistic interpretability research.
>
> ---
>
> We would be happy to clarify any further concerns/questions in the discussion period.

---

> > ### Author Response · Authors · 2024-08-11
> >
> > Dear Reviewer,
> >
> > As the discussion period is ending in two days, we wanted to follow up to check whether our response has adequately addressed your concerns for an improved score. If there are any remaining questions or concerns, we would be happy to provide further clarification.
> >
> > Thank you for your time and consideration.

---

> > ### Comment · Reviewer_UaEF · 2024-08-12
> >
> > Thanks for the rebuttal. I think my concerns are partially addressed. While I appreciate the effort of conducting relevant numerical experiments, I feel like it only provides marginal intuition and understanding. I also understand that it is not easy to provide a thorough numerical experiment in such a short rebuttal period.
> >
> > Therefore, my concerns are only partially addressed. I will maintain my original rating, but I will take your effort into consideration in the later stage.

---

> ### Author Response · Authors · 2024-08-13
>
> We thank the reviewer for the updated evaluation. We would like to address the reviewer's outstanding concern with the following remarks:
>
> * As a theory paper, our primary goal is to provide the *first* end-to-end learning guarantee (that accounts for the model width, number of tasks, prompt length, etc.) when learning a *nonlinear* class of functions in context. The single-index function class is significantly more challenging than the linear regression class examined in prior works. Our statistical guarantees entail a separation in statistical efficiency between pretrained transformers and algorithms that directly learn from the test prompt, while also demonstrating the adaptivity to "prior knowledge" of the target function and the benefits of nonlinear MLP layer. We believe that rigorously establishing these points, even in an idealized setting, represents an important contribution.
> We would also like to point out that even in the simpler linear setting, existing end-to-end analyses typically assume specific model parameterization and training procedures (e.g., gradient flow, infinite tasks) [[Zhang et al. 2023]](https://www.jmlr.org/papers/volume25/23-1042/23-1042.pdf) [[Ahn et al. 2023]](https://openreview.net/pdf?id=LziniAXEI9). We argue that these results do not offer more "*intuition or insights into understanding ICL in standard LLMs*" than our work does.
>
> * As mentioned in the Introduction, the single-index target function has been extensively studied in the deep learning theory literature; however, to our knowledge, we are the first to theoretically investigate the learning of this function class in the more challenging *in-context* setting.
> It is worth noting that even in the "easier" setting of standard neural network training (i.e., not considering ICL), some of the most influential papers on the theory of representation learning have considered algorithmic modifications similar to ours, such as layer-wise training, one-step feature learning, and reinitialized bias units [[Abbe et al. 2022]](https://proceedings.mlr.press/v178/abbe22a/abbe22a.pdf) [[Damian et al. 2022]](https://proceedings.mlr.press/v178/damian22a/damian22a.pdf) [[Ba et al. 2022]](https://proceedings.neurips.cc/paper_files/paper/2022/file/f7e7fabd73b3df96c54a320862afcb78-Paper-Conference.pdf).
> These works are not dismissed by the community because they "*deviate from standard training approaches*"; rather, they serve as an important first step toward understanding the statistical and computational efficiency of neural networks beyond simple linear problems.
> We hope the reviewer can recognize the nontrivial theoretical contribution of our analysis in this context.

---

### Official Review · Reviewer_gjHz · 2024-07-12

**Soundness:** 3
**Presentation:** 3
**Contribution:** 2
**Rating:** 5
**Confidence:** 4

**Summary:**

This work studies in-context learning on nonlinear functions, which are Hermite polynomials, using Transformers with linear attention and nonlinear MLP. When the nonlinear target function depends on a low-rank dimension, the ICL risk trained by gradient descent is proven to be a function of the rank. The proof supports the theory. I am willing to update my review after the rebuttal.

**Strengths:**

1. ICL on nonlinear functions is a significant problem to solve.
2. The proof is solid and reasonable to me.
3. Characterizing the ICL performance as a function of the internal complexity of the nonlinearity of the target function is a novel and nice result.

**Weaknesses:**

1. The theoretical result seems not to be tight. When $r$ is very close to $d$, the sample complexity is much worse than using two-layer neural networks. However, the Transformer considered in this paper already includes a nonlinear and trainable MLP layer. Then, I think the derived bound is not tight. In other words, why is the bound worse than two-layer NN?

2. No experiments are provided to justify the result, especially for the comparison with kernel methods and two-layer NN.

3. The presentation of the proof sketch can be improved. It is better to emphasize that finding $\Gamma$ is solving a convex problem (mentioned in line 206) between lines 262 and 264. I think the convexity is the key to the reason why the ICL risk of the constructed $\Gamma$ is attainable by gradient descent.

4. Some references on ICL optimization and generalization are missing. Li et. al., ICML 2024. "How Do Nonlinear Transformers Learn and Generalize in In-Context Learning?" (This work considers softmax attention and nonlinear MLP, which could be mentioned in line 93); Gatmiry et al., ICML 2024. "Can Looped Transformers Learn to Implement Multi-step Gradient Descent for In-context Learning?"
There is also a concurrent work: Yang et al., 2024. "In-Context Learning with Representations: Contextual Generalization of Trained Transformers", https://openreview.net/pdf?id=fShWHkLX3o It will be interesting to compare this concurrent work, although it is not required.

**Questions:**

1. I think Algorithm 1 is proposed to handle the non-convexity of the problem, right?

**Limitations:**

There is no potential negative societal impact of this work.

---

> ### Author Rebuttal · Authors · 2024-08-07
>
> We are thankful for your helpful comments. We address the concerns and answer the questions below.
>
> ---
>
> **"The theoretical result seems not to be tight."**
>
> Note that although we have an MLP block in our architecture, it is not trainable during the inference time. Indeed a sample complexity of $O(d^{\Theta(k)})$ for MLPs can be attained when we train the MLP parameters in response to each task (e.g., see [[Bietti et al. 2022]](https://openreview.net/pdf?id=wt7cd9m2cz2)).
> However, when considering ICL via Transformer, we need to fix the MLP layer and use the same parameters for all tasks, where the true function and direction $\beta$ can vary.Thus it is not surprising that our sample complexity is worse than NN+GD (trained on the test prompt) in the $r\simeq d$ regime.
> Instead, our focus is the regime where $r\ll d$, for which we show that pretrained transformer can adapt to this low-dimensional structure (target prior) and outperform baseline algorithms that only have access to the in-context examples.
>
> ---
>
> **"No experiments are provided to justify the result."**
>
> Following your suggestion, we conducted experiments that supports our theoretical claim on the statistical efficiency of ICL -- please see the .pdf document attached in our general response. Specifically, in Figure 1 we observe that when $r\ll d$, there is indeed a separation in the test loss between pretrained transformers and baseline algorithms that directly learn from the test prompt, including kernel ridge regression and two-layer neural network + gradient descent.
>
> ---
>
>
> **"The presentation of the proof sketch can be improved."**
>
> Thank you for the suggestion. As you pointed out, the convexity of the ridge regression problem for $\Gamma$ plays an important role because the optimal solution $\Gamma^*$ can be efficiently solved; this allows us to state that the approximation and generalization error of $\Gamma^*$ is not comparable to that of the constructed $\Gamma$.
> We will clarify this point in the proof sketch section.
>
> ---
>
> **"references on ICL optimization and generalization are missing"**
>
> Thank you for your suggestion.  We will add citations and comparison with these concurrent works.
>
> ---
>
> **"Algorithm 1 is proposed to handle the non-convexity of the problem?"**
>
> Yes, this simplification using one-step gradient descent circumvents the difficulty of analyzing the entire optimization trajectory with non-convex landscape. Such a strategy is commonly-used in the neural network theory literature -- training the representation for one gradient step (followed by SGD on the top layer) is a well-studied optimization procedure for two-layer neural networks (with the hope being that the "early phase" of representation learning already extracts useful features) -- see [[Ba et al. 2022]](https://openreview.net/pdf?id=akddwRG6EGi) [[Barak et al. 2022]](https://openreview.net/pdf?id=8XWP2ewX-im); but to our knowledge, we are the first to apply this dynamics to nonlinear transformers in the ICL setting and establish rigorous statistical guarantees.
>
> ---
> We would be happy to clarify any further concerns/questions in the discussion period.

---

> > ### Comment · Reviewer_gjHz · 2024-08-09
> >
> > Thank you for the response. My remaining question focuses on Weakness 1. I can get the point that you mainly study the case when $r\ll d$. That's OK. I just want to know more details of the comparison. For example, 1, in experiments, do you still fix the MLP layer or let it be trainable? 2. How many trainable parameters are there for the Transformer you use and the two-layer neural networks you use in the experiment? I would like to see whether the comparison makes sense.

---

> > > ### Author Response · Authors · 2024-08-09
> > >
> > > We thank the reviewer for the followup question. We make the following clarifications:
> > >
> > > - In the experiment, the MLP parameters of the transformer model are pretrained using Algorithm 1, and *fixed* during the in-context phase (across all tasks). Whereas for the two-layer MLP baseline, the parameters are directly trained on the in-context examples.
> > > - We set the MLP layer of the transformer and the first layer of the baseline two-layer network to be of the same size ($m\times d$). The transformer model also has ($m\times m$) attention parameters to be optimized.
> > > We note that the experiment aims to compare the sample complexity of ICL against baseline algorithms (which is the reviewer's concern in *Weakness 1*), not the parameter efficiency. A side remark is that for algorithms that directly learn the target function from in-context examples (e.g., kernel regression and two-layer NN + GD), the information theoretically optimal sample complexity is $n\gtrsim d$, meaning that when $r\ll d$, these baseline algorithms cannot match the in-context efficiency of transformers (which is independent of $d$) even with *infinite* compute or parameters.
> > >
> > > We would be happy to clarify any further concerns/questions in the discussion period.

---

> > > > ### Comment · Reviewer_gjHz · 2024-08-09
> > > >
> > > > Thank you for your response. I will raise the score to positive. The most attractive thing to me in this work is the framework for studying nonlinear functions for ICL, which characterizes the impacts of nonlinearity.

---

### Official Review · Reviewer_3HLy · 2024-07-12

**Soundness:** 3
**Presentation:** 3
**Contribution:** 4
**Rating:** 7
**Confidence:** 3

**Summary:**

* The authors theoretically study the capacity for (nonlinear) transformer
  models to perform in-context learning with a low-complexity function class.
* The set-up involves pretraining transformers on sequences encoding
  input-output examples from a reduced-rank Gaussian single-index model.
  * The use of Gaussian single index models gives the transformer a chance to
    implement a more intricate (but still analytically tractable) in-context
    learning algorithm than in-context linear regression for linear data.
  * The restriction of the in-context target functions to a low-rank subspace
    gives the transformer an opportunity to internalise (during pretraining)
    a 'prior' that it can rely on during in-context learning for a sample
    efficiency advantage.
* Within this setting the authors prove their main theorem, characterising
  the in-context sample efficiency of a transformer trained under certain
  conditions.
  * Importantly, the efficiency scales with the dimension of the
    low-dimensional model subspace rather than the ambient dimension.
  * This suggests that the pretrained transformer model is capable not only
    of performing ICL of Gaussian single-index models, but also of encoding
    the 'prior' and gaining a corresponding sample efficiency advantage from
    doing so.

**Strengths:**

I thank the authors for their contributions in this important area of
research. In my opinion the work shows at least the following strengths.

1. The work is well motivated by an important research goal within the field:
  advancing our theoretical understanding of more complex forms of ICL
  exhibited by transformers. This helps to bridge the gap between the
  nuanced ICL capabilities of LLMs and the simpler abilities of in-context
  linear regression transformers, for example.

2. The setting proposed by the authors, of learning Gaussian single-index
  models, appears to provide a neat example of a non-linear in-context
  learning capability making non-trivial use of a transformer's MLP layers,
  while still being analytically tractable and therefore amenable to
  informative comparison with baselines from the statistics and ML
  literature, as the authors have demonstrated.

3. The main theoretical result by the paper clearly demonstrates that
  pretrained transformers are capable of learning ICL for this nonlinear
  function class and that they are able to take advantage of the special
  low-dimensional structure of the function class in-context to improve their
  sample complexity over an ICL method that considers arbitrary directions.

4. The paper is reasonably well-organised and the writing is clear. Though for
  me the technical content is outside of my background and  zone, with
  careful study I was able to understand the setting and theorem statement
  thanks to the detailed technical description and accompanying discussion
  provided by the authors.

**Weaknesses:**

I lack the background to evaluate the soundness of the main theorem, but the
high-level structure of the main theorem statement seems plausible to me and
so I am open to taking the authors at their word.

Instead, the concerns I have with the paper are about the framing of the
contributions. In particular, my concern focuses on the second research
question, "Can a pretrained transformer adapt to certain structures of the
target function class, and how does such adaptivity contribute to the
statistical efficiency of ICL?" (Lines 45/46), to which the authors answer,
in summary, that, yes, there is a 'separation' (improved sample complexity)
between these transformers and "algorithms that directly learn the
single-index function from the test prompt".

My concerns with this framing are as follows:

1. **These baselines are potentially misleading.** The authors compare the
   sample complexity of a transformer model that has access to more
   information (through pretraining) against that of a baseline that lacks
   this information. It is not surprising that prior information can reduce
   sample complexity and this is why the transformer's performance beats this
   baseline.

     I think the appropriate framing would be more along the lines of saying
     that the transformer is able to (in some sense) 'match' other algorithms
     that have this same access to prior information, if these baselines are
     available in the literature.

     Note: I grant that the authors have shown that transformers can take
     advantage of this additional information, and this is a meaningful
     contribution. The specific concern is with the misleading comparison
     that might suggest transformers have some radical sample efficiency
     properties, when the improvement over these baselines has clearly come
     from access to additional information that the baselines don't have.

2. **This does not appear to be a completely novel observation.** There are at least
   empirical examples of transformers showing improved sample efficiency
   through making use of prior information gleaned from pretraining. For
   example, this is implied by the task diversity experiments of Raventos et
   al. (cited by the authors). In the low task-diversity regime, the model
   appears to memorise a short list of task vectors and when asked to perform
   ICL of these particular tasks it would obviously outperform a ridge
   regression model entertaining a larger hypothesis class. Therefore it
   appears that in some related work this second question has already been
   addressed.

     *Note:* These empirical observations are different from the present
     work's theoretical characterisation of the sample efficiency advantage
     of the pretrained transformer and I think this does not fundamentally
     detract from the authors' claims to novelty. However, I think it would
     be appropriate to introduce this question in the context of these
     previous attempts to answer it.

I would invite the authors to consider changing the framing of their paper to
more clearly qualify the nature of their contribution with respect to this
second research question.

**Questions:**

To begin with I will flag that before reading this paper I was not familiar
with Gaussian single-index models. Based on the information in the paper and
my own light additional reading, I think I have assembled enough
understanding to ask sensible questions, but I could be mistaken, and I
apologise in advance if my questions are ill-formed.

0. To this point, do the authors know of an accessible reference work that covers the topic of Gaussian single-index models? In case other potential readers are, like myself, not familiar with the topic, it could be useful to include a reference to a statistics textbook or similar.

1. A basic question: I don't think I understand line 52: "In single-index
   model, outputs only depend on the direction of $\beta_t$ in the
   $d$-dimensional input space." Is this just foreshadowing the constraint
   that $\beta_t$ is restricted to the unit sphere in the $r$-dimensional
   subspace? (Perhaps the constraint on the magnitude of $\beta$ should be
   mentioned in the introduction.)

2. I understand that the first goal addressed by this work is to create a
   setting where transformers exhibit nonlinear in-context learning behaviour
   (and I consider this a strength of the work). However, the shift from
   regression to single-index models brings with it additional statistical
   and architectural complexity. Therefore, with respect to your second
   stated goal of demonstrating adaptivity of ICL to low-dimensional
   structure in the training function class, I am wondering if it would be
   possible to get away with a simpler demonstration:

    1. In particular, what would happen if one considered the problem setting
       of in-context linear regression where the regression tasks are confined
       to a low-dimensional subspace (or a unit sphere on such a subspace)? Do
       you think that such a setting would offer a similar chance for the
       transformer to show its ability to internalise informative 'prior'
       information, but without the added complexity of considering
       single-index models?

    2. If I am not mistaken, this simpler low-rank regression setting is
       *not* a special case of your setting, since it would require the link
       function to be always the identity function which violates your
       assumptions about the coefficients in the Hermite expansion. Is that
       correct?

    3. Nevertheless, would you conjecture that a similar result to your main
       theorem---a generalisation bound that scales with the dimension of the
       low-dimensional subspace rather than that of the ambient
       dimension---would hold in such a setting?

Finally, some questions about future work.

3. Your theoretical work does not offer an empirical component. However, your
   analysis appears to make some clear predictions about what a transformer
   trained in this setting would learn, in terms of its ICL performance for
   this kind of data and even the internal mechanisms with which it might
   learn to perform ICL (such as using the MLP layers to encode the
   low-dimensional subspace).

    1. Do you think these performance predictions would be borne out by
       empirical experiments to this effect?

    2. Do you think that trained transformers would implement ICL in this
       setting using the mechanisms you have proposed?

    3. Are there any differences between your theoretical training algorithm
       and modern transformer training procedures that you expect might cause
       these predictions not to hold?

**Limitations:**

The authors have adequately addressed the limitations of their work. Assumptions are clearly stated and documented. My only concern in this regard is covered in the "Weaknesses" section of this review.

---

> ### Author Rebuttal · Authors · 2024-08-07
>
> We thank the reviewer for the thoughtful comment and constructive feedback. We address the technical concerns below.
>
> ---
>
> **Potentially misleading baseline and prior works**
>
> Thank you for the insightful comments. Indeed there are empirical examples of transformers achieving better statistical efficiency by making use of prior information of the function class -- we do not claim that we are introducing a new phenomenon. We will discuss [Raventos et al. 2023] in the Introduction and clarify the overlap in the high-level message. Our motivation is to provide rigorous end-to-end learning guarantee that captures this adaptivity in an idealized theoretical setting.
> We also agree that the comparison of sample complexity (pretrained transformers vs. algorithms on in-context test examples) is not straightforward, since transformers benefit from additional data in the pretraining phase. From the theoretical point of view, our goal is to highlight the statistical benefit of this prior information, since many previous theoretical works on ICL proved that transformers implement a simple "prior-agnostic" algorithm, such as one gradient descent step on the least squares objective for linear regression.
>
> ---
>
> **On Gaussian single-index models (Questions 0, 1, and 2)**
>
> * A single-index model is defined by an unknown univariate
> "link" function applied to an unknown one-dimensional projection of the input. Efficiently learning this function requires some extent of *adaptivity* to the low-dimensional structure, and hence this function class has been extensively studied in the neural network theory literature, to demonstrate the benefit of representation learning.  Here we list some works covering this topic: [[Ba et al. 2022]](https://openreview.net/pdf?id=akddwRG6EGi) [[Bietti et al. 2022]](https://openreview.net/pdf?id=wt7cd9m2cz2) [[Damian et al. 2023]](https://openreview.net/pdf?id=73XPopmbXH). We however realize that this problem setting has not been introduced in the context of transformer theory.
> We will revise the Introduction and Problem Setting section to provide more background and intuition.
>
> * Regarding the question 1, there we aimed to simply explain that in single-index models, the function value of $f_*$ changes only along the direction parallel to $\beta$ in the input space $\mathbb{R}^d$, because $f_*$ only depends on the inner product $\langle\beta,x\rangle$.
> We will make that explanation clearer.
>
> * We select this problem setting because it simultaneously meets two desiderata: $(i)$ a nonlinear function class, and $(ii)$ adaptivity to target "prior". On one hand, single-index model is arguably the simplest nonlinear extension of linear regression in which a nonlinear link function is applied to the response. On the other, this function class allows us to specify low-dimensional structure which the learning algorithm should adapt to. Moreover, for this function class, the statistical efficiency of simple algorithms (that can be implemented on the in-context examples) has been extensively studied.
>
> * If we are not concerned with point $(i)$ above, it is possible that a linear function class (with low-rank prior) can exhibit similar phenomenology, where a transformer with linear MLP layer can adapt to the class prior.
> More generally, we indeed conjecture a generalization bound that does not depend on the ambient dimensionality of the problem, but some notion of "intrinsic dimensionality" of the function class.
>
> ---
>
> **Empirical evidence and future work (Question 3)**
>
> * We conducted experiments that supports our theoretical claim on the statistical efficiency of ICL -- please see the .pdf document attached in our general response.
> Specifically, in Figure 1 we indeed observe a separation in the test loss between ICL and "prior-agnostic" algorithms such as kernel ridge regression, as the theory predicts.
>
> * We believe that the mechanism (employed in our theoretical analysis) of *feature extraction \& adaptivity to low-dimensionality* by MLP layer + *in-context function approximation* by attention layer could also appear in modern transformer training. However, since our statistical guarantees are derived for a particular architecture and gradient-based optimization procedure, it is likely that the quantitative scalings (e.g., dependence on context length, task diversity) have limited predictive power in realistic settings.
>
> ---
>
> We would be happy to clarify any further concerns/questions in the discussion period.

---

> > ### Comment · Reviewer_3HLy · 2024-08-11
> > **Thanks and follow-up questions about baselines**
> >
> > Thank you for your response. I think the proposed revisions (especially adding more background about single-index models and better contextualisation with prior work including the work of Raventos).
> >
> > I also welcome the proof-of-concept experiments that you have added, which I think improve the validity of this paper's contribution, assuming the experiments would be documented and discussed in detail in a future revision of the paper.
> >
> > At the moment I maintain my recommendation that the paper should be accepted.
> >
> > I have two further questions, both regarding baselines of one kind or another:
> >
> > 1. Regarding theoretical in-context sample complexity of the pre-trained transformer in this setting: Is it known what is the sample efficiency for other single-index model estimators *assuming knowledge of the low-rank subspace* like the pre-trained transformer? I understand why you have emphasised the comparison between your result and known sample efficiency for estimators without this prior knowledge. I think if you could *also* show that the transformer performs in some sense *as well as* other estimators with the same knowledge, that would strengthen your contribution further.
> >
> > 2. Regarding the empirical demonstration of the separation, how does the performance of each estimator you studied compare to theoretical predictions? I am assuming the transformers used for these experiments (described as 'our transformer' in your top-level rebuttal) matches fairly closely the subject of your theoretical analysis, so a direct comparison may be possible in some form. At the moment, the high-level story of 'sample efficiency separation' is corroborated by the experiments, but I'm asking if it is possible to show that the experiments also corroborate additional details of your theoretical results.

---

> > > ### Author Response · Authors · 2024-08-11
> > >
> > > We thank the reviewer for the positive evaluation and insightful followup questions.
> > >
> > > **Sample efficiency for single-index model estimators assuming knowledge of the low-rank subspace**
> > >
> > > If the relevant $r$-dimensional subspace is known to the learner, then information theoretically $n\simeq r$ samples would be sufficient and necessary (because intuitively speaking we are estimating $r$ parameters). However, this information theoretic limit does not entail the existence of efficient (polynomial time) algorithms.
> > > As for existing estimators, it is known that doing kernel ridge regression (KRR) on the low-rank subspace requires $n\gtrsim r^{P}$ samples, where $P$ is the degree of the nonlinear link function $\sigma_*$; this contrasts the $n\gtrsim d^P$ rate when the subspace is not known. The most sample-efficient estimator that we are aware of is the partial-trace procedure studied in [[Damian et al. 2024]](https://arxiv.org/pdf/2403.05529), which achieves a sample complexity of $n\gtrsim r^{k_*/2}$ (assuming knowledge of the $r$-dimensional subspace) where $k_*\le P$ is the *generative exponent* depending on structure of the link function; this tailored algorithm makes use of the tail of Gaussian data, and the obtained complexity matches the SQ lower bound under polynomial compute.
> > > Our generalization error bound for ICL entails a sufficient sample size of $n\gtrsim r^{\Theta(P)}$, which is parallel to the sample complexity achieved by KRR on the low-rank subspace. We however note that the exponent of $r$ in Theorem 1 does not exactly match the KRR rate due to additional multiplicative factors; closing this gap is an interesting future direction.
> > >
> > > **Empirical validation of theoretical predictions**
> > >
> > > At a *qualitative* level, our theoretical analysis suggests a statistical efficiency separation between pretrained transformers and baseline algorithms on the in-context examples when $r\ll d$, which is supported by Figure 1.
> > > Obtaining a more *quantitative* comparison, such as evaluating the tightness of our generalization bound in terms of polynomial dependence on $d,r$, would be more challenging computationally, as it requires the training of an ensemble of transformer models, each with training data of different dimensionality and context length (analogous to Figure 3 in [[Ba et al. 2023]](https://openreview.net/pdf?id=HlIAoCHDWW)). Following your suggestion, we are currently working on such an experiment with varying dimensionality to verify that the sample complexity of the in-context phase only depends on the subspace dimensionality $r$ rather than the ambient dimensionality $d$.

---

> > > > ### Comment · Reviewer_3HLy · 2024-08-11
> > > > **Thanks again**
> > > >
> > > > Thank you for your reply. I found these comparisons informative and they have helped me better understand and appreciate the contribution of this paper.

---

### Official Review · Reviewer_TSPc · 2024-07-13

**Soundness:** 3
**Presentation:** 3
**Contribution:** 2
**Rating:** 5
**Confidence:** 3

**Summary:**

This work shows that a single layer transformer can learn low dimensional structure in pretraining, and use it to learn a single index function in-context with a number of samples that scales in the size of the low-dimensional structure. Algorithm: The MLP layer is trained for one step of GD and the attention weights are trained for the rest of the time.

**Strengths:**

The paper shows that low dimensional structure can be leveraged in pre-training. It is generally well written.

**Weaknesses:**

1. The MLP is before the attention layer instead of after. (My initial impression was that the attention would do something like a projection onto the low dimensional subspace and the MLP would learn the non-linearity somehow). Can the authors say something about this?
2. There is no soft-max.
3. This is not true GD/SGD (and I think this is not obvious from the introduction).
    1. The MLP is only trained for one iteration.
    2.  The function is initialized to compute 0, and following that the $w_i$ are exactly “erased” by the regularization.
    3. The biases are then reset to be uniform on a unit interval.

**Questions:**

There are three interesting things in this model:
1. The non-linearity
2. The single index function
3. The low rank structure
But I am not sure how they individually play a role. The low rank structure I think serves the purpose of making the context length smaller, and isnt necessary, but what about the others?

Can you include some examples of function classes satisfying Assumption 1 with low P? For instance, what if my ground truth $\sigma_*$ is ReLU?

Can this be empirically verified? An experiment that shows that w aligns with S under true GD would be nice.

Is there some easy interpretation of Gamma? Without the MLP, for linear functions, it has been shown to be the inverse covariance of the data. With the MLP it appears to play a similar role.

**Limitations:**

Yes

---

> ### Author Rebuttal · Authors · 2024-08-07
>
> We thank the reviewer for the helpful feedback. We address the technical concerns below.
>
> ---
>
> **Nonstandard architecture and training procedure**
>
> We agree that Algorithm 1 deviates from the most standard training procedure in practice. We make the following remarks on our architecture and optimization algorithm.
>
> - **MLP before attention.** This simplified theoretical setting reflects the scenario where lower layers of the trained transformer act as data representations on top of which the upper layers perform ICL.
> For example, see Figure 1 in [[Guo et al. 2023]](https://openreview.net/pdf?id=ikwEDva1JZ): they considered a Transformer architecture with alternating MLP and attention layers and suggested that the earlier layers grasp the features of the true function, while the subsequent layers act as linear attention.
> Our architecture, with an MLP layer for feature learning followed by a linear attention module, idealizes such process occurring in more complicated Transformer.
> The same architecture has been analyzed in prior works on in-context learning of nonlinear features such as [[Kim and Suzuki 2024]](https://proceedings.mlr.press/v235/kim24af.html).
>
> - **Linear attention.** In the ICL theory literature, the Softmax attention is often simplified to linear attention to facilitate the optimization analysis. Such simplification can capture several prominent aspects of practical transformer training dynamics -- see [[Ahn et al. 2023]](https://openreview.net/pdf?id=0uI5415ry7) and references therein. Also, in our setting we already observe interesting in-context behavior using a linear attention. Extending our analysis to Softmax attention is an interesting future direction.
>
> - **Layer-wise training and one-step GD.**
> In the neural network theory literature, layer-wise training and reinitialized bias units are fairly standard algorithmic modifications when proving end-to-end sample complexity guarantees, as seen in many prior works on feature learning [[Damian et al. 2022]](https://proceedings.mlr.press/v178/damian22a/damian22a.pdf) [[Abbe et al. 2023]](https://proceedings.mlr.press/v195/abbe23a/abbe23a.pdf). Without such modifications, stronger target assumptions are generally needed, such as matching link function [[Ben Arous et al. 2020]](https://jmlr.csail.mit.edu/papers/volume22/20-1288/20-1288.pdf).
> In addition, training the representation for one gradient step (followed by SGD on the top layer) is a well-studied optimization procedure for two-layer neural networks (with the hope being that the "early phase" of representation learning already extracts useful features) -- see [[Ba et al. 2022]](https://openreview.net/pdf?id=akddwRG6EGi) [[Barak et al. 2022]](https://openreview.net/pdf?id=8XWP2ewX-im); but to our knowledge, we are the first to apply this dynamics to nonlinear transformers in the ICL setting and establish rigorous statistical guarantees.
>
> ---
>
> **Comments on the problem setting**
>
> The single-index setting is arguable the simplest nonlinear extension of linear regression in which a nonlinear link function is applied to the response. Moreover, for this function class, the statistical efficiency of simple algorithms (that can be implemented on the in-context examples) has been extensively studied. We therefore believe that it is a natural first step for the theoretical understanding of nonlinear ICL.
> As for the low-rank structure, our motivation is to introduce "prior" information of the function class which cannot be exploited by algorithms that directly learn from the test prompt, but can be adapted to during pretraining (see paragraph starting line 40 and 63). This enables ICL to outperform baseline algorithms such as linear/kernel regression.
>
> ---
>
> **"Can you include some examples of function classes satisfying Assumption 1?"**
>
> Our Assumption 1 requires the link function to be a bounded-degree polynomial, such as a Hermite polynomial of degree $P$. This assumption is analogous to those appearing in the feature learning literature, see [[Abbe et al. 2023]](https://proceedings.mlr.press/v195/abbe23a/abbe23a.pdf) and [[Ba et al. 2023]](https://openreview.net/pdf?id=HlIAoCHDWW).
>
> ---
>
> **"An experiment that shows that w aligns with S under true GD would be nice"**
>
> Following your suggestion, we have included numerical experiments demonstrating the alignment phenomenon in SGD with multiple steps. Please refer to Figure 2 in the pdf document in our general response.
>
> ---
>
> **"Is there some easy interpretation of Gamma?"**
>
> Intuitively speaking, $\Gamma$ plays the role of approximating (nonlinear) basis functions that express the true function. As shown in Section 4.2, $\Gamma$ is decomposed as $ADA$, where $A$ is analogous to the second-layer of a two-layer MLP which can approximate basis functions $\{g_i\}$. This is one crucial difference from the linear case, where the attention matrix only needs to learn the data covariance.
> We will provide more explanations on this in the revision.
>
> ---
>
> We would be happy to clarify any further concerns/questions in the discussion period.

---

> > ### Author Response · Authors · 2024-08-10
> >
> > Dear Reviewer,
> >
> > As the discussion period is ending in three days, we wanted to follow up to check whether our response has adequately addressed your concerns for an improved score. If there are any remaining questions or concerns, we would be happy to provide further clarification.
> >
> > Thank you for your time and consideration.

---

> > ### Comment · Reviewer_TSPc · 2024-08-13
> >
> > Thank you for your response. The empirical verification of alignment is nice. I also appreciate the use of the low rank structure as a means to learn something in pretraining. I will raise my score to a 5. I am still somewhat unsatisfied with the analysis of gradient descent. I understand that this is proof technique is somewhat common in the analysis of single hidden layer networks, but I believe this should be described as its own algorithm, since the main difficulty with analyzing gradient descent *is* dealing with the weights of the first layer.

---

### Author Rebuttal · Authors · 2024-08-07

Dear Reviewers and Area Chair,

We appreciate your time and effort to provide detailed comments on our paper. In response to the suggestions and comments on the experimental verification, we conducted the following experiments (the results of which are reported in the attached .pdf).

---

First, we compare the statistical efficiency of our studied transformer model trained via Algorithm 1, against two baseline methods: kernel ridge regression, and two-layer neural networks trained by gradient descent.

We set $d=64,m=8000$, and the true function is $\mathrm{He}_2(\langle\beta,x\rangle)$, where $\beta$ is randomly drawn from $r=2$-dimensional subspace in $\mathbb{R}^{64}$. At test time, context length is altered from $2^6$ to $2^{10}$, and we compared our Transformer, a two-layer neural network (MLP) with width $m=8000$, and kernel ridge regression using radial basis function kernel.
Hyperparameters for our Algorithm 1 are set as $T_1=300,000,N_1=10,000,\eta_1=1,000,T_2=20,000,N_2=1,500,$ and $\lambda_2=0.01$.

We plotted the test error for each context length in Figure 1 in the PDF, where we observe that Transformer indeed achieves lower test error.

---

Second, we demonstrate the alignment between the weights of the MLP layer $w$ and the target $r$-dimensional subspace, when the transformer is trained by standard SGD with multiple steps.

The target function and $d,r,m$ are the same as the first experiment. Here we trained MLP layer $w$, bias $b$ and $\Gamma$ in our architecture *simultaneously* via single-pass SGD with weight decay in $T_1=300,000$ steps, with context length $N_1=5,000$.
The parameters are initialized as follows (here we employ a *mean-field* scaling to encourage feature learning).
$w_j\sim \mathrm{Unif}(\mathbb{S}^{d-1})$, $\eta=0.1m,\lambda=10^{-6}$.  $b_j\sim \mathrm{Unif}[-1,1]$, $\eta=0.1m,\lambda=10^{-6}$.  $\Gamma= 0.01I_m/m$, $\eta=0.00001/m,\lambda=10^{-3}$.

---

We hope that these experiments can address the reviewers' concern on the lack of empirical support.

Please refer to our detailed response to each reviewer where we respond to the individual comments.

---

### Author Response · Authors · 2024-08-14
**Summary of our responses and revisions**

Dear Area Chair,

As the rebuttal period draws to a close, we would like to briefly summarize our responses and revisions to our manuscript to address the reviewers' concerns.
- **Experimental verification.** Several reviewers noted that the manuscript could be strengthened by an experiment that supports our theoretical findings. Our [general response](https://openreview.net/forum?id=uHcG5Y6fdB&noteId=Ygau6T3C2K) presented experiments that demonstrate a separation in statistical efficiency.
- **Motivation of the studied setting.** Following the reviewers' suggestions, we have expanded our discussion on the background on single-index models. This *nonlinear* function class is common in the NN theory literature (see our [reply to reviewer 3HLy](https://openreview.net/forum?id=uHcG5Y6fdB&noteId=K3Xao71geS)), but to our knowledge, we are the first to provide end-to-end learning guarantees (with gradient-based training) in the more challenging *in-context* scenario, going beyond prior works on *linear* ICL. This idealized setting allows us to rigorously study $(i)$ how pretrained transformers adapt to the prior information of the function class, and $(ii)$ how the MLP block can act as a feature extractor.
- **Training procedure.** We agree with the reviewers that Algorithm 1 deviates from the most standard training paradigm.  We have added further explanations that this layer-wise training (with one-step GD) is commonly used in the theoretical analysis of representation learning [[Damian et al. 2022]](https://proceedings.mlr.press/v178/damian22a/damian22a.pdf) [[Ba et al. 2022]](https://proceedings.neurips.cc/paper_files/paper/2022/file/f7e7fabd73b3df96c54a320862afcb78-Paper-Conference.pdf) to handle the non-convexity, but we are the first to adapt such a procedure to the ICL setting (see our [reply to reviewer UaEF](https://openreview.net/forum?id=uHcG5Y6fdB&noteId=6WcM6yn1y9)).
---
We appreciate the reviewers' feedback which helped us improve the manuscript, and we hope that our nontrivial theoretical contributions can be recognized (in light of our rebuttal and revisions).

---

### Decision · Program_Chairs · 2024-09-25

**Decision:**

Accept (poster)

**Comment:**

The auhtors investigate the ability for a single-layer softmax attention transformer to in-context learn a low-dimensional target function by training on in-context examples.  The authors leverage a somewhat unnatural variant of GD which involves a single step of (non-convex) GD followed by a re-initialization and ridge regression step, an approach that has appeared prior in the deep learning theory literature.  Using this they show that the sample complexity depends only on the intrinsic dimension of the target function rather than the ambient dimension.

There was some support among the reviewers, with some stronger and some weaker. I'm inclined to agree with the positive reviewers: I think this work provides a nice contribution to our understanding of the sample complexity of canonical learning problems translated to the in-context learning setting.  I recommend acceptance.